



# Sediment and carbon accumulation in a glacial lake in Chukotka (Arctic Siberia) during the late Pleistocene and Holocene: Combining hydroacoustic profiling and down-core analyses

Stuart A. Vyse[1,2], Ulrike Herzschuh[1,2,4], Gregor Pfalz[1,2], Lyudmila A. Pestryakova[5], Bernhard Diekmann[1,2,3], Norbert Nowaczyk[6], Boris K. Biskaborn[1]

[1]Alfred Wegener Institute Helmholtz Centre for Polar and Marine Research, Research Unit Potsdam, Telegrafenberg A45, 14471 Potsdam, Germany

[2]Institute of Environmental Science and Geography, University of Potsdam, Potsdam, Germany

[3]Institute of Geosciences, University of Potsdam, Potsdam, Germany

[4]Institute of Biochemistry and Biology, University of Potsdam, Potsdam, Germany

[5]Northeastern Federal University of Yakutsk, Yakutsk, Russia

[6]Helmholtz-Centre Potsdam GFZ, Climate Dynamics and Landscape Evolution, Telegrafenberg, 14473 Potsdam, Germany

*Correspondence to:* **Stuart Andrew Vyse** (*stuart.vyse@awi.de*); **Boris K. Biskaborn** (*boris.biskaborn@awi.de*)

## Abstract

Lakes act as important sinks for inorganic and organic sediment components. However, investigations of sedimentary carbon budgets within glacial lakes are currently absent from Arctic Siberia. The aim of this paper is to provide the first reconstruction of accumulation rates, sediment and carbon budgets from a lacustrine sediment core from Lake Rauchuagytgyn, Chukotka (Arctic Siberia). We combined multiple sediment-biogeochemical and sedimentological parameters from a radiocarbon-dated 6.5 m sediment core with lake basin hydroacoustic data to derive sediment stratigraphy, sediment volumes, and infill budgets. Our results distinguished three principal sediment and carbon accumulation regimes that could be identified across all measured environmental proxies including Early MIS2 (ca. 29–23.4 cal. ka BP), Mid-to-late MIS2 (ca. 23.4–11.5 cal. ka BP), and Holocene (ca. 11.5– present). Estimated organic carbon accumulation rates (OCARs) were higher within Holocene sediments (average 3.53 g OC m$^{-2}$ a$^{-1}$) than Pleistocene sediments (average 1.09 g OC m$^{-2}$ a$^{-1}$) and are similar to those calculated for boreal lakes from Quebec and Finland and Lake Baikal but significantly lower than Siberian thermokarst lakes and Alberta glacial lakes. Using a bootstrapping approach, we estimated the total organic carbon pool to 0.26 ± 0.02 Mt and a total sediment pool of 25.7 ± 1.71 Mt within a hydroacoustically derived sediment volume of ca. 32990557 m$^3$. The total organic carbon pool is substantially smaller than Alaskan Yedoma, thermokarst lake sediments, and Alberta glacial lakes but shares similarities with Finnish boreal lakes. Temporal variability in sediment and carbon accumulation dynamics at Lake Rauchuagytgyn is controlled predominantly by palaeoclimate variation that regulates lake ice-cover dynamics and catchment glacial, fluvial and permafrost processes through time. These processes, in turn, affect catchment and within-lake primary productivity as well as catchment soil development. Spatial differences to other lake systems at a trans-regional scale likely relates to the high-latitude, mountainous location of Lake Rauchuagytgyn.



## 1. Introduction

Lakes represent key sentinels of environmental change and can respond rapidly to changes in environmental
conditions (Adrian et al., 2009). Lakes act as sinks of clastic sediment derived from weathering of their
catchments and as such gradually accumulate sediment mass over time (Hinderer and Einsele, 2001). They also
constitute a significant net sink of carbon, as they can accumulate organic and inorganic carbon within their
sediments derived from allochthonous (lake external) and autochthonous (lake internal) pathways (Ferland et al.,
2012; Dean and Gorham 1998; Kortelainen et al., 2004; Sobek et al., 2014). Recent syntheses suggest that
global lakes represent a carbon sink of around 0.2–0.6 Pg C $y^{-1}$ (Cole et al., 2007; Battin et al., 2009). In turn,
inland waters can also act as major sources of the greenhouse gases $CO_2$ and $CH_4$ and thereby contribute
further to global climate change (Anthony et al., 2014).

Lake sediment cores contain sedimentological and biogeochemical proxies of environmentally driven
sedimentation and carbon dynamics (Birks and Birks, 2006; Biskaborn et al., 2019; Smol et al., 2002). When
sedimentation rate data is available via dating of sediment cores, estimations of rates of sediment mass and
carbon accumulation can be reconstructed (Ferland et al., 2012). Moreover, if the sediment volume stored within
a lake basin can be estimated, sediment and carbon pools can be obtained that allows the lake function as a
sediment and carbon sink to be assessed (Campbell et al., 2000; Lehman., 1975; Munroe & Brencher., 2019;
Pajunen., 2000).

The region of Chukotka (Arctic Siberia) represents an important area with limited lacustrine environmental
reconstructions (Lozhkin and Anderson, 2013). Though a number of regional records across Arctic Siberia are
becoming increasingly prevalent (Biskaborn et al., 2012; Diekmann et al., 2016; Melles et al., 2012; Subetto et
al., 2017), studies have often neglected reconstructions of accumulation rates. This is largely due to limited age-
controls related to problematic radiocarbon dating of organic poor systems that often lack dateable macrofossil
remains and possess low dating resolutions (Lozhkin et al. 2016, Strunk et al., 2020). Current eastern Siberian
research has primarily focussed on the reconstruction of Holocene accumulation rates of carbon using sediment
cores derived from thermokarst lake systems (Anthony et al., 2014). Such studies have shown that lake systems
may operate in transitional modes between carbon sink and source stages depending on the prevailing climatic
background conditions (Anthony, et al., 2014). Recent syntheses have characterized the sedimentological
characteristics and carbon budgets of Bykovsky peninsula lagoons of northern Yakutia, eastern Siberia and
carbon inventories within Alas and Yedoma deposits from Central Yakutia and are interpreted in terms of
palaeoenvironmental variability since the Pleistocene (Jenrich et al., in review; Windirsch et al., 2020). Such
works have also been extended to drained thermokarst lake basins (DTLB) and Pleistocene Yedoma deposits in
north-western Alaska (Jongejans et al., 2018). The reconstruction of accumulation rates in these syntheses has
however been avoided due to significant reworking of carbon material within permafrost landscapes (Strunk et
al., 2020; Windirsch et al., 2020). The role however, of Arctic Siberian glacial lakes as sediment and carbon sinks
has not yet been accounted for.

Several, broad regional studies of dated sediment cores have been applied in an attempt to calculate
accumulation rates and carbon pools within small glacial lakes in the Uinta mountains, north America (Munroe
and Brencher, 2019) as well as in lakes in southern Greenland (Anderson et al., 2009) and from large glacial
lakes in Alberta, Canada (Campbell et al., 2000). A major drawback to these studies however is the absence or
oversimplification of sediment volume estimation which does not account for lateral variations in underlying





sediment stratigraphy resulting from sediment focussing and winnowing (Ferland et al., 2012). Campbell et al., (2000) suggested an empirical equation for the estimation of sediment volume by simplifying lake sediments to

an oblate semi-spheroid to account for sediment thinning at the lake margins. However, such approaches have been shown to lead to overestimations of sediment volumes and carbon pools up to four times the true value (Ferland et al., 2012; Munroe and Brencher, 2019). Hence the lack of efficient derivation of sediment volumes represents a major disadvantage of current works. Recent studies that include seismic or hydroacoustic based appraisals of sediment volume account for these disadvantages and show a significant improvement of pool

calculations (Einola et al., 2011; Ferland et al., 2012; Pajunen et al., 2000). Such methods are simple, rapid and give significant additional insight into lake bathymetry and basin-wide sediment stratigraphy that permits the retrieval of sediment thicknesses and volumes (Ferland et al., 2012). Recent hydroacoustic based studies from central and western Siberia have concentrated on determining Holocene and late Pleistocene sediment distributions and volumes within glacial lakes and have proven highly useful for understanding glacial history in

these regions (Haflidason et al., 2019; Lebas et al., 2019). The inclusion of sediment budgets within investigations of carbon pools is important as it allows the future storage capacity of lakes to be estimated (Hinderer and Einsele, 2001).

This paper aims to provide a first estimation of late Pleistocene-Holocene sediment and carbon accumulation within an Arctic glacial lake basin within the remote Chukotka region (Arctic Siberia). This includes the first

reconstruction of Pleistocene-Holocene sediment and organic carbon accumulation rates and pools provided for Chukotka derived by integrating proxy analyses from a high-resolution dated sediment core with high-resolution hydroacoustic data for sediment volumes. Our results are interpreted in the context of regional palaeoenvironmental variability during the late Pleistocene and are compared to trans-regional studies from Yakutia (east Siberia), North America and Europe. As such, our research questions are:

1. How has the palaeoenvironmental variability in Chukotka and resulting sedimentological processes since Marine Isotope Stage 2 (MIS2) influenced the sediment and carbon accumulation dynamics within an Arctic glacial lake?

2. How much sediment and carbon is stored within the Lake Rauchuagytgyn basin and how is this proportioned between the glacial and interglacial periods?

3. How do accumulation rates compare with other systems at a trans-regional scale?






## 2. Study area

Lake Rauchuagytgyn (67.7922° N, 168.7312° E) is situated within the glacially eroded V-shaped, Rauchua

mountain valley within the north-western Anadyr Mountains of Chukotka (Arctic Siberia) where elevations reach
up to 1600 m a.s.l (Fig. 1).  The lake lies at an elevation of ca. 625 m a.s.l with a surface area of 6.1 km$^2$ and
maximum water depth of 36 m. The lake is supplied by fluvial inflows at the lakes southern margin (Rauchua
River) and via an alluvial fan at the lakes south-eastern margin. One major outflow drains the lake to the north
and several minor streams drain the basin sides. The basin is divided into several sub-regions (southern and

south-eastern Inflows, southern sub-basin, northern sub-basin and northern shelf) based on water depth
characteristics. The catchment area comprises 214.5 km$^2$. The bedrock surrounding the lake and within the
catchment is predominantly composed of cretaceous extrusive and intrusive igneous facies consisting of silicic-
intermediate lithologies dominated by Andesite (Zhuravlev and Kazymin, 1999). Catchment evidence for
glaciation includes moraine structures to the north of the lake that denote the maximum extent of glaciation

(Glushkova, 2011). Several glacial cirques are found within the catchment (Glushkova, 2011). The Arctic
continental climate of the area is characterised by mean annual temperatures of -11.8 °C and average July and
January temperatures of 13 °C and -30 °C with low annual precipitation of ca. 200 mm (Menne et al., 2012).  A
surface ice-layer is present on lakes in this area from October to early July and likely reaches a winter maximum
thickness of ca. 1.8 m (Nolan et al., 2002). Catchment vegetation is dominated by open herb- and graminoid

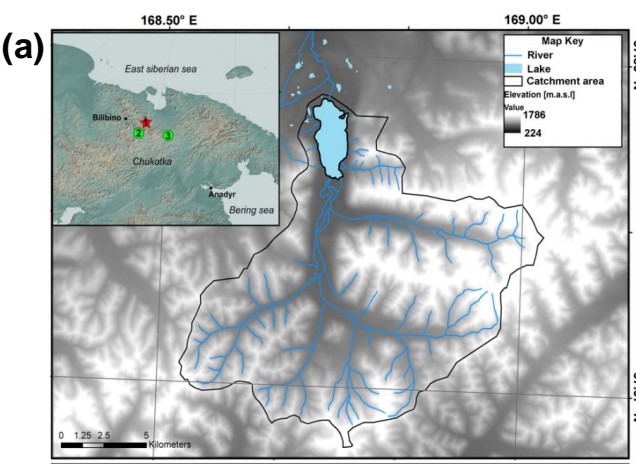

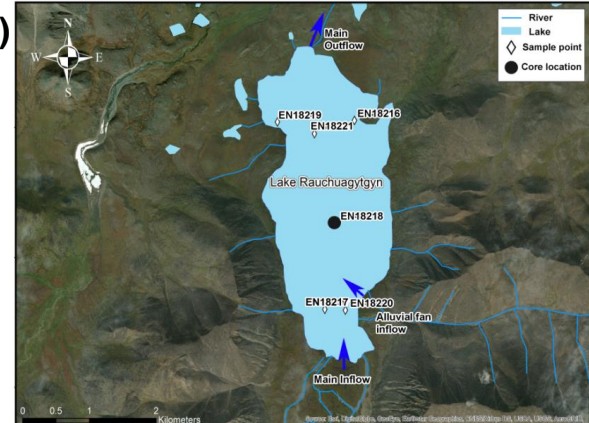

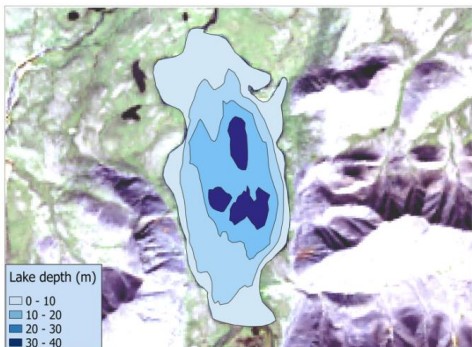

**Figure 1. (a)** Overview map of the Lake Rauchuagytgyn catchment plotted with a digital elevation map of the Anadyr mountains. Also shown is the situation of Lake Rauchuagytgyn (1) compared to other studied regional lakes: (2) Lake Ilirney and (3) Lake El´gygytgyn (ESRI 2020). **(b)** Zoomed image of the lake and surrounding features. Shown are the location of surface sampling points and the location of sediment core EN18218 (black dot) as well as an alluvial fan at the lakes south-eastern margin (ESRI 2020). Flow directions of major inflow and outflow river systems are shown (blue arrows). **(c)** Simplified water depth map adapted from Huang et al., 2020.





tundra with no vegetation at higher elevations. Some forest tundra is found at lower elevations and adjacent to
       river valleys (Shevtsova et al., 2020).

## 3. Materials and methods:

### 3.1 Fieldwork

       Fieldwork took place at Lake Rauchuagytgyn in July 2018. An SES-2000 compact parametric sub-bottom profiler
was used to hydroacoustically survey the basin to identify major acoustic units (AUs) and sediment boundaries
       and to locate an optimal coring location (Innomar technologies GmbH). A low frequency (8–10 kHz) and low
       pulse interval (2 seconds) was used for high sediment penetration to retrieve sediment architecture. A high
       frequency (100 kHz) and high pulse interval (253 microseconds) allowed derivation of lake bathymetry. In total,
       hydroacoustic profiles containing > 23000 data points were retrieved.

Sediment core EN18218 (ca. 653 cm) (Fig. 1) was collected from the southern sub-basin using an UWITEC
       piston coring system. The core was retrieved in parallel overlapping ca. 3 m segments, cut into 1 m sections and
       transported in transparent 60 mm PVC tubes for further processing. Further core penetration and retrieval was
       prevented by sand-pebble material at the core base most likely representing the basal sediment within the basin.

       Epilimnetic water samples were collected in 2016 (Huang et al., 2020) and in 2018 (this study) from multiple
surface locations and were immediately measured for pH, conductivity and oxygen content (WTW Multilab 540,
       Germany). Water sub-samples were then filtered (0.45 μm) and transported for analyses of dissolved organic
       carbon (DOC), anions and cations (Table S1).

### 3.2 Laboratory analyses

#### 3.2.1 Core processing

Sediment cores were split into halves at the Alfred Wegener Institute Helmholtz Centre for Polar and Marine
       Research (AWI) with one half being immediately subsampled for radiocarbon dating and subsequently at ca. 10
       cm resolution for proxy analyses. The other half was cleaned and logged for macroscopic lithological changes
       that permitted the derivation of three lithological units (LU-I to LU-III) and prepared for non-destructive scanning
       approaches.

#### 3.2.2 Radiocarbon dating and age-depth modelling

       Due to the lack of suitable plant remains and low organic content of the retrieved sediment core, 25 bulk
       sediment samples and one surface sample (0–0.5 cm) were dated for radiocarbon using accelerator mass
       spectrometry (AMS) with the Mini Carbon Dating System (MICADAS) at AWI Bremerhaven, Germany (Table 1).
       Samples were prepared following the standard MICADAS chemical pretreatment procedure (see Gentz et al.,
2017 for details).

       The age-depth relationship was established by using the open-source MATLAB software package 'Undatable'
       (Lougheed and Obrochta, 2019). For modeling we used 23 bulk sediment samples. Two samples (Lab-ID: AWI -
       3001.1.1; AWI - 3002.1.1) were slightly older than their successive dates further down, which suggest reworking
       in these depths (81.25 cm and 114.75 cm, respectively). We treated these two dates as outliers and excluded



them from the modeling process. All dates were calibrated to 'calibrated years Before Present' (years before 1950 C.E., 'cal. a BP') using the IntCal20 calibration curve (Reimer et al., 2020). In order to account for an existing age offset, we added the results from the surface sample (785 ± 31 $^{14}$C years BP) to the software as reservoir age (Undatable settings: nsim = $10^6$, bootpc =70, xfactor = 0.1). Median sedimentation rates (SR) (mm a$^{-1}$) were extracted from the final age-depth model. In order to account for age-depth model uncertainty in

subsequent calculations, age-depth model derived 1σ and 2σ SR uncertainty ranges were also extracted.





**Table 1.** Radiocarbon dates from sediment core EN18218. Calibrated median ages and 2σ confidence intervals were calculated with CALIB 8.2 (Stuiver et al., 2020) using the IntCal20 (Reimer et al., 2020) calibration curve. Calibrated ages are given as calibrated years Before Present (years before 1950 C.E., 'cal yr BP'). The surface sample used for down-core correction of the reservoir effect was not calibrated. Samples in italic and marked with an asterisk (*) were excluded from the age-depth modeling.


| Lab code | Sample ID | Composite depth (cm) | Radiocarbon age with error ($^{14}$C years BP) | Calibrated median age (cal. $^{14}$C years BP) | Calibrated 2σ age range (cal. $^{14}$C years BP) | Sample type |
|---|---|---|---|---|---|---|
| AWI - 5627.1.1 | EN18218-1 Surface 0-0.5 cm | 0.25 | 785 ± 31 | - | - | Bulk, TOC |
| AWI - 2998.1.1 | EN18218-2_0-100_20-20.5 | 18.75 | 2787 ± 33 | 2887 | 2783 – 2961 | Bulk, TOC |
| AWI - 2999.1.1 | EN18218-2_0-100_36.5-37 | 35.25 | 3629 ± 33 | 3942 | 3842 – 4080 | Bulk, TOC |
| AWI - 3000.1.1 | EN18218-2_0-100_61-61.5 | 59.75 | 3832 ± 33 | 4233 | 4098 – 4404 | Bulk, TOC |
| *AWI - 3001.1.1** | *EN18218-2_0-100_82.5-83* | *81.25* | *4985 ± 33* | *5702* | *5602 – 5882* | *Bulk, TOC* |
| *AWI - 3002.1.1** | *EN18218-2_100-200_116-116.5* | *114.75* | *5417 ± 34* | *6234* | *6119 – 6295* | *Bulk, TOC* |
| AWI - 3003.1.1 | EN18218-2_100-200_140-140.5 | 138.75 | 5074 ± 34 | 5816 | 5739 – 5908 | Bulk, TOC |
| AWI - 3004.1.1 | EN18218-2_100-200_164-164.5 | 162.75 | 5382 ± 34 | 6201 | 6008 – 6284 | Bulk, TOC |
| AWI - 3005.1.1 | EN18218-2_100-200_189-189.5 | 187.75 | 5852 ± 34 | 6672 | 6559 – 6746 | Bulk, TOC |
| AWI - 3006.1.1 | EN18218-2_200-240_222.5-223 | 221.75 | 6472 ± 35 | 7370 | 7310 – 7460 | Bulk, TOC |
| AWI - 3007.1.2 | EN18218-3_0-100_15-15.5 | 248.75 | 8872 ± 37 | 10011 | 9779 – 10177 | Bulk, TOC |
| AWI - 3008.1.1 | EN18218-3_0-100_37-37.5 | 270.75 | 9085 ± 37 | 10236 | 10185 – 10362 | Bulk, TOC |
| AWI - 3009.1.1 | EN18218-3_0-100_59.5-60 | 293.25 | 9516 ± 38 | 10827 | 10604 – 11074 | Bulk, TOC |
| AWI - 3010.1.1 | EN18218-3_0-100_83-83.5 | 316.75 | 9901 ± 39 | 11298 | 11214 – 11595 | Bulk, TOC |
| AWI - 3011.1.1 | EN18218-3_100-200_105-105.5 | 338.75 | 10197 ± 39 | 11866 | 11655 – 11994 | Bulk, TOC |
| AWI - 3012.11 | EN18218-3_100-200_129-129.5 | 362.75 | 11687 ± 30 | 13550 | 13481 – 13598 | Bulk, TOC |
| AWI - 3013.1.1 | EN18218-3_100-200_150-150.5 | 383.75 | 12205 ± 46 | 14112 | 14026 – 14310 | Bulk, TOC |
| AWI - 3014.1.1 | EN18218-3_100-200_171-171.5 | 404.75 | 13017 ± 48 | 15596 | 15371 – 15760 | Bulk, TOC |
| AWI - 3015.1.1 | EN18218-3_200-292_210-210.5 | 443.75 | 14330 ± 52 | 17427 | 17197 – 17803 | Bulk, TOC |
| AWI - 3016.1.1 | EN18218-3_200-292_239-239.5 | 474.75 | 15686 ± 48 | 18941 | 18847 – 19068 | Bulk, TOC |
| AWI - | EN18218-3_200- | 503.75 | 17708 ± 56 | 21473 | 21191 – 21783 | Bulk, |



| 3017.1.1 | 292_270-270.5 | | | | | TOC |
|---|---|---|---|---|---|---|
| AWI - 3018.1.1 | EN18218-4_0-100_35-35.5 | 536.25 | 18000 ±55 | 21945 | 21740 – 22117 | Bulk, TOC |
| AWI - 3019.1.1 | EN18218-4_0-100_64.5-65 | 565.75 | 22649 ±66 | 27059 | 26479 – 27226 | Bulk, TOC |
| AWI - 3020.1.1 | EN18218-4_0-100_95-95.5 | 596.25 | 21786 ±204 | 26077 | 25725 – 26442 | Bulk, TOC |
| AWI - 3021.1.1 | EN18218-4_100-163_123-123.5 | 624.25 | 25689 ±325 | 29941 | 29205 – 30735 | Bulk, TOC |
| AWI - 3022.1.1 | EN18218-4_100-163_145-145.5 | 646.25 | 25081 ±300 | 29393 | 28780 – 29978 | Bulk, TOC |





### 3.2.3 Sediment-geochemistry, Magnetic Susceptibility and Grain-Size

Semi-quantitative sediment geochemical data were obtained by X-ray fluorescence line-scanning (XRF) of EN18218 halves. XRF was carried out using an Avaatech core-scanner with a Rh X-ray tube at 0.75 mA and 1.5 mA for 10 and 15 seconds, at 10 kV (no-filter) and 30 kV (Pd-thick filter) at the Federal Institute for Geosciences and Natural Resources (BGR), Germany. A scanning resolution of 5 mm was chosen. The main rock-forming (Aluminium (Al), Silicon (Si), Calcium (Ca), Potassium (K), Titanium (Ti), Rubidium (Rb), Strontium (Sr), Zircon (Zr)) and redox/productivity linked elements (Manganese (Mn), Iron (Fe), Bromine (Br)) were selected for further
processing. Of these single elements we focus primarily on Br, Ca, K and Ti within this study. The element ratio of K/Ti is used as proxy for clay input and weathering (Arnaud et al., 2012; Cuven et al., 2010; Kilian et al., 2013; Marshall et al., 2011). Zr/K is used as a grain-size proxy and Mn/Fe for redox conditions (Baumer et al., 2020; Cuven et al., 2010). Br/Al is utilized alongside Br as a proxy for productivity (Lenz et al., 2020). Ratios of element intensities were log transformed using the additive log ratio (ALR) transformation within the package
"compositions" (version 1.40) in R (Aitchison, 1984; van den Boogaart et al., 2020; Weltje and Tjallingii, 2008). Magnetic susceptibility measurements (MS) were carried out at 1 mm intervals using a Bartington MS2E spot reading sensor integrated in a fully automatic split-core logging device, developed at the GFZ Potsdam, Germany (Bartington Instruments Ltd). Core images and commission on Illumination (CIE) l* (lightness), b* (blue-yellow) colour data were retrieved using an Avaatech line scan camera. XRF and MS data were smoothed using 5- and
15-point running means respectively.

Sediment grain-size analysis was performed on 63 core samples and four surface samples following a 3-week hydrogen peroxide ($H_2O_2$) treatment to remove organic matter. Samples were homogenised using an elution shaker for 24 hours and split into eight subsamples. At least three subsamples were analysed thrice providing overall nine individual measurements using a Malvern Mastersizer 3000 laser diffraction particle analyser
(Malvern Panalytical Ltd). The nine measurements were then averaged to produce a grain-size distribution for each sample. Data was processed using „GRADISTAT" 8.0 software (Blott and Pye, 2001). Intervals of 2 mm–63 µm, 63–2 µm and <2 µm, were used to define percentages of sand, silt and clay respectively. The folk and ward method was used for mean grain-size calculation.

### 3.2.4 Dry bulk density, elemental analyses and accumulation rate calculations

Water contents (wt%) and dry bulk densities (DBD in g $cm^{-3}$) were determined using the volumetric approach of Avnimelech et al, (2001) on 63 samples of known volume obtained with a one $cm^3$ cube during subsampling and calculated from weight-loss following freeze-drying (See also Pajunen, 2000). The total sediment mass accumulation rate (MAR) in g $cm^{-2}$ $a^{-1}$ was calculated according to the ocean drilling project (ODP) methodology (http://www-odp.tamu.edu/) (Eq. 1).

$$MAR = DBD \times SR$$

210 (1)

Where DBD is dry bulk density and SR is age-depth model derived sedimentation rate (cm $a^{-1}$). MAR uncertainty ranges were determined by applying Eq. 1 to sedimentation rate 1σ and 2σ uncertainty ranges extracted from the age-depth model.



dried and milled sub-samples were subsequently analysed for total organic carbon (TOC) and total inorganic
carbon (TIC) using a Vario SoliTOC cube elemental analyser following combustion at 400 °C (TOC), and 900 °C
(TIC) (Elementar Corp., Germany). Total carbon (TC) was calculated as the total sum of TOC and TIC. Total
sulphur (TS) was measured using an Eltra Carbon and Sulfur determinator (Eltra GmbH, Germany). Device
results are given as weight percent (wt%) in relation to sample mass. Organic carbon accumulation rates
(OCARs) were then calculated by combining TOC (wt%) & DBD to produce organic carbon content per volume (g
OC cm$^{-3}$) and then with SR for each depth (Eq. 2).

$$OCAR = (DBD \times (\frac{TOC}{100})) \times SR$$

(2)

OCARs are reported as grams of organic carbon per square meter per year (g OC m$^{-2}$ a$^{-1}$) for comparison with
published organic carbon accumulation rates. OCAR uncertainty ranges were calculated identically to MAR
uncertainty ranges.

Average SR, MAR and OCARs were determined for each lithological unit (LU) and the composite sediment
succession.

### 3.3 Data analyses

### 3.3.1 Estimation of lake basin sediment volumes

Hydroacoustic profiles were processed and major hydroacoustic boundaries delineated using ISE2 2.9.5
software software (Innomar technologies GmbH). Boundary depths were extracted using the ISE2 "Depth
extraction" tool and were imported into ArcMap 10.5.1 software. Data gaps between measurement points and
profiles were interpolated using the "Topo to Raster" tool to produce interpolated depth surfaces to the sediment-
water interface and to basin-wide boundary surfaces (Supplement Fig. S2). The interpolated sediment-water
interface depths were subtracted using the raster calculator tool from interpolated boundary depths to estimate
total basin and acoustic unit sediment thicknesses (T$_{sediment}$). Sediment volumes (V$_{sediment}$) were subsequently
derived using the "Surface Volume" tool following subtraction of the lake water volume (V$_{water}$) (Table 2).

**Table 2.** Overview of general acoustic (AU) and lithological unit (LU) divisions with associated depth and age ranges and age-
depth model calculated average sedimentation rates (SR). Additionally shown are basin properties calculated using ArcMap
10.5.1 software including Interpolated water and sediment thicknesses (T$_{water}$, T$_{sediment}$) and water and sediment package
volumes (V$_{water}$, V$_{sediment}$).

| Acoustic Unit (AU) | Lithological Unit (LU) | Depth range at core location (cm) | Age range (cal. ka BP) | Average Sedimentation rate (SR) (mm a$^{-1}$) | Average Interpolated thickness (m) (T$_{water}$, T$_{sediment}$) | Estimated volume (m$^3$) (V$_{water}$, V$_{sediment}$) |
|---|---|---|---|---|---|---|
| Water | Water | - | - | - | 14.5 | 90103775 |
| 1 | I | 0 - 341 | 0 - 11.5 | 0.36 | 2.44 | 14935205 |





| 2 | II | LU-II = 342 - 560 | 11.5 – 23.4 | 0.20 | 2.96 | 18055352 |
| | III | LU-III = 561 - 653 | 23.4 – ca. 29 | 0.17 | | |

### 3.3.2 Estimation of carbon and sediment pools using bootstrapping

In order to estimate sediment (Sed$_{pool}$, Mt) and carbon pools (TOC$_{pool}$, Mt) at Lake Rauchuagytgyn, we used an
established bootstrapping approach modified from Strauss et al., (2013), Jongejans et al., (2018) and Windirsch
et al., (2020) that excluded ice-wedges not present within the lake sediments and was enhanced to include
sediment volumes estimated from GIS-based methods (Eq. 3 & 4).

$$Sed_{pool} = \frac{V_{sediment} \times DBD}{10^6}$$

(3)

$$TOC_{pool} = \frac{V_{sediment} \times DBD \times (\frac{TOC}{100})}{10^6}$$

(4)

The bootstrapping approach was carried out using the "boot" (version 1.3-25) and "Bootstrapping-permafrost-OC"
packages (Jongejans and Strauss, 2020) in the R environment and included 10,000 iterations of random
sampling and replacement of values (Davison and Hinkley, 1997). Combined DBD and TOC values were used
for carbon pool estimation and simply DBD for the sediment pool. The approach was applied firstly to the entire
sediment core and basin sediment volume and then individually for the LU-II/LU-III (AU2) and LU-I (AU1)
sediment packages. The mean sediment and carbon pools (in megatons, Mt), densities (Kg m$^{-3}$) and associated
standard deviations and confidence intervals of all iterations were computed (See Strauss et al., 2013 for details)
(Supplement S4 Code).

### 3.3.3 Multivariate statistics

In order to reduce data dimensionality and increase interpretability of presented physicochemical sediment
stratigraphic and accumulation rate data obtained from core EN18218, principal component analysis (PCA) was
carried out following data centering and standardization by dividing by standard deviation using the „prcomp"
function in R (R Core team, 2013). Probability ellipses were drawn in the resulting PCA biplot for samples
included within each lithological unit.




## 4. Results

### 4.1 Basin hydroacoustic stratigraphy and sediment distributions

Hydroacoustic and bathymetric data (Fig. 2) permit the division of the lake basin into four morphological regions:
northern shelf, northern sub-basin, southern sub-basin and southern inflow (Fig. 3a). The basin is characterised by steep east and west margins and a central basin bathymetric high. A pronounced river delta is present at the southern lake margin at the major lake Inflow (Fig. 1b). An alluvial fan at the lakes south-eastern margin represents drainage from a steep, lake-proximal, west-east oriented valley (Fig. 1b). Hydroacoustic data allowed effective imaging of much of the sediment infill except within inflow proximal profiles where acoustic blanking
limited data retrieval (Fig. 2b).

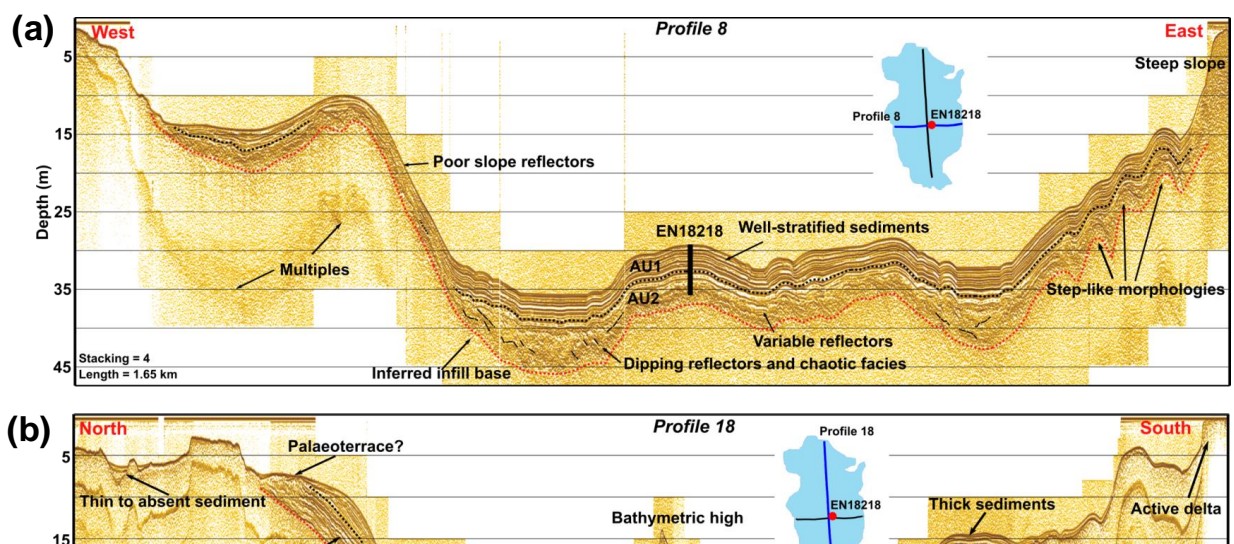

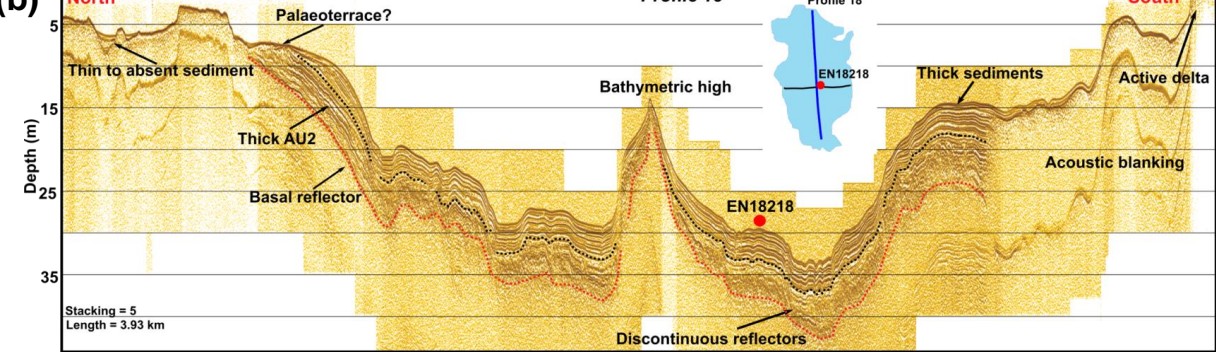

**Figure 2.** Interpreted hydro-acoustic profiles from Lake Rauchuagytgyn (Profile 8 and Profile 18). Both profiles pass close to the drilling location of EN18218. Shown are inferred major hydroacoustic boundaries defined that are connected with major lithological changes recorded by EN18218 and interpreted across the basin.

Average sediment thickness of the entire lake basin infill is estimated as ca. 5.33 m with the thickest infill predominantly within the southern sub-basin and generally correlates with greater water depth. Average total sediment thickness equates to a total sediment volume of ca. 32990557 $m^3$ (0.033 $km^3$) (Fig. 3d). Sediment thicknesses decrease towards the basin margins, particularly within the northern-shelf where water depths < 7.5
m show reduced sedimentation and evidence of ponded sediment lenses and erosion (Figs. 2b, 3b–d). Comparison of hydroacoustic with lithological data and boundaries from core EN18218 allow the subdivision of



basin infill into two acoustic units (AU1 & AU2) corresponding with sediments deposited within lithological units I (AU1) and II & III (AU2) (Table 2). The boundary between AU1 and AU2 represents a major lithological boundary. The base of AU2 likely constitutes the base of sediment infill and is marked by progressively weakening reflectors

with depth. Both boundaries trace closely the modern-day lake bathymetry (Fig. S2). Acoustic reflectors within AU1 demonstrate generally well-stratified, high-amplitude, continuous, horizontal and sub-parallel reflectors bounded at the top by the sediment-water interface. The thickness of AU1 shows a basin-wide average of 2.44 m and is thickest within the southern sub-basin proximal to the fluvial inflow and alluvial fan (6.5–7 m) and thins towards the northern shelf (Fig. 3b). A volume of ca. 14935205 $m^3$ (0.015 $km^3$) was estimated for AU1.

Reflectors within the uppermost portion of AU2 show generally lower amplitude, sub-horizontal and sub-parallel reflectors and show some continuity with AU1 reflectors (Fig. 2a, b). The lowermost part of AU2 consists of highly discontinuous reflectors with little structure and rare sub-horizontal reflectors and includes sediments deposited within the lowermost LU-III including sandy-gravelly material. The upwards transition to sub-parallel reflectors likely represents sediments deposited within the upper LU-III and throughout LU-II. AU2 is more variable in

thickness (average ca. 2.96 m) than AU1 but is generally thickest within the southern sub-basin and to a lesser-extent within the northern sub-basin and at localities close to the northern shelf/northern sub-basin boundary observed within hydroacoustic profiles (Fig. 3c). AU2 possesses a volume of ca. 18055352 $m^3$ (0.018 $km^3$) with complex internal architecture with hummock like structures and ridges.


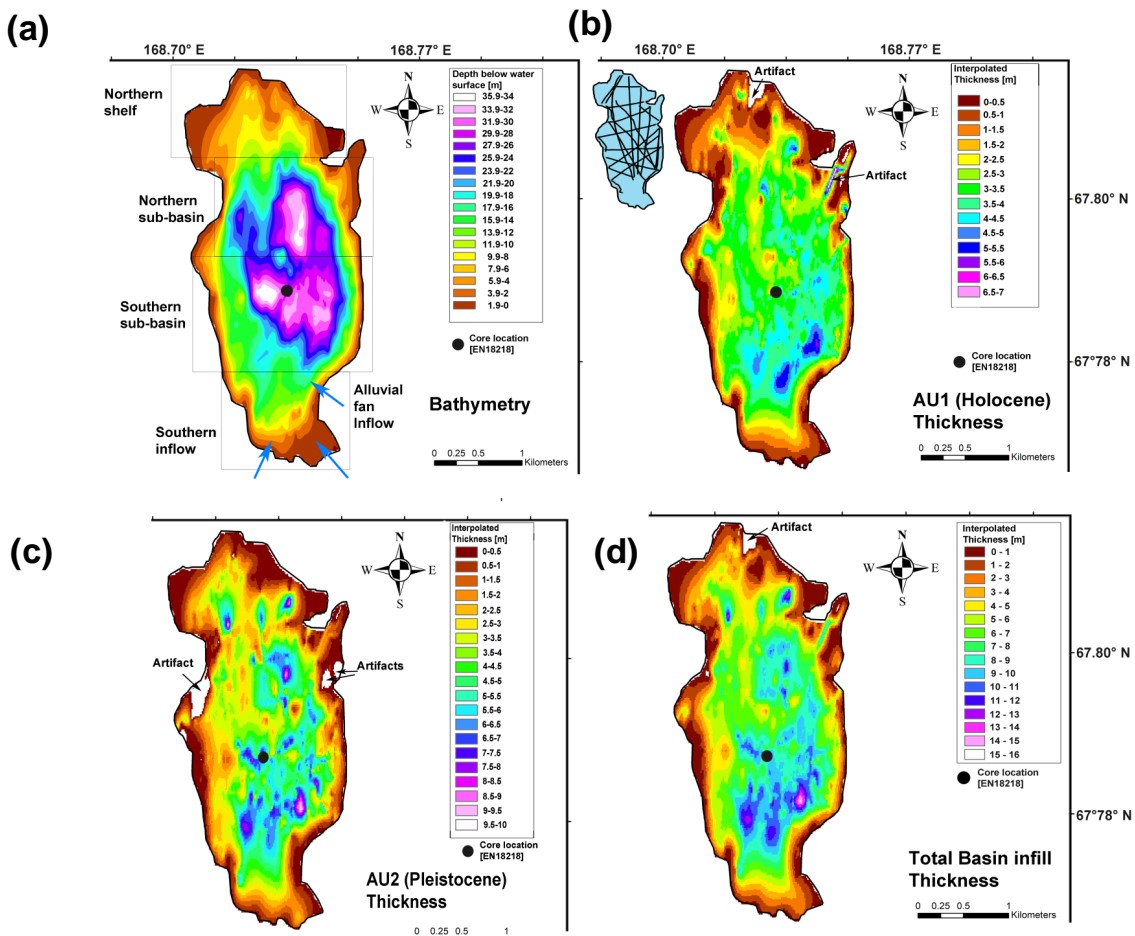

**Figure 3. (a)** Interpolated high-resolution lake bathymetry with morphological regions denoted by black boxes. **(b)** Interpolated AU1 thicknesses. Blue and pink colors denote the thickest sediment within the southern sub-basin. **(c)** Interpolated sediment thicknesses within AU2. Similarly to AU1, the greatest thicknesses are predominantly located in the southern sub-basin but with other locations in the northern-sub basin showing thick sediments too. **(d)** Whole sediment package interpolated thickness. Thin sediment is found proximal to the basin margins and at the north-western shelf. The sediment distribution generally reflects sediment focussing into the deeper lake at greater water depths. Interpolation artefacts are attributed to data sparsity in some regions with larger artefacts being removed prior to data analysis.

### 4.2 Core chronology and sedimentation rates

The age-depth model (Fig. 4a) shows maximum modelled sediment ages of ca. 29 cal. ka BP and thus the core provides an entire record of the Holocene and MIS2 (Elias and Brigham-Grette, 2013). The occurrence of a surface sample that shows an age older than zero, and two samples (Lab-ID: AWI - 3001.1.1; AWI - 3002.1.1) older than the surrounding samples may reflect sediment mixing or redeposition of organic carbon from the catchment through permafrost processes (Abbott & Stafford, 1996; Björck & Wohlfarth 2002). A similar process

may explain the scatter of ages within LU-III samples (Fig 4a). Median sedimentation rates (SR) are plotted alongside 1 and 2σ uncertainty ranges and vary from a maximum of 0.54 mm a$^{-1}$ to a minimum of 0.12 mm a$^{-1}$ (Fig. 4b). Sedimentation rates are highest within LU-I (average 0.36 mm a$^{-1}$, average 1σ uncertainty 0.25–0.45 mm a$^{-1}$). Low-intermediate but progressively increasing rates are seen within LU-II (average 0.20 mm a$^{-1}$,


average 1σ uncertainty 0.16–0.23 mm a⁻¹) with pronounced peaks at 510.5 and 371 cm. LU-III demonstrates low

rates but with larger uncertainty (average 0.17 mm a⁻¹, average 1σ uncertainty 0.15–0.27 mm a⁻¹) due to age

model scatter. Changes in sedimentation rates generally correspond with major lithological variations.

Sedimentation rate reductions occur at the LU-III/LU-II (ca. 560 cm) boundary and increases are observed across

the LU-II/LU-I (ca. 341 cm) boundary (Fig 4a, b).

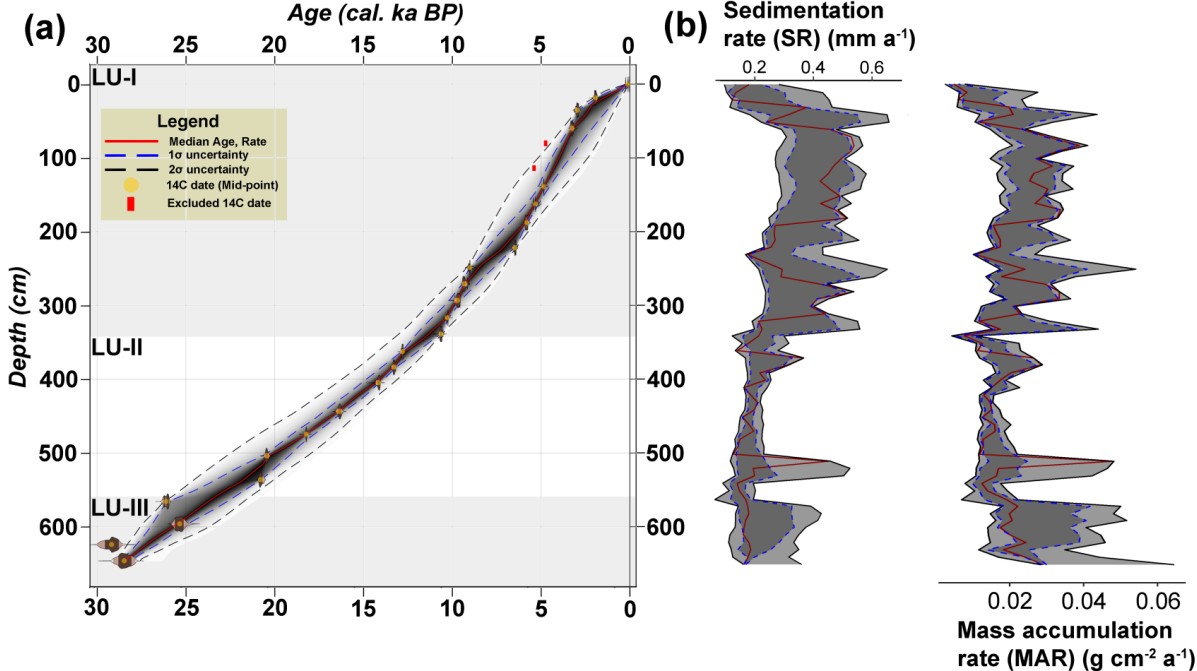

**Figure 4. (a)** High-resolution age-depth model for sediment core EN18218 created from 23 radiocarbon dates with sigma uncertainty range. **(b)** Sedimentation and mass accumulation rates (dark red lines) and 1 and 2σ uncertainty ranges (light and dark grey ribbons).





### 4.3 Core sedimentological and biogeochemical characteristics

LU-III (Table 2) represents the lowermost unit of core EN18218 comprising sand-gravel material at the core base (below 653 cm) succeeded by light-coloured (low b*), layered silty-clay sediments to ca. 560 cm. Sediments

within LU-III show low water contents (average 39.5 %), high DBD (average 1.22 g cm$^{-3}$), very high magnetic susceptibility values (average 681 [10$^{-6}$ SI]) and low mean grain-sizes (average 3.9 μm) traced by Zr/K and K/Ti ratios (Fig. 5). The grain-size distribution shows a predominance of silt-clay with minor sand concentrated above the core base. MARs vary between 0.029 g cm$^{-2}$ a$^{-1}$ at the core base and 0.019 g cm$^{-2}$ a$^{-1}$ at the top of LU-III (average 0.022 g cm$^{-2}$ a$^{-1}$, average 1σ uncertainty 0.019–0.033 g cm$^{-2}$ a$^{-1}$) but are associated with high

uncertainty in the positive sigma direction (Fig. 4). TOC contents (wt%) (average 0.27 %), TOC per volume (average 0.0033 g OC cm$^{-3}$), Br/Al ratio values demonstrate their lowest values at any depth within the core alongside TC (average 0.28 %) and TS (average 0.018 %) (Fig. 6). OCARs are extremely low and stable with low uncertainty (average 0.58 g OC m$^{-2}$ a$^{-1}$, average 1σ uncertainty 0.50–0.92 g OC m$^{-2}$ a$^{-1}$). Equally Mn/Fe shows stable, low values. The LU-III/LU-II boundary is characterized by a decrease in magnetic susceptibility and

increase in the K/Ti ratio corresponding to fine grain-sizes at around 560 cm with a grain-size minimum (2.3 μm) at 550 cm. This corroborates a change in sediment lithology to non-layered, very fine–fine silty sediments in LU-II. Sediment grain-size increases around 506 cm and again around 483 cm clearly viewed by an increase in the Zr/K ratio and a peak in mean grain-size marking an increasing contribution of sand to the particle size distribution. Following a unit maximum in grain-size at 431 cm (5.1 μm) a decrease is observed with a value at

391 cm similar to the unit average of 3.82 μm. Grain-size then rises towards the LU-II/LU-I transition at 341 cm. Water content is significantly higher (average 51.8 %) and dry bulk density lower (average 0.87 g cm$^{-3}$) than in LU-III. MARs are generally lower than in LU-III, associated with lower uncertainty (average 0.017 g cm$^{-2}$ a$^{-1}$,

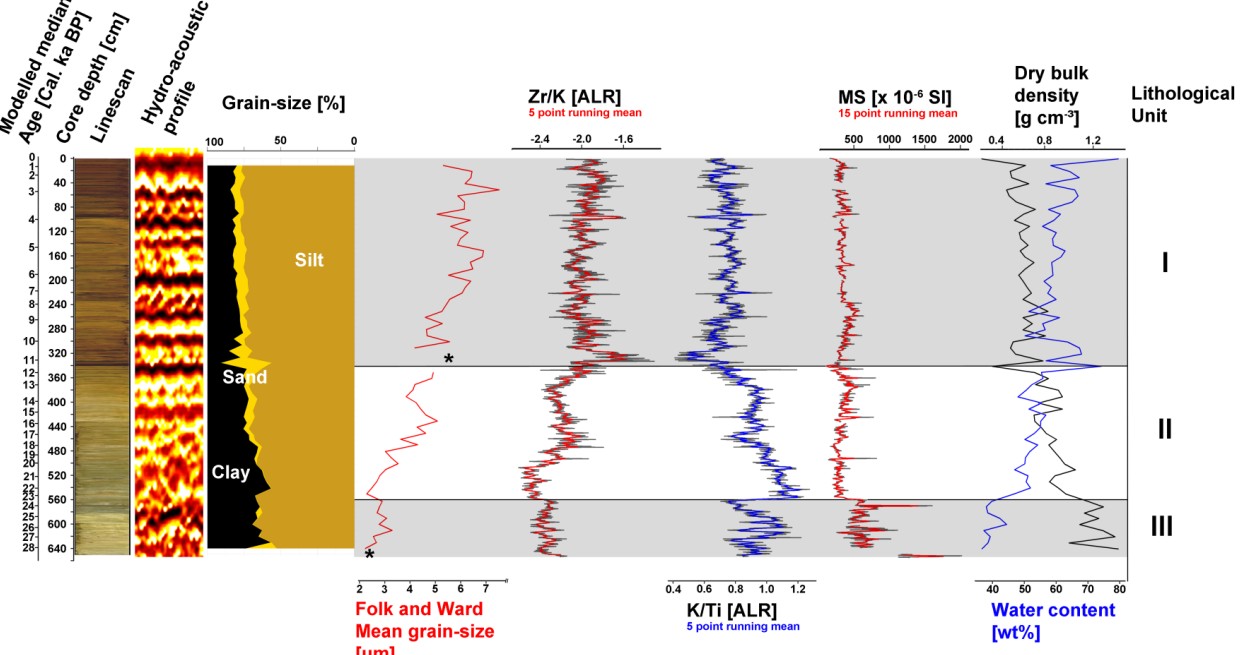

**Figure 5.** Sedimentological and sediment-geochemical proxies obtained from core EN18218 including measured grain-sizes, XRF grain-size proxies, magnetic susceptibility water content and dry bulk density alongside the hydroacoustic profile from the drilling location and lithological unit derivation (Grey-boxes). * in mean grain-size plot refers to three excluded data points (650.5, 341, 321 cm) of very high grain-size (up to 25 μm).

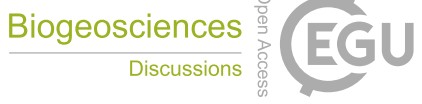

average 1σ uncertainty 0.013–0.019 g cm$^{-2}$ a$^{-1}$) and follow variations in sedimentation rate with maxima at ca.

510.5 cm and 380–371 cm. TOC & TC values are significantly higher in LU-II (average TOC 0.77 %, average TC 0.78 %) and increase concomitantly to a maximum (> 2 % C) at the LU-II/LU-I transition in line with darkening sediment colour and increasing Br/Al. TOC per volume tracks TOC wt% with a value range of 0.005–0.008 g OC cm$^{-3}$ (average 0.0065 g OC cm$^{-3}$). The average OCAR in LU-II is more than double LU-III (average 1.34 g OC m$^{-2}$ a$^{-1}$, average 1σ uncertainty 1.03–1.48 g OC m$^{-2}$ a$^{-1}$) and rises from a minimum at 550 cm (0.72 g OC m$^{-2}$ a$^{-1}$) to a unit maximum at 371 cm (2.67 g OC m$^{-2}$ a$^{-1}$). TS shows higher values within LU-II (average TS 0.04 %) and

increases in the upper part towards the LU-II/LU-I boundary. Nonetheless TS contributes very little to the total core biogeochemistry. The Mn/Fe ratio is more variable within LU-II and gradually increases from the LU-III/LU-II boundary upwards forming a plateau 472–363 cm before decreasing to the LU-II/LU-I transition.

LU-I is typified by dark-coloured, fine-silt sediments of coarser grain-size than observed for all other units (average 6.5 µm) mirrored in Zr/K and K/Ti ratios. MARs achieve high values particularly between 311 and 251

cm and between 181.5 and 61.5 cm consistent with sedimentation rates (average 0.22 g cm$^{-2}$ a$^{-1}$, 1σ uncertainty 0.015–0.028 g cm$^{-2}$ a$^{-1}$). TOC & TC contents attain their highest values (average TOC 1.62 %, average TC 1.63 %) with a prominent maximum at the LU-II/LU-I boundary and unit minimum (1.2 %) between 261 and 251 cm before increasing to values > 2 % at the modern sediment surface. Br/Al follows TOC and TC and increases slightly throughout the unit towards the sediment surface. TOC per volume values are higher than LU-II and LU-

III (average 0.0097 g OC cm$^{-3}$). TS demonstrates a broad peak between 371 and 271 cm commensurate with extremely low Mn/Fe ratio values. TS values then decrease and remain low with the exception of a maximum at the sediment-water interface. OCARs are significantly higher than in LU-II and LU-III, associated with higher uncertainty and vary between 1.56 to 6.3 g OC m$^{-2}$ a$^{-1}$ (average 3.53 g OC m$^{-2}$ a$^{-1}$, average 1σ uncertainty 2.50–4.50 g OC m$^{-2}$ a$^{-1}$) with values increasing across the LU-II/LU-I transition to high values between 311 and 271 cm

generally consistent with higher TS and the Mn/Fe minimum. High OCARs are also observed between around

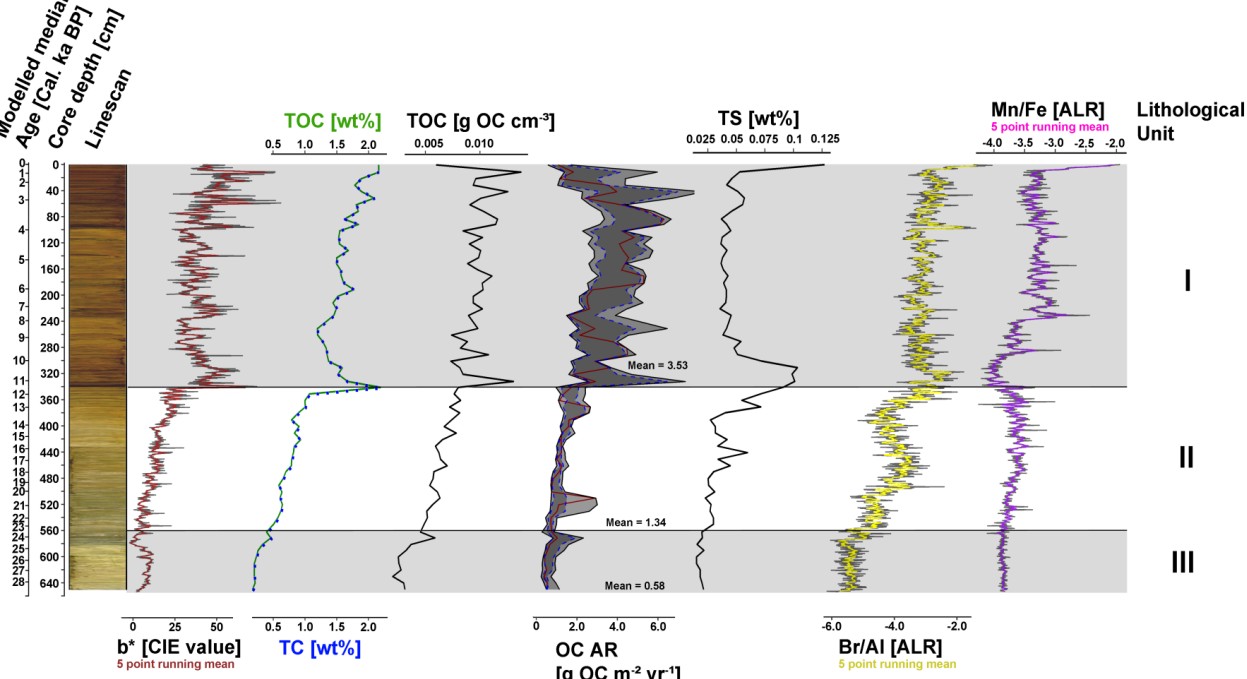

**Figure 6.** Biogeochemical proxies measured from samples obtained from core EN18218 (TOC, TC, TS) as well as from XRF (Br/Al, Mn/Fe) and line-scanning (b*). Average OCAR (Red line) is plotted alongside sigma uncertainty ranges (grey ribbons) synonymous with descriptions in Figure 4.



181.5 and 41.5 cm alongside high SRs and MARs in line with increasing TOC contents (maximum 6.3 g OC m$^{-2}$ a$^{-1}$).

### 4.4 Lake carbon and sediment pools

Average sediment thicknesses and volumes are similar between both sediment packages but slightly greater within the LU-II/LU-III (AU2) package (Table 2). The total carbon pool (TOC$_{pool}$) calculated amounts to 0.26 ± 0.02 Mt with a respective average carbon density (TOC-density) of 7.85 ± 0.60 kg m$^{-3}$. LU-II & LU-III sediments at Lake Rauchuagytgyn displayed the lowest estimates with around 0.1 ± 0.007 Mt of carbon stored and a TOC-density of 5.65 ± 0.40 kg m$^{-3}$ when compared with the more carbon rich LU-I sediments (TOC$_{pool}$ = 0.15 ± 0.005 Mt, TOC-density = 9.87 ± 0.34 kg m$^{-3}$ ). The total calculated sediment pool for total sediment mass deposited was estimated via bootstrapping as 25.7 ± 1.71 Mt with an average sediment density of 780 ± 52 kg m$^{-3}$. Individual calculation of sediment pools in LU-I and LU-II/LU-III yielded values of 9.33 ± 0.32 Mt and 17.1 ± 0.89 Mt respectively.

### 4.5 Results of principal component analysis

The PCA biplot (Fig. 7) of sedimentological, biogeochemical and accumulation rate data shows that PC1 (60.2 %) and PC2 (12.4 %) together explain 72.6 % of the total variance within the dataset. A clear division and clustering of data is observed that confirms the lithological unit definition and the main data trends described in sect. 4.2 & 4.3. The variance in the positive PC1 direction is controlled chiefly by productivity variables (TOC, TC, Br/Al, Br, TS, b*), grain-size (Sand, Silt, Mean GS, Zr/K) and to a lesser degree sedimentation (SR) and accumulation rates of carbon (OCAR). Negative values of PC1 are explained by reduced organic content, increasing fine-grained contribution of clay, increased DBD, light colouration (l*) and MS, Ti, Ca, K and K/Ti. The mass accumulation rate controls chiefly variance within the PC2 direction (Fig. 7, Fig. S3).

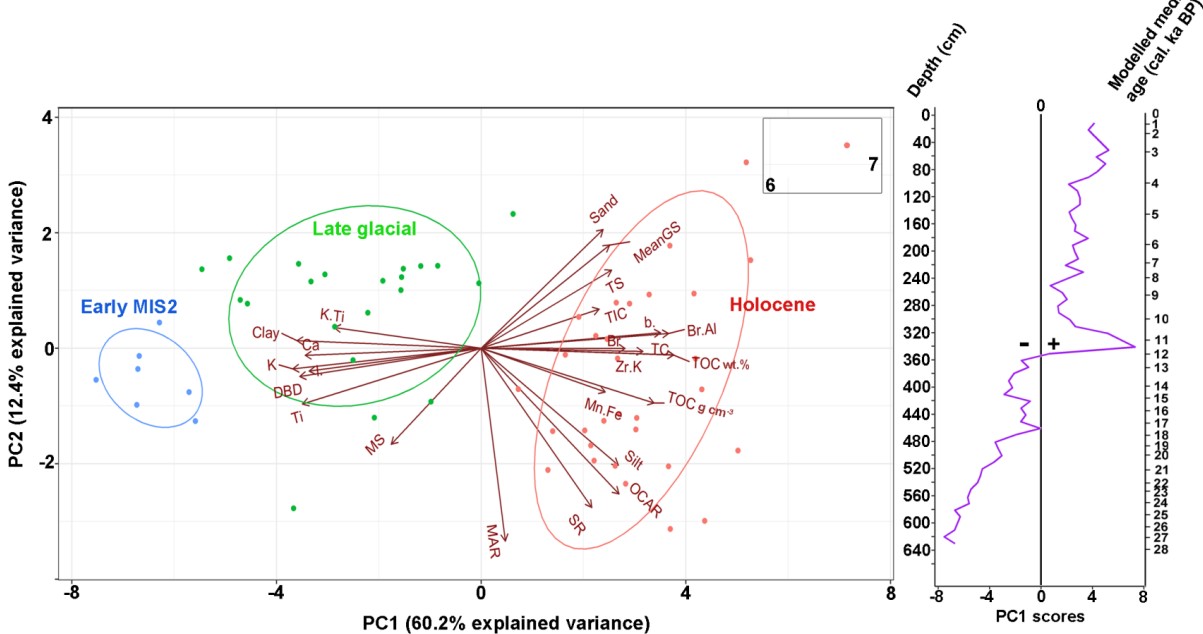

**Figure 7.** Principal component analysis biplot of sedimentological, biogeochemical and accumulation rate data from core EN18218. Clustering according to lithological unit definitions.





### 5. Discussion:

#### 5.1 Accumulation history of Lake Rauchuagytgyn in a palaeoenvironmental context

##### 5.1.1 Glacially influenced sediment and carbon accumulation during the early Marine Isotope Stage 2 (ca.
**29–23.4 cal. ka BP)**

The sediment base at Lake Rauchuagytgyn is suggested from the occurrence of sand and pebble-grade material, alongside discontinuous hydroacoustic reflectors to be glacial till (Mangerund & Svendsen, 1990). Although radiocarbon ages show some scatter within LU-III, a modelled basal age of ca. 29 cal. ka BP agrees with the timing of cold conditions during MIS2 recorded by other regional records and Alaskan lake records (Asikainen et

al., 2007; Finkenbinder et al., 2014; Vyse et al., 2020). Basal ages of several large northern latitude lakes show similar ages to basement sediments at Lake Rauchuagytgyn (Anderson and Lozhkin, 2015; Brosius et al., 2021; Finkenbinder et al., 2014). Basement structures observed in hydroacoustic data likely represent remnants of glacial activity within the basin including moraines and sub- or post-glacial melt channels that subsequently influenced basin sediment distributions (Lebas et al., 2019; Lebas et al., 2021). Overlying layered sediments

characterized by high magnetic susceptibility and dry bulk density corroborates a lake-proximal glacier and deposition of glacigenic sediment (Bakke et al., 2005; Gromig et al., 2019, Van der Bilt et al., 2015). This is further supported by abundant minerogenic elements Ca, Ti and K and a light sediment colouration that may support glacially derived material possibly in the form of rock flour (Fig. 7) (Vyse et al., 2020). The fingerprint of glacigenic sediment is recorded clearly by strongly negative PC1 scores from principal component analysis

clustered as "Early MIS2" (Van der Bilt et al., 2015). Catchment conditions during deposition were likely cold with minimal within-lake and catchment vegetation productivity indicated across all biogeochemical proxies. Extremely low OCARs during the early MIS2 of ca. 0.58 g OC m$^{-2}$ a$^{-1}$ suggest minimal organic carbon input to the basin sediment that was predominantly autochthonous in origin with limited contribution from catchment vegetation (Kokorowski et al., 2008). Lacustrine palynological records and vegetation reconstructions of the catchment

surrounding Lake Ilirney (ca. 50 km south) during MIS2 agrees well with cold conditions with low pollen productivity of catchment flora represented by low pollen concentrations (Andreev et al., accepted). In addition, pollen reconstructed July temperatures were 4–5 °C lower than modern in agreement with unfavourable climate conditions for organic carbon productivity (Andreev et al. accepted). Sedimentological records from Lake Ilirney during MIS2 also agree with extremely low within-lake productivity represented by very low TC and absent

diatoms (Vyse et al., 2020). Preliminary diatom analyses also show a complete absence of diatoms within LU-III in line with unfavourable conditions (Biskaborn, unpublished data). Organic material arriving at the sediment-water interface was likely preserved by low oxygen conditions. This is recorded by low and stable values of the XRF-derived Mn/Fe ratio that well represents the redox conditions at the sediment-water interface (Baumer et al., 2020; Biskaborn et al., 2019; Fritz et al., 2018; Heinecke et al., 2017; Naeher et al., 2013).

Small grain-sizes directly measured by laser diffraction are supported by indirect, XRF-derived grain-size proxies of coarse (Zr/K) and clay-dominated sediment (K/Ti). Fine-grained composition likely reflects deposition beneath a quasi-permanent lake ice layer that further reduced within-lake productivity through limiting light penetration with insufficient irradiance required by algal communities and hence diminishing organic matter export to the hypolimnion (Andersen et al. 1993; Bouchard et al., 2011; Croudace & Rothwell., 2015; Cuven et al., 2010;

Mclaren & Bowles. 1985). Thus, during deposition of LU-III the lake may have been of cold-monomict type (Cremer and Wagner, 2003). Sediment input at Lake Rauchuagytgyn was generally low during LU-III deposition



as represented by comparatively low SRs and MARs. The uncertainty in the positive sigma direction may suggest that rates could be higher when integrating uncertainty associated with age inversions within LU-III. Sedimentation rates reported for Lake El´gygytgyn crater lake (Fig. 1) were low during MIS2 (4.8 cm/ka),

compared to higher rates (7.6 cm/ka) during the Holocene (Nowaczyk et al., 2007). A similar finding was made for Harding lake in Alaska (Finkenbinder et al., 2014). In both of these settings, low sedimentation rates were interpreted as a consequence of thick, quasi-permanent lake surface ice cover and an extremely thin or absent catchment tundra active layer during cold episodes that reduced catchment sediment delivery and wind-driven sediment redistribution (Asikainen et al., 2007; Francke et al., 2013; Melles et al., 2007). A similar finding at lake

Karakul, Tajikistan of low sedimentation rates during MIS2 (0.15 mm a$^{-1}$) was also explained by reduced sediment input during MIS2 compared to the Holocene alongside reduced organic matter accumulation (Heinecke et al., 2017). The low sedimentation and mass accumulation rates at Lake Rauchuagtygn during the early MIS2 and the late glacial may be supported by the generally equitable thicknesses and sediment volumes between the AU2 (LU-II/LU-III) and AU1 (LU-I) sediment packages derived hydroacoustically (Table 2).

Differences are however observed when MIS2 accumulation rates are compared with other Siberian glacial lake sites including Lake Bolshoye Shchuchye in the Ural Mountains (average last glacial maximum (LGM) 1.68 mm a$^{-1}$). Lake Bolshoye Shchuchye demonstrates a voluminous LGM sediment package (0.325 km$^3$) 6 times the volume of the Holocene sediment package (0.05 km$^3$) which shows markedly lower sedimentation rates (0.36 mm a$^{-1}$) and is interpreted to result from intense glacial denudation of the catchment during the LGM (Haflidason

et al., 2019).

The difference in glacial sediment accumulation between sites may relate to a combination of factors including the duration of lake surface ice cover, the thickness of the catchment active layer, the existence/absence of deep-water conditions and also differences in glacier dynamics between catchments (Asikainen et al., 2007; Francke et al., 2013). Remote sensing based studies of Chukotkan glacial geomorphology and structures within

the Rauchua valley have suggested that the catchment glacier was likely a passive glacier, ca. 25 km in length that extended along the length of the Rauchua river valley and discharged into the Rauchuagytgyn basin (Glushkova, 2011). Thus, the catchment glacier may have been predominantly non-erosive during the early MIS2 and hence not contributed significant sediment to the lake basin supporting the low rates discussed here (Gurnell et al., 1996). It must be however noted that the radiocarbon age scatter in LU-III contributes uncertainty to the

SRs and MARs derived for this interval as marked by the wider uncertainty band within the presented age model (Fig. 4b).

Sediment accumulation preserved within LU-III can be attributed regionally to the MIS2 "Sartan" glaciation recorded within Chukotka (Anderson & Lozhkin 2015; Brigham-Grette et al., 2003; Melles et al., 2012; Vyse et al., 2020) and the Itkillik II glaciation in Alaska, eastern Beringia (Hamilton and Ashley, 1993). Glaciation during this

time was associated with asynchronous glacial advances in Alaskan mountain ranges between 31 and 28 cal. ka BP and advances in western Beringian glaciers to maximum extents between 27 and 20 cal. ka BP (Brigham-Grette et al., 2003; Elias and Brigham-Grette, 2013 and references therein). The presence of a valley glacier at Lake Rauchuagytgyn is in contrast to more westerly, eastern Russia sites from the Verkhoyansk mountains where no glacier advance was witnessed during MIS2 (Diekmann et al., 2017; Stauch and Lehmkuhl, 2010). This

supports previous suggestions that despite cold conditions, higher moisture derived from the Pacific Ocean permitted some glacier growth within Chukotka (Stauch and Gualtieri, 2008).

**5.1.2 Mid-to-late MIS2 accumulation during progressive climate amelioration (ca. 23.4–11.5 cal. ka BP)**



Sediment accumulation recorded by LU-II sediments represents a transitional mode between the early MIS2 and Holocene clearly viewed within PCA results between ca. 23.4 and 11.5 cal. ka BP and likely reflects a time-
progressive shift towards the early stages of paraglacial deposition (Fig. 7). Hydroacoustic data show an upward transition to stable, deep water, lacustrine deposition represented by the increasing prevalence of stronger amplitudes from well-stratified reflectors (Lebas et al., 2021; Haflidason et al., 2019). A clay maximum and grain size minimum at ca. 550 cm (ca. 22.7 cal. ka BP) may suggest initial increases in lake water-depth through glacial melt additions of rock-flour rich meltwater that could have led to the observed reduced SRs and MARs at
this time. This is may be supported by the high values of K/Ti (clay contribution) and low values of Zr/K (proxy for coarser grain-sizes) (Kříbek et al., 2017; Cuven et al., 2010). It is also likely that the persistence of the quasi-permanent ice layer continued to contribute to predominantly fine-grained, low energy deposition (Asikainen et al., 2007). The strong increase in Zr/K accompanied by increased sand and silt contribution firstly at ca. 20.5 cal. ka BP consistent with a dramatic peak in SR and MAR values and again ca. 18.5 cal. ka BP reflects increasingly
rapid glacial retreat (Gromig et al., 2019). The timing of these system responses is generally consistent with depositional changes recorded at Lake Ilirney in response to deglaciation ca. 20 ka (Vyse et al., 2020) and those proposed for Alaskan glacial retreat (20–19 ka BP)(Elias and Brigham-Grette, 2013 and references therein). Progressive deglaciation facilitated the gradual opening of the lake catchment area and development of fluvial systems that plausibly enhanced sediment load and input to the lake basin through the initiation of paraglacial
processes (Ballantyne, 2002; Dedkov, 2004). Sediment dynamics were likely controlled by rivers at the lakes southern margin as indicated by the greatest thicknesses of sediment within the southern sub-basin deposited within AU2 (Fig. 3c). Thick sediments within AU2 in the northern sub-basin may indicate additionally enhanced sedimentation derived from the north. The occurrence of a rhomboidal-shaped deposit similar to preserved lake sediments within other lacustrine and continental shelf environments at the northern shelf-northern sub-basin
transition may represent a mass transport deposit during some part of AU2 deposition (Fig. 2b, Fig. S4) (Baster et al., 2003). Catchment sediment availability likely also began to increase during this time due to gradual sub-aerial exposure of subglacial sediments that subsequently became entrained within paraglacial river systems or transported to the lake via aeolian processes contributing to progressively increasing sedimentation rates (Tripathi & Rajamani, 1999; Wang et al., 2015).

Organic carbon accumulation rates at the base of LU-II were slightly higher than LU-III signifying increasing organic carbon production and accumulation across the LU-III/LU-II transition. Subsequently increasing rates alongside up-core sediment darkening and increases in all organic proxies across LU-II points towards continuous growth in carbon accumulation related to ameliorating conditions during the mid-to-late MIS2 onwards. Organic productivity was most likely dominated by autochthonous productivity from within-lake algae
with only minor contributions from catchment vegetation. Though nitrogen was not measured as part of this study, extremely low TOC/TN ratios measured for sediments at the nearby glacial Lake Ilirney during the late glacial implied that most organic matter was derived from protein rich algal sources and not from cellulose-rich and protein-poor vascular land plants from the lake catchment and from macrophytes within the lake (Baumer et al., 2020; Meyers & Teranes., 2005; Vyse et al., 2020). This is supported by low pollen concentrations recorded
within the Lake Ilirney pollen record and the persistence of Poaceae and *Artemisia* during the late glacial that signify persistently cold conditions and low catchment pollen productivity (Andreev et al., accepted). Based on the similarities between the lakes and their catchments, their proximity, and a lack of macrophytes at both sites, a similar interpretation can be postulated for Lake Rauchuagytgyn (Sifeddine et al., 2011). Higher and more variable Mn/Fe ratio values throughout much of LU-II point towards increased oxygen at the sediment-water



interface that could have enhanced degradation of organic matter (Baumer et al., 2020; Fritz et al., 2018). This
could be indicative of a gradually increasing summer open water season with more intensive mixing of the water
column through wind and also through enhanced fluvial inflow during progressive late glacial climate amelioration
(Baumer et al., 2020).

A reduction in accumulation rates ca. 12.6 to ca. 11.5 cal. ka BP, broadly associated with a small grain-size fining
may represent some evidence of reduced accumulation due to cooling associated with the younger Dryas (YD).
The magnitude of this is however not substantial when all proxies are considered together within principal
component analysis (Fig. 7). Moreover, missing evidence of younger moraines from the catchment implies that
there was no glacial re-advance (Glushkova, 2011). These findings are consistent with recent regional and trans-
regional records that suggest a spatially variable, and possibly more limited younger dryas event in east & far
east Russia and parts of eastern Beringia (Anderson & Lozhkin, 2015; Kokorowski et al., 2008; Lozhkin &
Anderson, 2013; Lozhkin et al., 2018).

### 5.1.3 Holocene controls on basin sediment and carbon dynamics (ca. 11.5 cal. ka BP–present)

The Holocene start is marked across all proxies by increasing organic proxy values and sediment grain-size
across the LU-II/LU-I boundary alongside increasing accumulation rate values (Figs. 5,6,7). High TOC and TS
and a dark sediment colour form a prominent peak during the early Holocene between ca. 11.5 and 9.7 cal. ka
BP synchronous with lowest values of Mn/Fe that could reflect increased preservation of organic material under
low-oxygen conditions at the sediment-water interface alongside enhanced organic productivity (Fig. 6) (Fritz et
al., 2018). Increasing carbon accumulation is supported by rising organic carbon accumulation rates that
demonstrate high early Holocene values between ca. 10.2 and 9.2 cal. ka BP. Low K/Ti values during the early
Holocene that subsequently persist throughout the mid-late Holocene may reflect increased chemical relative to
physical weathering due to the leaching of K from clastic material entering the lake by enhanced catchment
pedogenic processes that influenced OCAR values (Baumer et al., 2020; D'Arcy & Carignan, 1997). The timing
of this phase agrees well with the timing of warm and humid environments interpreted during the early Holocene
recorded at Lake Emanda, central Yakutia (11.5–9.0 cal. ka BP) and coincides with the maximum in local solar
insolation (Baumer et al., 2020; Berger & Loutre 1991). Moreover, this finding is consistent with palynological
records from Lake Ilirney and from a Holocene short core from Lake Rauchuagytgyn which show evidence for a
Holocene thermal maximum ca. 10.6–7 cal. ka BP (Andreev et al., accepted).

Increasing early Holocene sediment and mass accumulation rates alongside greater sand contribution to the
grain-size distribution may relate to the input of coarser grained fluvial detrital input from a paraglacial,
deglaciated catchment and could result from a water budget change associated with warmer and more humid
conditions (Ballantyne, 2002). Annual precipitation ($P_{ann}$) reconstructed from palynological records at Lake Ilirney
suggested an increase from ca. 225 mm during the late glacial to ca. 300 mm in the early Holocene that could
have enhanced catchment surface runoff and hence fluvially driven reworking of unstable-metastable glacially
deposited sediment by paraglacial processes (Andreev et al., accepted; Ballantyne, 2002). Heinecke et al.,
(2017) proposed a similar scenario to explain increased Holocene sedimentation rates and increased grain-sizes
during the end of the late glacial and early Holocene (13.3–6.6 cal. ka BP) at glacial Lake Karakul. Moisture in
north-eastern Siberia during interglacials, is derived mostly from warm North Atlantic surface waters and brought
via the westerlies to the region (Melles et al., 2007). The waning of the Scandinavian and Barents-Kara ice-
sheets likely allowed the passage of moister Westerlies to the region that were blocked during the last glacial





maximum (Elias & Brigham-Grette 2013; Meyer et al., 2002). Increased sedimentation rates also additionally
        enhanced carbon burial and preservation (Einsele et al., 2001). Enhanced precipitation and fluvial input to Lake
        Rauchuagytgyn during the early Holocene could have led to lake level increase and deep lacustrine conditions at
        the core location that restricted mixing of the water column, leading to stratification and may provide a possible
        explanation for low oxygen conditions (Baumer et al., 2020). Deep lacustrine conditions alongside enhanced
early Holocene within-lake and catchment productivity thus likely led to enhanced carbon deposition and
        preservation.

        A decrease in TOC content (wt%) to a minimum at ca. 8.6 cal. ka BP along with decreasing organic carbon and
        sediment and mass accumulation rates from 8.9 cal. ka BP to a minimum at ca. 7.7 cal. ka BP alongside lower
        grain-size values may reflect some local environmental change and is generally consistent with a Holocene $T_{July}$
minimum at 7.8 cal. ka BP and $P_{ann}$ minimum at 7.6 cal. ka BP recorded within a Lake Rauchuagytgyn short core
        (Andreev et al., accepted). This may possibly be associated with cooling associated generally with the 8.2 cal. ka
        BP event recorded at other locations within Siberia (Biskaborn et al., 2016). A degree of similarity between the
        TOC trend following the early Holocene at both Lake Rauchuagytgyn and Lake Emanda may also suggest that
        bacterial sulphate reduction contributed to the reduced sedimentary organic carbon content during this interval
(Fig. 6) (Baumer et al., 2020). A subsequent increase in OCARs from ca. 6.1 cal. ka BP consistent with
        progressively increasing TOC values towards the modern lake sediment surface represents increased
        accumulation of organic carbon consistent with increasing mean July temperatures ($T_{July}$) to a maximum ca. 13.3
        °C during the mid-late Holocene by ca. 4.6 cal. ka BP (Andreev et al., accepted). Synchronous increases in SR
        and MAR along with coarser grain-sizes may reflect extended periods of ice-free lake surface conditions during
summer associated with high $T_{July}$ (Fig. 4,5) (Finkenbinder et al., 2014). Extended open water conditions during
        this time and throughout the Holocene likely led to longer phases of summer catchment aeolian input, as well as
        increased wind-driven shoreline erosion and sediment redistribution due to the impact of summer storms
        (Asikainen et al., 2007; Francke et al., 2013; Vologina et al., 2003). This hypothesis is supported by the highest
        Mn/Fe ratio values for the core, supportive of enhanced wind and fluvial driven lake mixing processes and
weaker stratification (Doran, 1993; Baumer et al., 2020; Regnéll et al., 2019). Francke et al., 2013 argued that
        coarser grain-sizes and enhanced sedimentation at Lake El´gygytgyn during Interglacials was controlled by
        summer temperatures affecting lake ice cover and hence aeolian input, wind-driven sediment redistribution as
        well as catchment active layer thicknesses and water availability and was not affected by lake-level variations
        (Francke et al., 2013).

Decreasing accumulation rates from ca. 3.6 cal. ka BP to present could reflect reduced late-Holocene sediment
        input due to neoglacial cooling and is in line with variable $T_{july}$ and low $P_{ann}$ recorded within the Ilirney and
        Rauchuagytgyn short core records and generally lower late Holocene Mn/Fe ratio values (Fig. 6) (Andreev et al.,
        accepted). This could tentatively be interpreted to represent a reduction in the summer open water season.

        Enhanced fluvial input to the lake basin throughout the Holocene from the Rauchua river valley and from the
south-east lake margin are reflected in hydroacoustic and interpolated sediment thickness data (Fig. 3b) from
        AU1 which show greater thicknesses within the southern sub-basin. A similar distribution has been observed at
        glacial lakes Bolshoye Shchuchye & Levinson-Lessing (Haflidason et al., 2019; Lebas et al., 2019). The presence
        of an alluvial fan ca. 600 m in diameter at the south-eastern margin is supportive of intensified Holocene
        paraglacial processes that resulted in an input source of coarser fluvial detrital material into the southern sub-
basin contributing to the observed increased sedimentation and mass accumulation rates (Doran, 1993; Smith





and Jol, 1997). This is further supported by coarse, sand-dominated surface sediments close to the alluvial fan front (site EN18220) (Fig. 1 & Figs. S5, S6). Modern observations at Lake El´gygytgyn have shown that summer fluvial activation associated with snowmelt can lead to significant deposition of coarse material of cobble-grade material to the lake basin (Nolan and Brigham-Grette, 2007). Heightened availability of catchment sediment material due to thicker catchment active layer thicknesses during the Holocene may also provide an additional sediment source leading to increased sedimentation. A thick tundra active layer present during the Holocene at Lake El´gygytgyn was suggested to have contributed to greater detrital input to the basin as well as promoting the weathering of clay minerals (Asikainen et al., 2007; Francke et al., 2013). This further supports the low K/Ti ratios during the Holocene that evidence intensified chemical weathering (Regnéll et al., 2019). Acoustic reflectors preserved within AU1 generally reflect well-developed lacustrine deposition under deep-water conditions prevalent across the basin (Fig. 2)(Lebas et al., 2019). Some variability is observed within hydroacoustic data at the northern shelf where sediments are thin and evidence of erosion of sediments are seen (Fig. S1). This likely represents low deposition due to sediment focussing into the deeper basin and feasibly some Holocene lake-level variability of several meters permitting localised erosion into shallow areas (Moernaut et al., 2010).

### 5.2 Comparisons of carbon and sediment dynamics on a trans-regional scale

### 5.2.1 Organic Carbon Accumulation Rates (OCARs)

The calculated organic carbon accumulation rates for Lake Rauchuagytgyn represent the first calculated values for an Arctic Siberian glacial lake. As such, limited comparable studies exist and are restricted to studies of Siberian thermokarst lake systems that are generally younger and smaller (Anthony et al., 2014). Comparisons must therefore be drawn to Boreal lakes from North America, Greenland and northern Europe. The overall Holocene average organic carbon accumulation rate of 3.53 g OC m$^{-2}$ a$^{-1}$ for Lake Rauchuagytygn is significantly higher than those reported for the Pleistocene units represented by LU-II & LU-III (AU2) of 1.09 g OC m$^{-2}$ a$^{-1}$ (Figs. 8, 9a). This is in accord to increased organic productivity, carbon accumulation and generally higher SR and MAR during the Holocene. The calculated Holocene rate is very similar to that obtained for Holocene deposits of boreal lakes from northern Quebec, Canada (mean 3.8 g OC m$^{-2}$ a$^{-1}$) that were also studied using an equivocal hydroacoustic approach to account for sediment focussing and to rates obtained for Lake Baikal (mean 2.8 g OC m$^{-2}$ a$^{-1}$) (Ferland et al., 2012; Ferland et al., 2014; Martin et al., 1998) (Figs. 8, 9a). The range of Holocene rates is on average lower but generally overlaps with Holocene organic carbon accumulation rates of Greenlandic lakes (mean 6 g OC m$^{-2}$ a$^{-1}$) (Anderson et al., 2009), and to Uinta glacial lakes, USA (mean 5.4 g OC m$^{-2}$ a$^{-1}$) (Munroe & Brencher, 2019). A strong resemblance is also observed when comparing to rates of accumulation calculated for Finnish Boreal lakes that became Ice free at the Holocene start (Fig. 8, 9a, 9b) (Pajunen, 2000; Kortelainen et al. 2004). The average Holocene and whole core Rauchuagytgyn rates plot well within the range of Finnish lakes and close to the mean of Quebec boreal lakes when considering sediment volumes derived from sub-bottom profiling approaches or estimated from core length and lake surface area (Uinta lakes) (Fig. 9a). Recent syntheses of average carbon accumulation rates within European lakes also suggest similar mean accumulation rates ca. 5.6 g OC m$^{-2}$ a$^{-1}$ (Kastowski et al., 2011).





Pronounced differences exist when compared with lake systems reported from Alberta, Canada (mean 15 g OC $m^{-2} a^{-1}$) (Campbell et al., 2000) and global lakes, reservoirs and peatlands (Dean & Gorham, 1998; Mendonça et al., 2017). Furthermore, average Holocene rates calculated for thermokarst lakes from the Cherskii-Kolyma Tundra, far east Siberia are markedly higher (mean 47 g OC $m^{-2} a^{-1}$) than Rauchuagytgyn rates (Anthony et al., 2014)(Fig. 8).

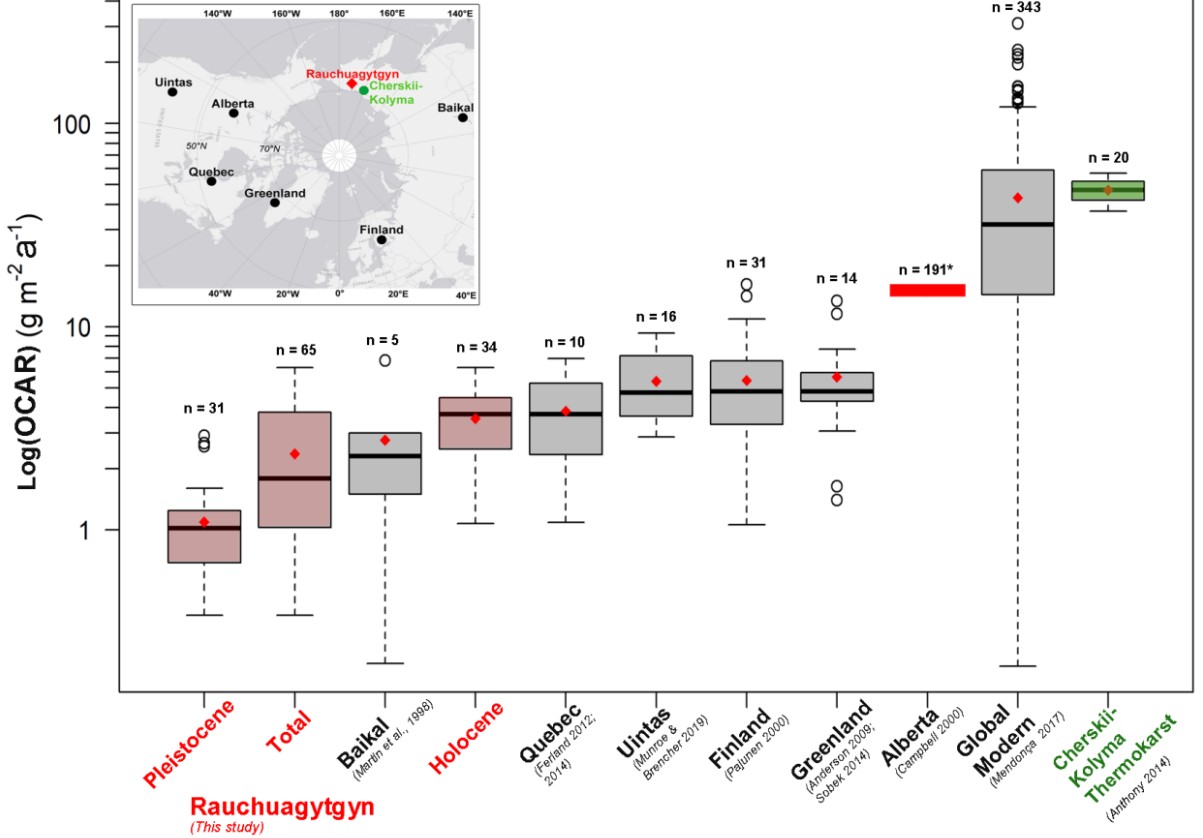

**Figure 8.** Comparison Log-boxplot of average organic carbon accumulation rates (OCARs) for Lake Rauchuagytgyn with carbon accumulation rates from other regions. Rauchuagytgyn OCARs include a total of 65 downcore points and plot close to Boreal lake sites and to Lake Baikal (ESRI 2020). The global modern dataset of Mendonça et al., 2017 included OCARS from 343 globally distributed lake sites (see Text S1). Study references are quoted below the study regions.

### 5.2.2 Sediment and carbon pools within a Siberian Arctic glacial lake basin

The total sediment pool of Rauchuagytgyn was estimated to 25.7 ± 1.71 Mt within a sediment volume of ca. 0.033 $km^{3,}$ with an average dry bulk sediment density of 780 kg $m^{-3}$(0.780 g $cm^{-3}$). Due to the very low organic carbon content of sediments, the majority of the total sediment comprises inorganic detritus. The average calculated sediment density shows high agreement with global syntheses of sediment bulk density measured from lacustrine and fluvial sediments (Avnimelech et al., 2001; Haflidason et al., 2019; Munroe & Brencher, 2019;





Sekellick et al., 2013). Sediment densities are however observed to be significantly higher than the average calculated for Holocene Finnish lakes (0.25 g cm$^{-3}$) mostly due to greater organic carbon and water contents than those remarked for Rauchuagytgyn sediments (Pajunen, 2000).

The estimated TOC$_{pools}$ (total ca. 0.26 Mt) and TOC-densities (total ca. 7.85 kg m$^{-3}$) of Lake Rauchuagytgyn sediments presented here, represent the first assessment of carbon pools within a Siberian Arctic glacial lake.

Our results show that the sediment TOC$_{pool}$ and TOC-density is relatively small, particularly during the deposition of LU-III and LU-II that covers the late Pleistocene and is in agreement with low OCARs discussed in sect. 5.2.1. The small carbon pool is especially evident when compared with carbon budgets and sediment volumes reported for Pleistocene-Holocene age Yedoma and thermokarst and drained thermokarst lake basin records (DTLB) from Alaska, as well as Holocene TOC$_{pools}$ and sediment volumes of Alberta glacial lakes, Canada  (Fig. 9b) (Campbell

et al., 2000; Jongejans et al., 2018). Pleistocene-Holocene sediments obtained from thermokarst lagoon and permafrost sediments from the Bykovsky Peninusla, northern Yakutia also show significantly higher estimated TOC$_{pools}$ (Bykovsky lagoons 5.6 Mt) and TOC-densities (mean 14.24 kg m$^{-3}$) relative to their small spatial extents (Jenrich et al., in review; Schirrmeister et al., 2011). Resemblance is however observed between the TOC$_{pools}$ (0.20 Mt) and TOC-densities (10.45 kg m$^{-3}$) of Polar Fox thermokarst lagoon (Bykovsky) and Rauchuagytgyn

particularly within Holocene sediment of LU-I (TOC$_{pool}$ 0.15 Mt, TOC-density 9.87 kg m$^{-3}$) (Jenrich et al., in review). Moreover, similar values of TOC$_{pools}$ and TOC-density relative to sediment volume have been estimated

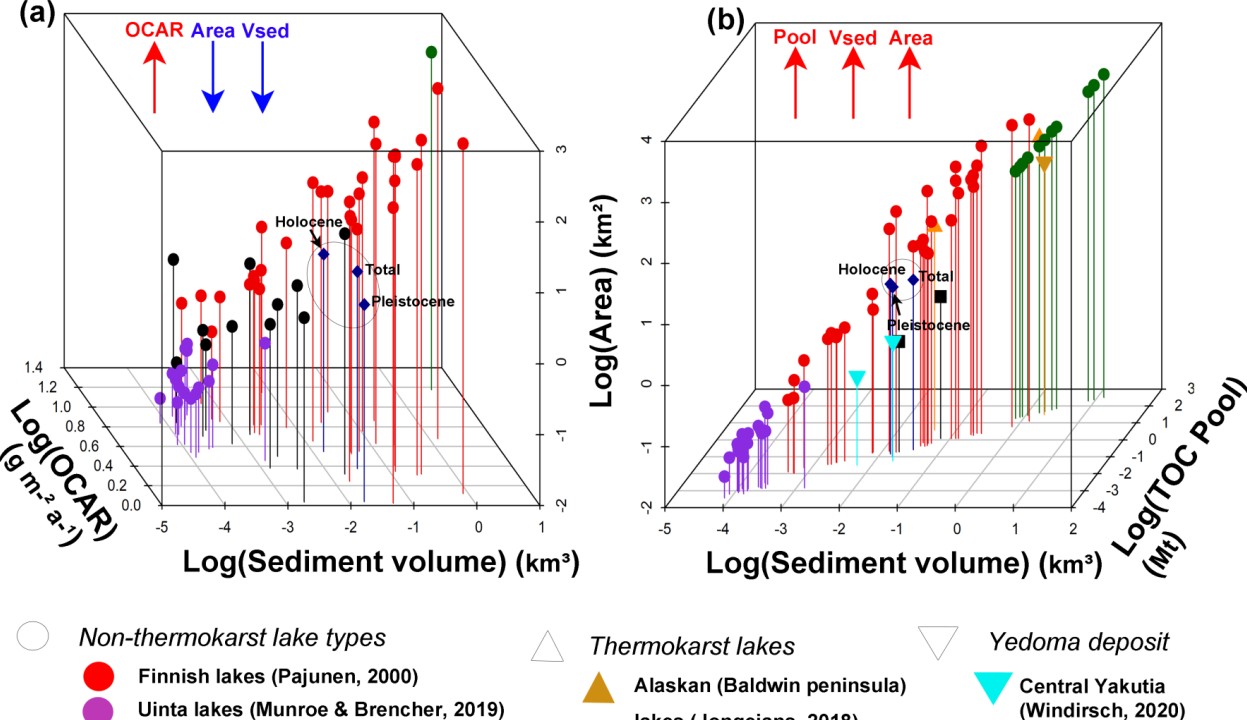

**Figure 9. (a)** 3D log scatter plot of organic carbon accumulation rate (OCAR) vs lake sediment volumes and surface areas. The average carbon accumulation rate for the Holocene at Lake Rauchuagytgyn closely resembles that of Holocene-age Finnish and Quebec lakes and generally lies on a trajectory of increasing OCAR with decreasing sediment volume and area. **(b)** 3D log scatter plot of Carbon pool vs sediment volume and area. The Rauchuagtygyn data-points are highlighted with ellipses in both plots. Rauchuagtygyn displays similar carbon pools to Finnish lakes, Bykovsky thermokarst lagoons and Central Yakutian Yedoma deposits. Rauchuagytgyn Sediments possess however significantly lower carbon pools relative to volume and area than Alaskan (Baldwin Peninsula) thermokarst lake and yedoma deposits and Alberta lakes.



for Pleistocene–Holocene Yedoma and Alas sediments from central Yakutia, eastern Russia (Yedoma: 0.057 Mt, 5.27 Kg m$^{-3}$, Alas: 0.032 Mt, 6.07 Kg m$^{-3}$) that are closer to the range of the Pleistocene values calculated at Rauchuagytgyn (0.1 Mt, 5.65 kg m$^{-3}$) (Windirsch et al., 2020). The highest degree of similarity is achieved with
comparisons of TOC$_{pools}$, surface areas and sediment volumes to Finnish lakes whereby Lake Rauchuagytgyn pools lie on a distinct trend in 3D logarithmic space within the range and around the mean of Finnish lakes (Fig. 9b) (Pajunen et al., 2000; Kortelainen et al. 2004). Estimated carbon pools from Uinta glacial lakes show markedly lower values that shows they store significantly less carbon within their smaller sediment volume (Munroe and Brencher, 2019).

### 5.2.3 Regional and local controls on carbon accumulation and methodological limitations


Lake Rauchuagytgyn stores predominantly inorganic sediment detritus and demonstrates carbon dynamics not strongly dissimilar from other high-latitude lake systems. Differences are however revealed when comparisons of carbon dynamics are made to thermokarst and drained thermokarst lakes (DTLB), large Alberta glacial lakes and global lakes in general. The interplay between many, varied factors likely contribute to major differences in
carbon accumulation dynamics between different deposits and regions that are beyond a major discussion here (Ferland et al., 2012; Kastowski et al., 2011; Sobek et al., 2009). Carbon accumulation at Lake Rauchuagytgyn is however likely affected by several key factors that can be identified.

Its mountainous, high-latitude, tundra location represents cold and dry environmental conditions that provide a first order climatic limit on catchment and lake internal primary productivity and hence carbon production. The
expression of a cold climate is marked by open herb- and graminoid tundra vegetation dominated by Poaceae and *Dryas octopetala* L. with the absence of vegetation at higher altitudes on lake proximal slopes and within the mountainous lake catchment (Shevtsova et al., 2020). The occurrence of a well-developed lake surface ice-layer up to ca. 1.8 m in thickness that is present for ca. 9 months additionally contributes to reduced rates of algal primary productivity by limiting the availability of light as well as the allochthonous input of particulate (POC) and
dissolved organic carbon (DOC) via fluvial and/or Aeolian pathways (Baron et al., 1991; Nolan et al., 2002; Woolway & Merchant, 2019). An additional contributing factor to reduced catchment carbon export to Lake Rauchuagytgyn is likely connected to the steep, mountainous topography and poor soil development within the lake catchment. Catchment soil development is limited primarily to thin cryosols but large areas of barren land exist at higher altitudes (Bünseler, 2019; Shevtsova et al., 2020). D´Arcy and Carignan (1997) suggested that
lakes situated in steep catchments with thin soils demonstrate low DOC exports to lake systems that may lead to reduced carbon loading and hence reduced carbon accumulation. Low modern carbon export and production is supported by the extremely low DOC contents of lake surface water samples at Lake Rauchuagytgyn (Table S1) and at other regional lake sites (Huang et al., 2020) that support low organic matter export from the catchment (Bouchard et al., 2016). These factors were plausibly enhanced under more strongly continental, colder and drier
Pleistocene climatic conditions that led to extremely low OCARs and a small carbon pool as evidenced by the multiple environmental proxies discussed throughout sect. 5.1 (Fig. 9)(Anderson & Lozhkin, 2015; Einsele et al., 2001). Changes in sedimentation rates due to temporal variability in available moisture may also regulate the preservation of organic carbon within lacustrine sediments by affecting the time available for oxygen exposure and hence microbial degradation (Einsele et al., 2001). Higher Holocene sedimentation rates at Lake
Rauchuagytygn resulting from increased precipitation, active layer thickness and aeolian input may thus have further contributed to enhanced Holocene carbon accumulation. Lake area likely also plays a role in modulating differences in accumulation between lake sites and regions (Kastowski et al., 2011). Generally lakes of greater



surface area possess lower rates of carbon accumulation due to reduced carbon loading (Ferland et al., 2012)
(Fig. 9a). Anthropogenic factors can also impact upon organic matter dynamics by changing the land use
dynamics within lake catchments (Biskaborn et al., 2021). However, the current pristine nature of the Lake
Rauchuagytgyn catchment and the low occurrence of human activity, limits the anthropogenic influence when
compared with lake catchments from more populous, lower latitude regions (Kastowski et al., 2011).
Nonetheless, the future impact of exploration and mining operations within the catchment and of climate change
within the permafrost landscape could lead to future changes to sediment and carbon dynamics that are yet to be
seen (Tranvik et al., 2009).

Differences in accumulation calculation and hence rates can also arise due to diverging methodological
approaches that can lead to significant bias. The necessary usage of empirical equations to estimate sediment
volumes (Campbell et al., 2000) owed to missing data or the simplification of sediment volumes to a "box" form
by combining deposit thickness with area (Jongejans et al., 2018; Windirsch et al., 2020) can lead to the
introduction of large uncertainty regarding pool calculations. Volume uncertainty within our study has been
reduced through the usage of hydroacoustic methods. Further uncertainty can arise through oversimplified
estimation of dry bulk density (DBD) from empirical equations of DBD and carbon content where discrete,
volumetric measurements do not exist (Avnimelech et al., 2001; Kastowski et al., 2011), varied approaches used
for the measurement of sample carbon contents (Elemental analyser vs LOI) and radiocarbon dating limitations
that limit effective sedimentation rate derivations (Munroe and Brencher, 2019). Despite this, the calculated
OCARs for Rauchuagytgyn are largely agreeable with multiple studies from northern Regions and reinforce the
reliability and applicability of the presented results.

## 6. Conclusions:

This study aimed to improve the understanding of accumulation rates and pools within a palaeoenvironmental
context within a Chukotkan Arctic glacial lake. The major outcomes are as follows:

- Sediment and carbon accumulation at Lake Rauchuagytgyn since MIS2 can be partitioned into three
principal regimes (early MIS, Mid-to-late MIS2 and Holocene accumulation) distinguished clearly from
multiple environmental proxy records and calculated accumulation rates and pools.
- Early MIS2 accumulation (ca. 29–23.4 cal. ka BP) was controlled by cold glacial-periglacial conditions,
quasi-permanent lake surface ice cover, the presence of a catchment glacier, and low sediment
availability that led generally to low mass (MARs) and organic carbon accumulation rates (OCARs) and
glacigenic sediment deposition.
- Mid-to-late MIS2 accumulation (ca. 23.4–11.5 cal. ka BP) reflects the increasing influence of paraglacial
processes, longer surface ice-free summers, a thickening catchment active layer and increasing moisture
availability. Carbon accumulation increased throughout and accompanied progressive climate
amelioration.
- Holocene (ca. 11.5 cal. ka BP – present) accumulation reflects a warmer and wetter climate with
enhanced runoff within a deglaciated catchment that led to alluvial fan activation and enhanced
catchment fluvial dynamics. Ice-free summers likely led to increased wind-driven processes leading to
more oxygenated lacustrine conditions during the mid-late Holocene and sediment redistribution. Higher
Holocene OCARs reflect the prevailing climatic conditions and possibly the increased preservation under
enhanced sedimentation rates.



- Estimated organic carbon pools reflect the presented accumulation regimes and demonstrates the larger carbon pool preserved within Holocene sediments (0.15 ± 0.005 Mt) when compared with Pleistocene
sediments (0.1 ± 0.007 Mt). The carbon pool is however significantly small in comparison to the total sediment pool (25.7 ± 1.71 Mt).

- Carbon pools estimated for Finnish lake and Yakutian yedoma sediments show the greatest similarity to pools calculated for Lake Rauchuagytgyn sediments. Baldwin Peninsula (Alaska) permafrost landscapes and Alberta glacial lakes (Canada) represent larger organic carbon pools.

- Lake Rauchuagytgyn OCARs resemble estimations for Lake Baikal, Finnish and Canadian boreal and Greenlandic lakes. Estimated rates are however significantly lower than the global lake mean as well as Cherskii-Kolyma thermokarst lakes in Far East Russia.

- The main drivers of OCAR and carbon pools on a temporal scale are related to palaeoclimate variations that control catchment and within-lake processes. Spatial differences with other lake systems is related to
the high-latitude, mountainous setting of Lake Rauchuagytgyn.

**Data availability**

Data used in this study will be accessible from the PANGAEA data repository (dataset is currently in review and a DOI will be provided upon dataset publication).

**Supplement**

See separate supplement file

**Author contributions**

BB and UH designed the study and together with LP organised field–work. SAV obtained hydroacoustic profiling data and was responsible for opening, splicing and subsampling the core as well as all biogeochemical and sedimentological analyses. He also wrote the first version of the manuscript. GP was responsible for the age-
depth model and sedimentation rates. NN provided magnetic susceptibility data. BB supervised the works of SAV. BD assisted in the interpretation of hydroacoustic data. All co-authors contributed to interpretation of the results and commented on the text.

**Competing interests**

The authors declare that they have no conflict of interest

**Acknowledgments**

We acknowledge funding from the Past Permafrost Project under the umbrella of the Earth System Knowledge Platform (ESKP) and the initiative and networking fund of the German Helmholtz Association. This research has been supported by the ERC European Union's Horizon 2020 research and innovation programme (grant agreement no. 772852, Project. Glacial*Legacy*) and the BMBF PALMOD (grant no. 01LP1510D) as well as the
Russian Foundation for Basic Research (grant no. 18-45-140053) and Ministry of Science and Higher Education of the Russian Federation (grant no. FSRG-2020-0019). The authors would like to thank all members that participated in the Chukotka 2018 expedition. We additionally thank Rebecca Morawietz, Jasmin Weise and Sebastian Golinski (AWI-Potsdam) for their assistance during laboratory work.

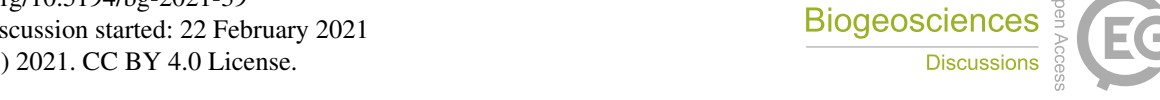



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
