# Peer review of "Sediment and carbon accumulation in a glacial lake in Chukotka (Arctic Siberia) during the late Pleistocene and Holocene: Combining hydroacoustic profiling and down-core analyses"

_Biogeosciences, 2021_

## Referee Comment (RC2)

[referee-annotated manuscript omitted]

---

## Author Comment (AC1)

Responses to Assoc. Prof. Mgr.Daniel Nyvlt

Dear Assoc. Prof. Mgr. Daniel Nyvlt,

Firstly we would like to thank you for taking the time to review our manuscript entitled "Sediment and carbon accumulation in a glacial lake in Chukotka (Arctic Siberia) during the late Pleistocene and Holocene: Combining hydroacoustic profiling and down-core analyses" in detail. We are especially grateful for all your comments and suggestions particularly during this time of the global pandemic. We appreciate highly that you consider the scientific significance and importance of our manuscript for increasing our understanding of carbon storage in Arctic glacial lake systems. In the following response, we provide detailed replies to each individual comment and provide our proposed changes and adjustments to the manuscript that will be carried out and shown within the revised manuscript version. As such your comments are highlighted in black and *italicised* and our replies are highlighted in blue. We hope that are you satisfied with our replies and our proposed changes!

Thank you again once for taking the time to review manuscript,

On behalf of all the authors,

Stuart Andrew Vyse

*Reviewer comments and author responses*

*It should be noted the the MIS1 starts at 14.7 ka in the marine isotope stratigraphy. Therefore your "Mid-to-Late MIS2" should read "Mid MIS2-early MIS1"*

Thank you very much for noting this. We are sorry for the incorrect usage of the marine isotope stratigraphy. We will adapt the phrasing throughout the manuscript so that it reflects your comment to "**Mid MIS2-early MIS1**" within our revised manuscript version. We will also alter the position of the boundary between LU-II and LU-I following your suggestions later within the comments and hence the age ranges will be changed accordingly.

*Lakes act also as sinks of atmospheric deposition, which is not necessarily of a material derived from its catchment...*

We agree with this and the importance of aeolian deposition that can be derived from further afield than the lake catchment area. We will account for this by rewording and adding in extra reference to this in line 40 so that it reads as follows: "Lakes act as sinks of clastic sediment derived from local catchment weathering processes **as well as from atmospheric deposition** and as such gradually accumulate sediment mass over time (**Dietze et al., 2014**; Hinderer and Einsele, 2001). We will add an appropriate reference to **Dietze et al., 2014** that considers aeolian processes within lake sediments from the Tibetan plateau.

*important for what?*

Sorry for the poor wording. "Important" will be replaced with "environmentally sensitive" so that line 55 reads as follows: "The region of Chukotka (Arctic Siberia) represents an

**environmentally sensitive area** with limited lacustrine environmental reconstructions (Lozhkin and Anderson, 2013)".

*do not use however in two subsequent sentences...*

Sorry for this. We will remove the "however" and subsequent comma from line 71 to avoid double occurrences. Lines 69 to 72 will thus read as follows: "The reconstruction of accumulation rates in these syntheses has however been avoided due to significant reworking of carbon material within permafrost landscapes (Strunk et al., 2020; Windirsch et al., 2020). The role of Arctic Siberian glacial lakes as sediment and carbon sinks has not yet been accounted for."

*V-shaped valleys are generally considered fluvial in origin, glacially eroded valleys are described as U-shaped valleys. You should probably better describe this to avoid any confusion. It looks as a U-shaped valley in the Fig 1a.*

We agree with this. The valley is in fact U-shaped and not V-shaped as initially written and thus we accept your suggestion and will change this to a "U-shaped valley" in line 114.  The line will be changed to read as follows: "Lake Rauchuagytgyn (67.7922° N, 168.7312° E) is situated within the glacially eroded **U-shaped**, Rauchua mountain valley...".

*...Cretaceous extrusive and intrusive igneous rocks consisting of silicic-intermediate lithologies by andesite...*

*...to have it in English*

Thank you for making this small change here. We accept the suggestion and will change line 122 to read as follows: "The bedrock surrounding the lake and within the catchment is predominantly composed of cretaceous extrusive and intrusive igneous **rocks** consisting of silicic-intermediate lithologies dominated by Andesite (Zhuravlev and Kazymin, 1999)".

*"moraines", rather than "moraine structures"*

Thank you for the suggestion. We will change it to read "moraines" so that line 124 reads as follows: "Catchment evidence for glaciation includes **moraines** to the north of the lake that denote the maximum extent of glaciation".

*...average July and January temperatures of 13 °C and -30 °C, respectively...*

We will add in "respectively" so that line 126 reads as follows: "The Arctic continental climate of the area is characterized by mean annual temperatures of -11.8 °C and average July and January temperatures of 13 °C and -30 °C, **respectively** with low annual precipitation of ca. 200 mm (Menne et al., 2012)."

*It would be very helpful to add glacial cirques in the Figure 1a*

Thank you for this suggestion. We agree that Figure 1a would benefit from the addition of some extra information regarding the position of glacial cirques alluded to in section 2. We will now add some of the most clearly identifiable glacial cirques from satellite data to the map in the revised manuscript version to account for this. In addition, we will add reference again to Figure 1a in line 125 where glacial cirques are mentioned.

*In inset is shown the situation of Lake Rauchuagytgyn (1) compared to other studied regional lakes: (2) Lake Ilirney and (3) Lake El´gygytgyn (ESRI 2020).*

We agree with your suggestion and will alter the caption text to read "**In Inset is shown** the situation of Lake Rauchuagytgyn (1) compared to other studied regional lakes: (2) Lake Ilirney and (3) Lake El´gygytgyn (ESRI 2020)."

*Enlarged orthophoto map of the lake...*

We will change the caption text to integrate this change. The revised text will read as follows: "**Orthophoto map** of the lake and surrounding features."

*Simplified bathymetric map...*

Thank you for the comment. This comment is however no longer relevant as we will remove the older bathymetric map (Figure 1c) in response to your suggestions in the following comments which replace this with an overview polygon with hydroacoustic profiles.

*Could you please add the hydroacoustic profiles paths in any of the detail map? I think the map in Fig. 1c could be enlarged in profiles could be included without the detailed relief of the lake surroundings.*

Thanks for the comment. We show the hydroacoustic profiles when presenting the interpolation results in figure 3b but we will modify figure 1 in response to your suggestion. As such we will remove the bathymetry and the relief of the lake surroundings that was presented in figure 1c. In the place of figure 1c, we will show an enlarged lake polygon with the plotted hydroacoustic profiles. We will also alter the figure caption to mirror this to "**(c) Lake polygon with hydroacoustic profiles retrieved during 2018**".

*Why do you present an older bathymetry here when bathymetry is one of your main findings in thsi study. I find this unnecessary.*

This comment has been acknowledged in the responses above. We will remove this bathymetric map as you are correct that it is unnecessary to have an older bathymetric map. We will replace the older bathymetric map with a lake polygon with plotted hydroacoustic profiles in figure 1c.

*show the profiles paths in the map*

This comment is acknowledged above in previous responses and will be plotted on a lake polygon in the revised manuscript version as Figure 1c.

*What do you mean by the basal sediment within the basin? I think they might be the Pleistocene pre-lacustrine sediments of ?glacial, or ?(glacio)fluvial origin - it would be helpful to desribe it better.*

Thank you for this comment. We meant the lowermost deposited sediments within the basin which are likely to be of mixed glacial and glaciofluvial origin. Though the core penetration was very limited into these sediments and it is difficult to say based on the very limited sedimentological data for certain what these sediments are and their age. To acknowledge this comment we will alter the description to "Further core penetration and retrieval was prevented by sand-pebble material at the core base most likely representing the **lowermost deposited sediments** within the basin.

*How large the samples for radiocarbon were from the viewpoint of the core depth? It seems they were sampled as 0.5 cm thick what I guess from the Table 1, but I think this should be also stated here.*

We agree with your suggestion. Indeed the samples were taken in 0.5 cm thick slices to retain as high a dating resolution as possible. We will now include a remark to this in the manuscript text to read as follows: "Due to the lack of suitable plant remains and low organic content of the retrieved sediment core, 25 bulk sediment samples **(0.5 cm thickness)** and one surface sample (0–0.5 cm **sediment depth**) were dated for radiocarbon....".

*Why these two samples were not used for age-depth modelling? I see the same possible way in deleting the samples 3002 and 3003, or even only the sample 3002. Please explain better what reasons you have for ommiting these two samples. It clearly shows a higher sedimentation rate in this part of the lacustrine succession.*

Thank you for your comment and input here. We found during the development of the age-depth model that including the two samples or even just sample AWI - 3001.1.1 would produce a sedimentation rate that was unreasonably high. Considering the nature of the sediment deposited during this interval, our sediment core yields limited sedimentological evidence for a drastically higher sedimentation rate across these depths, such as a turbidite event. Thus, we opted for a model that would - based on sediment characteristics - most likely represent a more realistic sedimentation rate for these depths. Moreover, it has been commonly found among Arctic lakes that input of older organic material influences more strongly radiocarbon reliability than younger ages, which supports the exclusion of at least sample AWI - 3002.1.1 from the age-depth modelling (Bronk Ramsey, 2008; Gaglioti et al., 2014; Abbott and Stafford, 1996).

In response to your comment, we have included an extra explanation between lines 162 and 165 as follows: "For modelling, we used 23 bulk sediment samples. Two samples (Lab-ID: AWI - 3001.1.1; AWI - 3002.1.1) were slightly older than their successive dates further down. This suggests possible reworking in these depths (81.25 cm and 114.75 cm, respectively) **and would lead to unrealistically high sedimentation rates when included within age-depth modelling that is not mirrored by sedimentological proxies**. We thus treated these two dates as outliers and excluded them from the modelling process."

*Mn and Fe are also rock-forming elements...*

Thank you for pointing this out. We will restructure lines 183 and 184 to read more efficiently as follows: "The elements (Aluminium (Al), Silicon (Si), Calcium (Ca), Potassium (K), Titanium (Ti), Manganese (Mn), Iron (Fe), Bromine (Br), Rubidium (Rb), Strontium (Sr), Zircon (Zr)) were selected for further processing".

*?element proportions*

We have now removed this sentence and hence this comment is no longer applicable. Based on suggestions from reviewer #1 we use instead the simple, non log-transformed elemental ratios which show the same patterns as the log-transformed ratios.

*The boundary of 4 micrometers is used in most sedimentological literature for the clay/silt boundary. The 2 microns boundary is more common in geochemical studies.*

Thank you for pointing this out. Indeed Udden (1914) and Wentworth (1922) put the division between clay and silt at 4 micrometers but later studies such as Friedman and Sanders (1978) set the boundary between clay and silt instead at 2 micrometers. As we have used the very commonly (cited 1928 times) used software "Gradistat" from Blott and Pye (2001) to process our grain-size data that utilizes the Friedman and Sanders (1978) boundary at 2 micrometers, we will opt to retain our clay/silt boundary here. We will add an additional citation to the grain-size classification of Friedman and Sanders (1978) to line 202 so that it reads as follows: "Intervals of 2 mm–63 µm, 63–2 µm and <2 µm, were used to define percentages of sand, silt, and clay respectively (**Friedman and Sanders, 1978**)".

*The Folk and Ward method... upper case in names*

Sorry for this. We will now capitalize these letters here. Line 202 will now read as follows: "The **F**olk and **W**ard method was used for mean grain-size calculation."

*et al.*

Thank you for noticing this. It will be corrected to in the revised manuscript version Avnimelech **et al.** (2001).

*Do not start a sentence with a number. e.g.: Subsequently, 65 dried and milled...*

We agree with your comment here and change line 214 to read as follows: "Subsequently, 65 dried and milled.........".

*delete 3x the commas after al.*

Thank you for seeing this. The three commas will be deleted after the al. in line 245.

*Is the term shelf correct when referring to the lake? I would prefer the describe it as submerged paleoterrace.*

Thank you for this comment. The term "shelf" has been used within published literature to describe similar features at other Arctic glacial lake sites. Examples may be observed in Lebas et al. 2019 "Seismic stratigraphical record of Lake Levinson-Lessing, Taymyr Peninsula: evidence for ice-sheet dynamics and lake-level fluctuations since the Early Weichselian" where similar terminology was utilised in reference to hydroacoustic and seismic data. We will thus opt to retain the term "shelf" here.

*Please unify the use of Birtish vs. American English. The text is written in American English, but in the Figure appear "Palaeoterrace", which is written in British English.*

Sorry for the mixed usage of British and American English within this manuscript. We will thoroughly check the manuscript for discrepancies existing between British and American English and the revised manuscript will be written only in British English with all discrepancies corrected.

*northern-shelf - see my comment above*

This comment is addressed in a previous response. As such we will retain the usage of "shelf" as it has been used to describe similar features within other Arctic glacial lake systems (see Lebas et al. 2019).

*shelf - see my comment above*

This comment is addressed in a previous response. As such we will retain the usage of "shelf" as it has been used to describe similar features within other Arctic glacial lake systems (see Lebas et al. 2019).

*I would avoid using the term hummocks here, as hummocky-cross stratification/bedding is a sedimentary feature, which leads to a sediment morphology, rather than to an erosional landform as is the case here. Use evelations and depressions as not genetically-bounded terms here.*

Thank you for this suggestion. We agree that this could be a confusing. We will instead opt to use the proposed terminology of "elevations and depressions" that you suggest. Lines 297-298 will thus read as follows: "AU2 possesses a volume of ca. 18055352 $m^3$ (0.018 $km^3$) with complex internal architecture with **elevation and depression-like** structures."

*kilometers - either "km", or "kilometers" (lower case initial) - all parts of the figure*

Thank you for pointing this out. We will now correct all the upper case initials to lower case initials in figure 3 within the revised manuscript version and check that this is the case in all other figures.

*The same thickness scale in Figs 3b, c, d would help to visually see the differences. Now the pink is 6-7 m in 3b, 8-9,5 m in 3c and 13-15 m in 3d - this is not good for visual comparison.*

Thank you for this hint to improve the readability of figure 3. We accept your suggestions and will adopt this in the new revised version of Figure 3. We will adjust this so that each interval is adapted to 1 m rather than the scale differences that you have noted.

*the artifacts are coloured in white, which is also the colour for the most thick sequence in Fig. 3c and 3b. I think using grey colour for the gaps (artifacts) would be better, as this colour is out of the scale used.*

Thank you for noticing this. We agree that is a little bit confusing within the figures and thus we agree with your suggestion and will change the colour of the artifacts to grey to account for this in the revised manuscript version.

*these figures show that the delta at the inflow is rather of Pleistocene origin, but how you know that it is of Pleistocene age as both parts (Pleistocene and Holocene) were not delimited in this part of the lake basin as shown in Fig. 2b.*

Thank you for this comment. It is difficult from the presented data to know the true age of the delta and hence we do not assign a strict age to the delta. We only know that processes have been operating at the delta during the Holocene due to the observance of active fluvial channels and coarser-grained surface sediments proximal to the delta that evidence input of fluvial material. What would certainly help us to know more information about the age of the delta would be reduced acoustic blanking in this area and perhaps the usage of another method such as seismic stratigraphy and the retrieval of cores through the delta sediments. Unfortunately, acoustic blanking, no additional seismic data, and no cores from this region hinder further interpretation as to the age of the delta here.

*shelf - see my comment above*

This comment is addressed in previous responses. As such we will retain the usage of shelf as it has been used to describe similar features within other Arctic glacial lake systems (see Lebas et al., 2019).

*Is the age of 29 cal. ka BP modelled, or calibrated - I see this is a calibrated age of the lowermost sample - why do you mention modelling here? And how does Elias and Brigham-Grette, 2013 refer to this information? This is simply an inappropriate reference here, as the age is part of your results! Besides, references should not be used in the Results chapter, as it should contain your results and no reference are needed here! Some journals even prohibit the use of references in the Results chapter.*

Thank you for pointing out this lack of clarity. It is common in the age-depth modelling domain that both calibrated radiocarbon samples and modeled ages are expressed as "calibrated years BP", in our case "cal. ka BP". Age determination samples are calibrated internally in the age-depth modelling software and hence the depth in between those samples are regarded as calibrated as well.

As stated in the current manuscript version we used the age-depth modelling software "Undatable" (Lougheed & Obrochta, 2019). In this software the lowermost point in the age-depth model does correspond with lowermost radiocarbon sample we took. Hence, we say "modeled" to emphasize that we derived the value from the age-depth modelling software instead of a separate calibration program, i.e. CALIB 8.2 (Stuiver et al., 2020).

Sorry for the inappropriate references here. We agree that results should not contain references wherever possible. We will subsequently move the references (Abbott & Stafford, 1996; Björck & Wohlfarth 2002) to the discussion section 5.2.3 where we will add some extra lines from lines 716 to 720 to consider possible reworking to read as follows: "from empirical equations of DBD and carbon content where discrete, volumetric measurements do not exist (Avnimelech et al., 2001; Kastowski et al., 2011) **as well as** varied approaches used for the measurement of sample carbon contents (Elemental analyser vs LOI) (Munroe and Brencher, 2019). **Limitations associated with radiocarbon dating of Arctic glacial lakes due to the absence of appreciable amounts of datable organic material as well as the influence of reworking processes associated with permafrost and glacial processes can also lead to uncertainty with regards to actual sediment ages and hence reconstructed accumulation rates (Abbott & Stafford, 1996; Björck & Wohlfarth 2002).**"
We will also remove the reference to Elias and Brigham-Grette, 2013 from line 306 and also check for further uses of references in the results chapter that will be subsequently removed.

*See my comment above regarding the two samples omitted from the age-depth model calculation.*

Thank you for the comment. We have addressed this comment in a previous response. We excluded these samples on the basis of an older age of sample 3002 relative to lower samples that suggested reworking and input of old carbon. Moreover, the inclusion of samples 3002 and 3001 within age-depth modelling would produce unrealistically high sedimentation rates that would not be reconcilable from the sedimentological data that shows no evidence of a drastic increase in sedimentation rate in these depths. We have thus made changes to lines 162 to 165 to account for your suggestions.

*Isn't the larger age scatter in Pleistocene samples (and age model), when compared with Holocene ages, connected with a larger age scatter after radiocarbon age calibration, because of larger uncertainties and less data for calibration curve calculation?*

Thank you for your question. We are using the newest published calibration curve (IntCal 2020) in this study that was published by Reimer et al. 2020 and hence the calibration curve represents the most recent state-of-the-art dataset with reduced uncertainties compared with previous calibration curves. Moreover, the uncalibrated years already show scatter and therefore the calibration curve likely has little influence. It is more likely that the scatter is related to lake development processes possibly related to the presence of a catchment glacier within this interval. We have taken account of this scatter by including sigma ranges in sedimentation rate calculations.

*a comma before but - "...low rates, but with..."*

Thanks for pointing this grammatical mistake out. Line 315 will now be corrected to the following "demonstrates **low rates, but with** larger uncertainty".

*"...uncertainty ranges (dark and light grey ribbons)." 1sigma is dark grey 2sigma is lighter grey*

Thank you for noticing this. We will adopt your change so that the caption of figure 4 reads as follows: "1 and 2σ uncertainty ranges (**dark and light** grey ribbons)."

*What do you mean by: Br/Al ration values demostrate their lowest values at any depth? It clearly has the lowest values in LU-III, not at any depth...*

Sorry for this confusing sentence structure here. We agree with your suggestion and will change line 336 and 337 to read as follows: "Br/Al ratio values demonstrate their lowest values **within LU-III** alongside TC..........".

*"cal. ka BP" - lowercase cal.*

Thank you for noticing this. We will correct to the lowercase **cal.** as you suggest. We will also check occurrences throughout the manuscript to ensure "cal." is written in lowercase.

*The Grain-size box should be logically made from the left by clay, then silt in the middle and sand to the right - it is very erroneous to put sand between clay and silt. Why the terms initiate with upper cases? - this is inappropriate here.*

We agree with your comment and feel that the current presentation is not logical. We will follow your suggestion by altering figure 5 to show clay, silt, and sand in that order as suggested. The starting letters of the terms displayed in figure 5 will also be altered to lower case as suggested.

*What do you mean by high grain-size? Is a coarse grain-size?*

Sorry, here we meant "coarser mean grain-size". We will subsequently alter the figure caption text of figure 5. to the following "mean grain-size plot refers to three excluded data points (650.5, 341, 321 cm) of **coarser mean grain-size** (up to 25 μm)."

*"cal. ka BP" - lowercase cal.*

Thank you for noticing this. We will correct the lowercase cal. in Figure 6 as you suggest as was also the case for Figure 5. Again, we will also check this throughout the manuscript.

*Why is the TOC displeyed twice? I do not see the need to show it in wt% and retain only the g OC cm-3*

Thank you for the comment. We agree with you that we do not need to present TOC twice within figure 6. Removal of one of the TOC curves will also improve the overall readability of Figure 6. We decide in this instance to remove the TOC curve in g OC cm$^{-3}$ as the wt% curve displays both the TC and TOC curves and is more traditionally presented within palaeolimnological studies.

*Basing on all graphs I have seen, I would put the boundary between LU-II and LU-I slightly lower!!! It is impossible to put the boundary at the peak of TOC (and some toher proxies), I strongly recommend to put this boundary slightly lower to have the peak already in LU-I, not at the boundary and it will probably also fit better with the beginning of the Holocene (11.7 ka b2k, rather than your 11.5 ka BP). I would put the boundary in the mid-point of the PC1 score rapis increase, if it fits with lithological data. Please think about this change!*

We thank you for this important suggestion and we agree with it. We will move the LU-II and LU-I boundary to the suggested mid-point of the rapid increase in PC1 scores at a depth of **346** cm that actually corresponds to an age of **11.69** cal. ka BP and hence more effectively represents the Holocene start. We will then correct this boundary in all figures and text passages that show and/or refer to this boundary to accommodate for this change throughout the manuscript. Thank you again for making this suggestion.

*sand, silt, clay, mean GS - lower cases - both in text and in Figure 7*

Sorry for the capitalization here. We will now change the first letters to lower case as per your comment in both the text in line 388 and in Figure 7.

*What is Early MIS 2, Late glacial and Holocene and how it relates to LU-I, LU-II, or LU-III units? Please explain it better in figure caption.*

Thank you for this comment. We will change the terms within figure 7 to LU-I, LU-II, and LU-III as the grouping is based on the lithological unit definition. We will slightly modify the caption of figure 7 to read as follows: "Principal component analysis **(PCA)** biplot of sedimentological, biogeochemical and accumulation rate data from core EN18218. **Samples are coloured and clustered according to their lithological unit definitions (LU-III, LU-II, LU-I).**"

*palaeo - British vs. American English*

Sorry for this. As we have mentioned previously within our responses, we will check the use of British vs American English throughout the manuscript and ensure that the manuscript is standardized in British English throughout.

*This is incorrect use of a reference. Mangerun and Svendsen worked on Svalbard - how this relates to Chukotka? You should rather use here "(cf. Mangerud & Svansen, 1990)."*

Sorry for this reference error. We agree with your suggestion and will use instead (**cf**. Mangerund & Svendsen, 1990) in line 397. We also follow all suggestions of reference changes to (cf.) notation where suggested.

*rather "basal sediments" than "basement sediments"*

Thank you for this suggestion. We will now refer to "basal" sediments within the revised manuscript version.

*Again, the term "basement" is mostly used for solid rocks, rather than for sediments by geologists.*

Ok, we will refer instead to "basal structures" in the revised manuscript version.

*again - (cf. Lebas et al, 2019; Lebas et al., 2021).*

We agree with your comment and will use now (**cf.** Lebas et al., 2019; Lebas et al., 2021) for the references in line 404.

*Could you explain how can a glacigenic sediment be layered? Glacigenic sediments are those deposited directly from glacier without subsequent sorting, which means tills and tills are hardly layered. But glaciofluvial and glaciolacustrine sediments could be layered. Please make this clear.*

Sorry, we understand that the uppermost portion of section 5.1.1 may be unclear as currently written. We will make adjustments to this section in order to address your comments here that will be included within the revised manuscript version.

Firstly, we will no longer refer to these sediments as "glacigenic". You are correct that we see some layering within this unit that most likely relates to deposition under glacio-lacustrine conditions with input of sediments derived through catchment glacial activity. Our interpretation of this unit is based on similar published findings at Harding Lake presented by Finkenbinder et al. (2014) as well as glacio-lacustrine sediments deposited in Lake Silvaplana by Leemann and Niessen, 1994b. At Harding lake, following deposition of a basal pebble-diamicton prior to ca. 30.7 cal. ka BP at the base of unit 1, which was tentatively interpreted to represent braided river sediments at Harding lake, sedimentation from ca. 30.7 cal. ka BP to 15 cal. ka BP throughout the rest of unit 1 consisted of finer-grained deposition of highly minerogenic sediment of high magnetic susceptibility, high and variable dry bulk density (of very similar values represented for much of LU-III at lake Rauchuagytgyn), and an abundance of the element Titanium. These sediments were interpreted to represent lacustrine sediments deposited continuously within a perenially ice-covered lake that may have been shallow with enhanced minerogenic sediment input during the global last glacial maximum. Moreover, sedimentation rates within unit 1 at Harding Lake displayed comparable values to those calculated for LU-III at lake Rauchuagytgyn of ca. 0.01 cm/yr.

In response to these sediments being layered, studies of glacially influenced layered glacio-lacustrine sediments from Lake Silvaplana by Leemann and Niessen, 1994b have shown that catchment glaciation can lead to the synchronous deposition of fine-grained (clay & silt) suspension load within layers. This may be a similar mechanism to that acting at lake Rauchuagytgyn during this interval. Karlen and Matthews (1992) have similarly reported the occurrence of silt/clay bands found within glacio-lacustrine sediments from southern Norway that possessed low organic content with high magnetic susceptibility that likely represent

minerogenic sediment bands associated with catchment glaciation. Moreover, van der Bilt et al. 2015 investigated sediments deposited in a lacustrine setting in Svalbard and suggested that silt and clay sediments of high bulk density, magnetic susceptibility, and titanium concentration represented glacial suspended load that could be used to identify periods of glacial advance. Sediments that were associated with glacigenic suspended load for example in unit 1 of van der Bilt et al. (2015), demonstrated layering in the form of centimeter scale lamination.

The scatter in radiocarbon ages reported for LU-III may also relate to scatter caused by glacial reworking of organic material from the lake catchment that has been noted at other glacial lake sites during glaciation. Based on this, it is our interpretation that the sediments deposited within LU-III could represent glacio-lacustrine sediments that were influenced by the input of minerogenic glacial suspended load in the form of silt and clay from a catchment glacier.

In the revised manuscript version, we will thus make this distinction clearer and refer to "glacio-lacustrine" deposition as well as "minerogenic glacial suspended load" and include additional references to support our interpretations along with some restructuring to account for your comments.

*How do you know that these are glacigenic sediments? Sediments and their facies should be described based on their texture and structure, not based on colour, elemental composition, or PCA. You should be more cautious when using sedimentological term without knowing the sedimentology of the material!*

Thank you very much for your comment here, we have partially considered this comment in the previous response. We meant to refer to suspended sediments that have been derived through glacial processes acting within the lake catchment. In general, we originally meant to use the term "glacigenic" to refer to minerogenic glacial suspended load deposited within a glacio-lacustrine setting which is generally dominated by clay and silt and characterised by a high bulk density as has been described by Leeman and Niessen, 1994b, Bakke et al. (2005) and van der Bilt et al. (2015). Van der Bilt et al. (2015) for example, referred to glacigenic sediment when discussing the input of clay and silt recorded by lacustrine sediment cores in Svalbard. Moreover, Van der Bilt et al. (2015) also used multivariate statistical approaches (PCA) to effectively "fingerprint" the sedimentological and geochemical nature of glacigenic sediment in a similar manner to that what we have carried out in this manuscript. As mentioned in the previous response, studies of glacio-lacustrine sediments from southern Norway have also suggested that sediment bands of silt/clay composition with high magnetic susceptibility represent glacigenic sediment deposited within a lacustrine environment (Karlen and Matthews, 1992).

*again, Van der Bilt et al., 2015 did not work on Chukotka, so "cf. Van der Bilt et al., 2015" would be better to refer to.*

Thank you for pointing this out. We will now refer to (**cf.** Van der Bilt et al., 2015).

*The principal question is - was there a lake at that time? Probably not, as you wrote about glacigenic sediments. So, why you write about lake here? It is all about the sound interpretation of individual sedimentary facies described in the section, without that it is very hard to follow your story!*

Sorry for this lack of clarity. We will now carry out changes to this section to make our story clearer as mentioned in the previous few responses. It is our opinion that apart from the coarse-grained basal sediments, that LU-III predominantly represents glacio-lacustrine sedimentation that was influenced by the input of minerogenic glacial suspended load from a catchment glacier. The lake was likely also covered with a quasi-permanent surface ice layer that acted to reduce sediment input and shares similarities to continuous lacustrine deposition that was interpreted for unit 1 at Harding Lake in Alaska (Finkenbinder et al., 2014).

*Which of the two Kokorowski et al. 2008 this is? They should be refered to as Kokorowski et al. 2008a and 2008 b both in the text and in the list of references.*

Sorry for this. We meant to refer to Kokorowski et al. 2008**a** in this case. We will ensure that at each reference to Kokorowski et al. 2008, the correct **a** and **b** notation is added throughout the manuscript. We will also add this notation to the references of both Kokorowski et al. 2008 papers within the references section in lines 1063 and 1066.

*?or in line with non-lacustrine environment?*

Thank you for this comment. We have addressed this comment in previous responses addressing section the upper part of section 5.1.1. We envisage a glacio-lacustrine environment for much of LU-III.

*again - "cf. Baumer et al., 2020;..."*

Ok, thank you for pointing this out. We will now alter the references here to **cf.** i.e. (**cf.** Baumer et al., 2020; Biskaborn et al., 2019; Fritz et al., 2018; Heinecke et al., 2017; Naeher et al., 2013).

*Is really Zr/K a proxy to coarse-grained lithology? What do you mean by "coarse" here? For most geoscientists coarse would mean gravel. Generally, all facies described in this study are rather fine-grained. Therefore, you might explain it better here. Generally, the Zr/K index increases with an increase of coarse silt to medium sand fraction, what we found in numerous our studies from Central Europe, Svalbard, Greenland, or Antarctic Peninsula. For detection of even coarses grains (fine sand to fine gravel), the Si/Al index is used, as quartz is a predominant in sand to fine gravel fraction and aluminosilicates are common in clay to medium silt fractions.*

Thank you for this comment. We agree that the sediment deposited within the lake is generally fine-grained in nature and should not be referred to as "coarse". In the revised manuscript we will be more cautious with the application of "coarse" and will subsequently alter lines 425 and 426 to read as follows: "**Finer** grain-sizes directly measured by laser diffraction are supported by indirect, XRF-derived grain-size proxies **for finer**, clay-dominated sediment (K/Ti)." In our context, we meant the coarsening of the fine-grained sediments that is represented by the Zr/K ratio and likely reflects the increased contribution of coarse silts and sand fractions to the grain-size signal during the Holocene. The Zr/K ratio as you said, is likely related to the increase in the proportions of coarse silt and very coarse silt and some sand fractions that is observed within Holocene sediments. Equally if the Si/Al ratio is used as a grain-size proxy we see increasing values of Si/Al within Holocene sediments that may also be related to grain-size increases. Si/Al may however relate additionally to biogenic sediment due to the occurrence of diatoms within the lake sediments.

*Harding Lake*

Sorry for this spelling mistake in line 436. We will follow your suggestion and change to "Harding Lake". We will also check for further misspellings of "Lake" throughout the manuscript.

*Is really 0.15 mm a-1 a low sedimentation rates? It equals to 15 cm/ka, which is higher that what is described above. It would be helpful to show comparison to Holocene sedimentary rates in Tajikistan.*

Thank you for pointing this out. Sorry, we meant to say lower Pleistocene sedimentation rates in comparison to the Holocene at Lake Karakul. We will alter the phrasing of these lines to make this clearer and also add in the Holocene sedimentation rate for comparison as suggested. Line 440 will now read as follows: "A similar finding at lake Karakul, Tajikistan of **lower** sedimentation rates during MIS2 **since ca. 29 cal. ka BP** (0.15 mm $a^{-1}$) **when compared with Holocene sedimentation rates (0.84 mm $a^{-1}$)** was also explained by reduced sediment input during MIS2 compared to the Holocene alongside reduced organic matter accumulation (Heinecke et al., 2017)".

*Rauchuagytgyn*

Sorry for the spelling mistake. This will be corrected in line 442 and will be checked for correctness throughout the rest of the manuscript.

*Late Glacial*

Sorry for the spelling mistake here too. Will be changed in line 443 and checked throughout the manuscript.

*Last Glacial Maximum*

We apologize again for the spelling mistake here. It will be changed in the revised manuscript version in line 446 and checked throughout the manuscript.

*"glacial erosion", rather than "glacial denudation"*

Thank you for this suggestion. We agree and will change "denudation" in line 449 to "**erosion**" in the revised manuscript.

*When was the glacier 25 km long? This is important to describe it here. If it was during the LGM than the early MIS2 facies are very probably glacigenic and not lacustrine. This is what is not well described and proven in the entire manuscript.*

Thank you for this comment. According to Glushkova, 2011, the glacier was likely a length of ca. 25 km and was suggested to have been so during the Sartan glaciation which is synonymous with glaciation during MIS2. This was somewhat of a relative approach, as no absolute dating methodologies were utilized and up to now, have not been available. Some dating methods, for example using cosmogenic nuclides on glacial features in the catchment, may provide some additional information regarding the timing, but is unfortunately not available for the Rauchua river valley. We will follow your suggestion and add this information in the revised manuscript version so that lines 454 to 457 read as follows: "Remote sensing based studies of Chukotkan glacial geomorphology and structures within the Rauchua valley have suggested that the catchment glacier was likely a passive glacier,

ca. 25 km in length that extended along the length of the Rauchua river valley and discharged into the Rauchuagytgyn basin **during marine isotope stage 2** (Glushkova, 2011)." We have addressed the rest of the comments regarding glacigenic suspended load input to the lake basin in previous responses. We consider sediments deposited within LU-III to be generally of glacio-lacustrine origin.

*...not contributed significant sediment volume to the lake...*

Thank you for this suggestion. We will alter lines 457 to 459 to read as follows: "Thus, the catchment glacier may have been predominantly non-erosive during the early MIS2 and **not contributed significant sediment volume** to the lake basin supporting the low rates discussed here (Gurnell et al., 1996).

*See my comment in the abstract for the timing of MIS2*

Thank you. We have noted your comment within the abstract and have agreed with your suggestion and will alter this throughout the manuscript to "**Mid MIS2-early MIS1**". The discussion title in line 472 will now be altered to "5.1.2 **Mid MIS2- early MIS1** accumulation during progressive climate amelioration (ca. 23.4–11.69 cal. ka BP)". We will also check for more occurrences within the manuscript and adjust accordingly.

*(cf. Lebas et al., 2021;...)*

Sorry for this. We will now correct the reference to (**cf.** Lebas et al., 2021) in line 477.

*I do not agree completely with this! Alexis Dreimanis made already 80 years ago pioneering studies (summarised e.g. in Dreimanis and Vagners 1971 In: Goldthwait RP, ed.: Till, a Symposium, Ohio State University Press) of what terminal grade (grain-size) are produced by glacial grinding and milling and found for most minerals that the final granolometry is in fine to middle silt fraction. Clay-sized grains are assumed to be sourced from clay minerals within the bedrock only (Haldorsen 1983 Norsk Geologisk Tidsskrift), but in principle most of the glacially derived material lies in silt fraction (e.g. Haldorsen 1981 Boreas). This means that rock flour is principally silt-sized. Besides, when applied 2 microns as the boundary between clay and silt then most of the glacially grinded and milled terminal grades would terminate in silt fraction, i.e. >2 micrometers.*

Thank you for your comment. We appreciate your input on the interpretation here and we recognize that these studies have suggested that rock flour is principally silt-sized. Despite this, it has been recognized more recently from the exceptional, ca. 155 ka Owens Lake record, that glacially produced rock-flour can be found within the clay-size fraction (Bischoff et al. 1997). To account for your comment, we will alter lines 477 to 480 to read as follows: "A clay maximum and grain size minimum at ca. 550 cm (ca. 22.7 cal. ka BP) may suggest initial increases in lake water-depth through glacial melt additions that could have led to the observed reduced SRs and MARs this time. This may be supported by the high values of K/Ti (clay contribution) and low values of Zr/K (proxy for coarser grain-sizes) (**Kilian et al., 2013**; Křbek et al., 2017; Cuven et al., 2010)." We will also include an additional reference of K/Ti as a proxy for clay contribution (**Kilian et al., 2013**).

*This may be supported...*

We will change this in line 480 in the revised manuscript version.

*...controlled by inflowing rivers...*

Thank you for noticing. We will alter this in line 490 in the revised manuscript.

*Late Glacial*

Sorry for the misspelling. This will be corrected in line 506 in the revised version.

*Late Glacial*

Sorry for the misspelling. This will be corrected in line 510 in the revised version.

*What do you mean by "small grain-size fining"?*

Sorry for this poor wording. We meant a small reduction in sediment mean grain-size. We will subsequently adopt this wording to make it clearer so that line 519 reads as follows: "A reduction in accumulation rates ca. 12.6 to ca. 11.5 cal. ka BP, broadly associated with **a small reduction in sediment mean grain-size.......**".

*Younger Dryas*

Thank you for noticing the lack of capitalization of "Younger". It will be corrected for in the revised manuscript version.

*This sentence does not provide any real information - what do you mean by a "more limited Younger Dryas event"? Is it meant as a glacial event, a climatic event - clarify this!*

Thank you for this comment. We will phrase this differently in the revised manuscript version and will remove the "more limited Younger Dryas event" as it does not add any real information as you stated. We have also corrected for the Kokorowski paper as we were referring to the Kokorowski et al. 2008**b** paper here. As such lines 523 and 524 now read as follows: "These findings are consistent with recent regional and transregional records that suggest a **spatially variable Y**ounger **D**ryas **climatic event** in **F**ar and **E**ast Russia and parts of **E**astern Beringia (Anderson & Lozhkin, 2015; Kokorowski et al., 2008b; Lozhkin & Anderson, 2013; Lozhkin et al., 2018)".

*Younger Dryas, Far and East Russia, Eastern Beringia*

Sorry for these misspellings. They will be corrected in the revised manuscript version and have been addressed in the response to the previous comment. We will also check for correct capitalization of these words throughout the manuscript.

*So, why you have put the boundary between LU-II and LU-I to the higher TOC value? Because the organic proxy values are decreasing now, as the highest value is at the boundary.*

Thank you for the comment. We have addressed the boundary between LU-II and LU-I in previous responses. The boundary will now be shifted to **346** cm (**11.69** cal. ka BP) and hence the highest TOC will no longer be directly at the boundary between LU-II and LU-I as was previously the case. Please be aware that we will also correct all the ages referring to the previous LU-II and LU-I boundary within section 5.1.3 to account for the movement of the boundary position to 346 cm.

*(Figs 5, 6, 7) - no dot after Figs and spaces after comma*

Many thanks for pointing this out. We will follow your suggestion and remove the dot and add spaces after each comma in line 529.

*...for a local Holocene thermal maximum...*

We agree that we should refer to a "Local" Holocene thermal maximum. We thus accept your suggestion and will utilize it in the revised manuscript version so that lines 541 and 542 read as follows: "which show evidence for a **loca**l Holocene thermal maximum ca. 10.6–7 cal. ka BP (Andreev et al., 2021)." We will also include this for all occurrences of "Holocene thermal maximum".

*...greater sand proportion may relate... why so complicated?*

Sorry for making this sound unnecessarily complicated. We will restructure lines 543 and 544 to make them easier to read by removing "to the grain-size distribution". The lines will thus read as follows: "Increasing early Holocene sediment and mass accumulation rates alongside greater sand contribution may relate to the input of coarser grained fluvial detrital input from a paraglacial....."

*Late Glacial*

Sorry for the misspelling. This will be corrected in line 547 in the revised version.

*Late Glacial*

Sorry for the misspelling. This will be corrected in line 551 in the revised version.

*Westerlies*

Thank you for noticing. We will correct by removing the capitalization in line 554.

*Last Glacial Maximum*

Will be corrected by capitalization in the revised manuscript in lines 554 and 555.

*?finer grain-size*

Thank you for the suggestion. We will now alter "lower grain-size values" to "finer grain-size" so that lines 563 and 564 read as follows: "alongside **finer grain-size** may reflect some local environmental change".

*Why is the reference given twice in one sentence?*

This was a referencing mistake. We will remove the second occurrence of the reference on line 584 (Francke et al., 2013) to correct for this.

*?a shortening of the summer open water season.*

Thanks for the suggestion. We agree with your wording and will change line 588 to the following "This could tentatively be interpreted to represent a **shortening of the** summer open water season."

*...coarser fluvial and alluvial detrital material...*

Many thanks for the suggestion. We agree with your suggestion and will adopt the phrasing so that line 594 will read as follows: "paraglacial processes that resulted in an input source of **coarser fluvial and alluvial detrital material** into the southern sub-basin...".

*again, Peter Doran have not worked here - cite it as follows: cf. Doran, 1993...*

Sorry for this. We will change the citation to your suggestion so that line 595 will read "mass accumulation rates (**cf.** Doran, 1993; Smith and Jol, 1997)."

*Figs*

Thanks for pointing this out. We will remove the decimal point from line 597 so that it will read as follows: "front (site EN18220) (Fig. 1 & **Figs** S5, S6)". As aforementioned, we will also check the entire manuscript for incidences and correct for these too.

*....sediment transport into the deeper...*

We accept your suggestion and will change line 608 to read as follows: "This likely represents low deposition due to **sediment transport into the deeper basin** and feasibly.......".

*Are really the lakes in Greenland boreal?*

Thank you for the question. The use of Boreal should have referred to lakes studied in North America and northern Europe and not to Greenland, where the studied lakes were predominantly of proglacial or bedrock-catchment type (Perren et al., 2009). We will subsequently alter line 616 to read as follows: "Comparisons must therefore be drawn to Boreal lakes from North America and northern Europe, **as well as to proglacial and bedrock-catchment lakes from Greenland**." We will also check the usage of Boreal for Greenlandic lakes throughout the manuscript.

*ice free*

Thanks for noticing. We will now remove the capitalization in line 627 so that is reads as follows: "accumulation calculated for Finnish Boreal lakes that became **ice free** at the Holocene start".

*Great figure to compare boreal to polar northern hemisphere lakes' OCAR!*

Thank you very much for this nice comment to figure 8.

*yedoma - lower case sediments - lower case*

Sorry for the incorrect capitalization within the figure caption of figure 9. We will remove the capitalization in the revised manuscript version so that it reads as follows: ".......Bykovsky thermokarst lagoons and Central Yakutian **yedoma** deposits. Rauchuagytgyn **sediments** possess......".

*I do not think that yedoma and alas are local names, therefore they should be written with lower case initials.*

We agree with your suggestion as yedoma and alas refer to permafrost deposits and not to areas. We will hence remove capitalization of "yedoma" and "alas" in lines 662 and 663 so

that they now read "(**yedoma**: 0.057 Mt, 5.27 kg m-3, **alas**: 0.032 Mt, 6.07 Kg m-3)" and check this throughout the manuscript.

*kg - lower case*

Sorry for this. It will be corrected in the revised manuscript version in line 663.

*Aeolian*

This will be corrected in line 685 to remove the capitalization.

*What about to apply the geomorphological concept of connectivity?*

Thank you for this great suggestion. We feel however that applying this concept fully would be beyond the scope of the current manuscript. That being said, we will certainly add a mention of this concept within section 5.2.3 as well as an additional reference to the paper of Singh et al. 2021 "Geomorphic connectivity and its application for understanding landscape complexities: a focus on the hydro-geomorphic systems of India".

The geomorphological concept of connectivity will be mentioned from lines 704 to account for this as follows: "**As geomorphic systems are hierarchical and operate at multiple spatio-temporal scales according to the concept of geomorphic connectivity, the diverse linkages and interrelationships between different catchment components and processes likely plays a large and complex role in regulating sediment and carbon dynamics at lake Rauchuagytgyn (Singh et al., 2021)**".

*...future changes in sediment and carbon...*

Thank you for noticing this mistake. We will change line 709 to read as follows: "....could lead to future changes **in** sediment and carbon dynamics that are yet.....".

*two times within in the sentence - maybe to changes as follows: ...palaeoenvironmental context of a Chukotkan...*

We will follow your suggestion and adjust lines 724 and 725 as follows: "This study aimed to improve the understanding of accumulation rates and pools within a palaeoenvironmental context **of a** Chukotkan Arctic glacial lake......".

*See my comment in the abstract for the timing of the MIS2*

Thank you for this. We have addressed this in previous responses and will follow your suggestions for this. As such, line 733 will read "**Mid MIS2-early MIS1 accumulation (ca. 23.4–11.69 cal. ka BP)** reflects the increasing influence of paraglacial processes, longer surface ice-free summers, a thickening catchment active layer, and increasing moisture availability. Carbon accumulation increased throughout and accompanied progressive climate amelioration."

References used in this author response

Bakke, J., Lie, øyvind, Nesje, A., Dahl, S.O., Paasche, ø.: Utilizing physical sediment variability in glacier-fed lakes for continuous glacier reconstructions during the Holocene, northern Folgefonna, western Norway, Holocene, 15, 161–176. https://doi.org/10.1191/0959683605hl797rp, 2005.

Bronk Ramsey, C.: Deposition models for chronological records, Quat. Sci. Rev., 27, 42-60, https://doi.org/10.1016/j.quascirev.2007.01.019, 2008.

Dietze, E., Maussion, F., Ahlborn, M., Diekmann, B., Hartmann, K., Henkel, K., Kasper, T., Lockot, G., Opitz, S., and Haberzettl, T.: Sediment transport processes across the Tibetan Plateau inferred from robust grain-size end members in lake sediments, Clim. Past, 10, 91–106, https://doi.org/10.5194/cp-10-91-2014, 2014.

Finkenbinder, M.S., Abbott, M.B., Edwards, M.E., Langdon, C.T., Steinman, B.A., Finney, B.P.: A 31,000 year record of paleoenvironmental and lake-level change from Harding Lake, Alaska, USA, Quat. Sci. Rev., 87, 98–113, https://doi.org/10.1016/j.quascirev.2014.01.005, 2014.

Friedman, G.M. and Sanders, J.E.: Principles of Sedimentology, Wiley, New York, 1978.

Gaglioti, B. V., Mann, D. H., Jones, B. M., Pohlman, J. W., Kunz, M. L., and Wooller, M. J.: Radiocarbon age-offsets in an arctic lake reveal the long-term response of permafrost carbon to climate change, J. Geophys. Res. Biogeosciences, 119, 1630–1651, https://doi.org/10.1002/2014JG002688, 2014.

Karlen, W and Matthews, J.A.: Reconstructing Holocene Glacier Variations from Glacial Lake Sediments: Studies from Nordvestlandet and Jostedalsbreen-Jotunheimen, Southern Norway, Geografiska Annaler. Series A, Physical Geography, 74, 327–348, 1992.

Kilian, R., Baeza, O., Breuer, S., Ríos, F., Arz, H., Lamy, F., Wirtz, J., Baque, D., Korf, P., Kremer, K., Ríos, C., Mutschke, E., Simon, M., De Pol-Holz, R., Arevalo, M., Wörner, G., Schneider, C & Casassa, G.: Late Glacial and Holocene Paleogeographical and Paleoecological Evolution of the Seno Skyring 1060 and Otway Fjord Systems in the Magellan Region, Anales Instituto Patagonia (Chile), 41, 5–26, http://dx.doi.org/10.4067/S0718-686X2013000200001, 2013.

Leemann A, Niessen F.: Holocene glacial activity and climatic variations in the Swiss Alps: reconstructing a continuous record from proglacial lake sediments, The Holocene, 4(3): 259-268, doi:10.1177/095968369400400305, 1993.

Lougheed, B. C. and Obrochta, S. P.: A Rapid, Deterministic Age-Depth Modeling Routine for Geological Sequences With Inherent Depth Uncertainty, Paleoceanogr. Paleoclimatology, 34, 122–133, https://doi.org/10.1029/2018PA003457, 2019.

Singh, M., Sinha, R., Tandon, S.K.: Geomorphic connectivity and its application for understanding landscape complexities: a focus on the hydro-geomorphic systems of India, Earth Surf. Process. Landf., 46, 110-130, 2020.

Stuiver, M., Reimer, P. J., and Reimer, R. W.: CALIB 8.2, http://calib.org, 2020.

Van der Bilt, W., Bakke, J., Vasskog, K., D´Andrea, W.J., Bradley, R.S., Ólafsdóttir, S.: Reconstruction of glacier variability from lake sediments reveals dynamic Holocene climate in Svalbard. Quat. Sci. Rev., 126, 15, 201–218. https://doi.org/10.1016/j.quascirev.2015.09.003, 2015.

---

## Author Comment (AC2)

Responses to anonymous reviewer

Dear anonymous reviewer,

Thank you very much for your review of our manuscript entitled "Sediment and carbon accumulation in a glacial lake in Chukotka (Arctic Siberia) during the Late Pleistocene and Holocene: Combining hydroacoustic profiling and down-core analyses". We really appreciate your contribution to helping us to improve our manuscript with helpful suggestions and comments, especially during this difficult time of the ongoing global pandemic. We are happy that you found our diverse, multi-proxy approach interesting and we hope that it will form a useful basis for future studies. In the attached .pdf file, we provide our replies to each individual comment and provide our proposed alterations, changes, and adjustments to the manuscript that will be carried out and shown within the future revised manuscript version. As such your comments are highlighted in black and italicised and our replies are highlighted in blue. We sincerely hope that you are satisfied with our replies and our proposed changes.

Thank you once again for reviewing our manuscript,

On behalf of all the authors,

Stuart Andrew Vyse

_Reviewer comments and author responses_

_- Line 67: Here and later, this unpublished study is cited very often, even in Fig.9. I understand that it is sometimes necessary to relate your own data to data collected in parallel by the working group that are not yet published elsewhere. However, these works are difficult to verify. In any case, these references should be removed, especially if there are other citations in the line. However, I leave the decision whether these references can be left in the article to the editor._

Thank you very much for your comment regarding the five citations of the paper of Jenrich et al. in review. We agree that it is difficult to verify papers that are currently going through the review process. We wanted to cite this upcoming paper due to the lack of comparable studies currently available on the topic of carbon storage within the Arctic Siberian landscape, particularly considering storage within lacustrine sediments. We can however confirm that the paper has since submission of this manuscript, been accepted for publication in Frontiers in Earth Science, and hence citations pertaining to Jenrich et al. will be updated to Jenrich et al. accepted. This also includes the complete reference in line 1045. It must be said, that some values pertaining to the calculated TOCpools and TOC-densities within the paper of Jenrich et al. accepted were slightly updated during the review process of their manuscript and will be updated in our manuscript text as follows:

Line 657:

Old:  TOCpools(Bykovsky lagoons 5.6Mt)and TOC-densities (mean 14.24 kg m$^{-3}$)

New:  TOCpools(Bykovsky lagoons **5.72Mt**) and TOC-densities (mean 15.29 kg m$^{-3}$)

Line 658:

Old:    TOCpools(0.20Mt) and TOC-densities (10.45kg m$^{-3}$) of Polar Fox

New:  TOCpools(**0.23Mt**) and TOC-densities (**12.54kg m$^{-3}$**) of Polar Fox

These values will also be updated within figure 9b.

We are aware that we also cited the paper of Andreev et al. accepted. This paper has also now been published, as such all occurrences of the reference to Andreev et al. accepted will be changed to Andreev et al., 2021 within the revised manuscript version.

*Figure 1: The inlet map in 1a and labels are hard to read and could be a little larger*

Thank you for pointing this out. We will make some changes to figure 1 following suggestions also from reviewer #2 to improve the readability of figure 1 in general. This will include removing the older simplified bathymetric map originally displayed in figure 1c and replacing this with an enlarged lake polygon with plotted hydroacoustic transects. We will also follow your additional suggestions and increase the font size of text within the inlet of figure 1a as well for the comparison sites to further improve the readability of the revised manuscript version.

*Line 184: I think it would be helpful and interesting if the other elements could be presented in the supplement, especially because only a relatively small selection was made at the end for discussion.*

Thank you for this suggestion. We wanted to limit our usage of excessive data within the manuscript due to the already large volume of included proxies that would otherwise lead to a reduction in the readability and interpretability of the presented manuscript as well as an extension to the current length of the manuscript. We agree however that this would be helpful and interesting to include within the supplement and hence we will subsequently include some stratigraphic plots of element data within the revised supplement version.

*Line 189: I don't quite see the advantage of log transformation of the data, especially in terms of comparability with other studies. Please clarify.*

You are correct that many studies do not use log transformation of the data and we take note of your suggestion that for comparability with the majority of other studies dealing with lake systems, it would be unnecessary and not advantageous to display log transformed data here. We hence decide to use the non-log transformed element ratios within the revised manuscript version. As such we will remove lines 188 to 190 from 3.2.3 i.e. "Ratios of element intensities were log transformed using the additive log ratio (ALR) transformation within the package "compositions" (version 1.40) in R (Aitchison, 1984; van den Boogaart et al., 2020; Weltje and Tjallingii, 2008)". We will display the non-log transformed data within the relevant figures and include them within statistical calculations in the revised manuscript version. The log-transformed and non-transformed data do not show significant differences between each other.

*Figure 3: Please enlarge labels and legends*

Thank you for this suggestion. We will enlarge the labels and legends to improve the readability of figure 3 as per your suggestion. We will also carry out some additional changes

to figure 3 as suggested by reviewer #2 which includes the adaptation of the scale so that intervals in each sub-figure are 1 m apart. Moreover, Artifacts that are currently marked in the colour white, will be changed to grey.

*Line 305 – 310: It is really impressive to see such a good age model, which is only made up of bulk ages and depends on low levels of organic matter in the sediment, but reflects an almost continuous and seamless stratigraphy for the last 30,000 years. The authors are discussing sediment mixing or re-deposition of organic material from the catchment area here already, but only for two inverse ages. How can you rule out mixing and rearrangement of older (and/or younger) organic matter from the catchment area for the rest of the stratigraphy? Can we always assume the actual sedimentation age here? I think a little more explanation on this in the discussion chapter would be useful.*

Thank you for your positive comments regarding our age model. We agree with you that dating of Arctic glacial lakes is often very challenging, especially when dealing with sediments of low organic matter content, such as at Lake Rauchuagytgyn.

In practice, it is very difficult to completely rule out mixing and rearrangement of older and/or younger organic matter for other intervals within the stratigraphy.

In addition to the two inverse ages present within LU-I, we could also expect some reworking of catchment organic material within LU-III and the lowermost LU-II units due to the larger scatter observed amongst the ages of dating samples. This may be related to processes associated with the presence of a catchment glacier that could provide a mechanism to rework palaeo-soils and organic-containing catchment sediments. We have partially alluded to this in lines 459 to 461 of section 5.1.1 of the discussion and we will add additional information and references to read as follows: "It must be however noted that the radiocarbon age scatter in LU-III contributes uncertainty to the SRs and MARs derived for this interval as marked by the wider uncertainty band within the presented age model (Fig. 4b). **Age scatter within this unit may be at least in part associated with catchment glacial activity that may lead to the reworking of older catchment organic material with subsequent deposition within the glacio-lacustrine environment (Lunkka et al., 2001)**".

We will also add an extra sentence to dating uncertainty within section 5.2.3 Regional and local controls on carbon accumulation and methodological limitations in lines 716 to 720 to read as follows: "from empirical equations of DBD and carbon content where discrete, volumetric measurements do not exist (Avnimelech et al., 2001; Kastowski et al., 2011) **as well as** varied approaches used for the measurement of sample carbon contents (Elemental analyser vs LOI) (Munroe and Brencher, 2019). **Limitations associated with radiocarbon dating of Arctic glacial lakes due to the absence of appreciable amounts of datable organic material as well as the influence of reworking processes associated with permafrost and glacial processes can also lead to uncertainty with regards to actual sediment ages and hence reconstructed accumulation rates (Abbott & Stafford, 1996; Björck & Wohlfarth 2002).**"

It must however be said, that apart from the two excluded dates within LU-I and scatter within LU-III, much of the rest of the sequence showed a lack of age inversions which may support the reduced influence of re-deposition of organic material from the catchment for most of the sedimentary succession. Moreover, we attempted to account for age uncertainty within this study by including the uncertainty bands for sedimentation rates, mass accumulation rates, and organic carbon accumulation rates to give an estimation of the possible error that might

be induced through age-model uncertainty, that is an advancement upon many studies within this field.

*Later the authors discuss wind-driven shoreline erosion and sediment redistribution during the Holocene as well as heightened availability of catchment sediments by increasing active layer thickness. They also explain the complex morphology of the lake basin, in particular the primary inflow in the south and the associated presence of a large alluvial fan. I don't want to doubt all of that, but I would like to see a little more critical examination of the dating results and the sedimentation history of the lake.*

Thank you for this comment and the suggestion to more critically examine the dating results and sedimentation history. We have responded to the dating results and also considered the implications for accumulation rate estimations in response to the previous comment and added extra information within sections 5.1.1 and 5.2.3 to account for this within the discussion.

With regards to the sedimentation history, we anticipate a complex Holocene sedimentation dynamic that likely reflects a mixture of processes that we believe has been aptly discussed within the current manuscript version. Delving further would be beyond the current scope of this paper, particularly due to the current length and the hybrid aims of this manuscript. Multiple sedimentological studies from lake El´gygytgyn (ca. 150 km away), have suggested a similarly complex intertwinement of processes that we consider here in our discussion in relation to our record. These processes were interpreted to have lead to higher sedimentation rates and coarser grain sizes during interglacial phases when compared with glacial phases. Asikainen et al. 2007 and Francke et al. 2013 suggested that the increased duration of summer ice-free conditions during the Holocene and other interglacials at lake El´gygytgyn played a crucial role in controlling detrital input to the lake basin by regulating wind-driven sediment redistribution by summer storms and aeolian input. During these warmer phases, increased moisture has been implied to have increased fluvial sediment delivery and warmer temperatures to a thickening of the catchment active layer which enhanced the sediment availability at lake El´gygytgyn.

Due to the proximity of both lake systems and a similar proxy response, a similarly complex interaction may be interpreted for lake Rauchuagytgyn during the Holocene during which time, the lakes were likely exposed to similar climate conditions.

*Section 5.2.1 and Line 616: I think in this context that the authors should also briefly discuss the completely different environmental and catchment area conditions of boreal, thermokarst, and glacial lakes.*

Thank you for this suggestion. We do consider these factors in more detail within the section 5.2.3 "Regional and local controls on carbon accumulation and methodological limitations" but we agree with your suggestion and will consider the major environmental and catchment area differences between these lake systems within section 5.2.1. We will thus add additional sentences to line 616 as follows: "As such, limited comparable studies exist and are restricted to studies of Siberian thermokarst lake systems that are generally younger and smaller (Anthony et al., 2014). Comparisons must therefore be **additionally** drawn to Boreal lakes from North America and northern Europe, **as well as to proglacial and bedrock-catchment lakes from Greenland. Significant differences however exist between these lake systems relating to contrasting environmental conditions prevailing at different**

**latitudes as well as high variability with regards to lake and catchment spatial extent and lake water depth and catchment environmental and vegetation properties.**"

We will also add some more simple information regarding the size and water depth of lake systems discussed by the comparison studies within section 5.2.1 in Lines 623 to 637: "The range of Holocene rates is on average lower but generally overlaps with Holocene organic carbon accumulation rates of **small (0.033- 0.73 km2)** Greenlandic lakes (mean 6 g OC m−2 a−1625 ) (Anderson et al., 2009), and to **small (0.022- 0.145 km2)** Uinta glacial lakes, USA (mean 5.4 g OC m−2 a−1) (Munroe & Brencher, 2019). A strong resemblance is also observed when comparing to rates of accumulation calculated for Finnish Boreal lakes that became Ice-free at the Holocene start (Fig. 8, 9a, 9b) (Pajunen, 2000; Kortelainen et al. 2004). The average Holocene and whole core Rauchuagytgyn rates plot well within the range of Finnish lakes and close to the mean of **shallow** Quebec boreal lakes when considering sediment volumes derived from sub-bottom profiling approaches or estimated from core length and lake surface area (Uinta lakes) (Fig. 9a). Recent syntheses of average carbon accumulation rates within European lakes also suggest similar mean accumulation rates ca. 5.6 g OC m−2 a−1 (Kastowski et al., 2011). Pronounced differences exist when compared with **larger** lake systems reported from Alberta, Canada (mean 15 g OC m-2 a-1) (Campbell et al., 2000) and global lakes, reservoirs and peatlands (Dean & Gorham, 1998; Mendonça et al., 2017). Furthermore, average Holocene rates calculated for thermokarst lakes from the 635 Cherskii -Kolyma Tundra, far east Siberia are markedly higher (mean 47 g OC m-2 a-1) than Rauchuagytgyn rates (Anthony et al., 2014)(Fig. 8)."

*Figure 8: Please enlarge the inlet labels. Also, what is the meaning of the red bar for Alberta?*

Thank you for commenting on the small font size of the inlet labels. We will increase the font size of the labels within figure 8 in the revised manuscript version to improve readability. The red bar refers to the average carbon accumulation rate (15 g C m−2 a−1) that was estimated for Alberta lakes by Campbell et al. 2000. Unfortunately only this value was available from the literature for plotting and comparison as the individual values per lake have not been reported in literature sources. This was plotted in a similar fashion within the paper of Munroe and Brencher, 2019 regarding Uintas lakes.

*Line 642: What is the meaning of this sentence. Isn't that just the other way around? What is meant here by inorganic detritus?*

Thank you for your question here. As currently written we agree that this phrasing may be unclear. We meant to convey that the sediment is predominantly organic poor as represented by the very low TOC values and is hence dominated by inorganic sediment. We will now restructure and rephrase line 642 to read as follows: "**The sediment at lake Rauchuagytgyn is predominantly inorganic as represented clearly by the very low total organic carbon content of sediments**".

*Line 685: Please change to "aeolian pathways"*

Sorry for this bad capitalization. We will now alter line 685 to read as follows: "dissolved organic carbon (DOC) via fluvial and/or **aeolian pathways**".

*Supplement:*

*I cannot find Figure S5 and Figure S6 referenced in the text.*

*We apologize for this lack of clarity. We did reference both figures S5 and S6 within line 595 of the original manuscript version that reads as follows: "This is further supported by coarse, sand-dominated surface sediments close to the alluvial fan front (site EN18220) (Fig. 1 &* **Figs S5, S6***)."*

Additional references cited in the authors responses

Lunkka, J.P., Saarnisto, M., Gey, V., Demidov, I., Kiselova, V.: Extent and age of the Last Glacial Maximum in the southeastern sector of the Scandinavian Ice Sheet, Glob. Planet. Change, 31, 407-425, https://doi.org/10.1016/S0921-8181(01)00132-1, 2001.

---

## Author Response (AR1)

**Point-by-point responses to anonymous reviewer**

Dear anonymous reviewer,

Thank you very much for your review of our manuscript entitled "Sediment and carbon accumulation in a glacial lake in Chukotka (Arctic Siberia) during the Late Pleistocene and Holocene: Combining hydroacoustic profiling and down-core analyses". We really appreciate your contribution to helping us to improve our manuscript with helpful suggestions and comments, especially during this difficult time of the ongoing global pandemic. We are happy that you found our diverse, multi-proxy approach interesting and we hope that it will form a useful basis for future studies. In the attached .pdf file, we provide our replies to each individual comment and our alterations, changes, and adjustments that have been carried out within our revised manuscript version. As such your comments are highlighted in black and italicised and our replies are highlighted in blue. Please also observe the changes within the attached mark-up version which shows the manuscript alterations. The line numbers within this document refer to those in the original manuscript version.

We sincerely hope that you are satisfied with our replies and our changes.

Thank you once again for reviewing our manuscript!

On behalf of all the authors,

Stuart Andrew Vyse

**Reviewer comments and author responses**

- Line 67: Here and later, this unpublished study is cited very often, even in Fig.9. I understand that it is sometimes necessary to relate your own data to data collected in parallel by the working group that are not yet published elsewhere. However, these works are difficult to verify. In any case, these references should be removed, especially if there are other citations in the line. However, I leave the decision whether these references can be left in the article to the editor.

Thank you very much for your comment regarding the five citations of the paper of Jenrich et al. in review. We agree that it is difficult to verify papers that are currently going through the review process. We wanted to cite this upcoming paper due to the lack of comparable studies currently available on the topic of carbon storage within the Arctic Siberian landscape, particularly considering storage within lacustrine sediments. We can however confirm that the paper has since submission of the revised manuscript version, been accepted for publication in Frontiers in Earth Science, and hence citations pertaining to Jenrich et al. in review will be updated to Jenrich et al. accepted. This also includes the complete reference in line 1045. It must be said, that some values pertaining to the calculated TOCpools and TOC-densities within the paper of Jenrich et al. accepted were slightly updated during the review process of their manuscript and have been updated in our manuscript text as follows:

Line 657:

Old: TOCpools(Bykovsky lagoons 5.6Mt)and TOC-densities (mean 14.24 kg m-3)

New: TOCpools(Bykovsky lagoons **5.72Mt**) and TOC-densities (mean 15.29 kg m-3)

Line 658:

Old: TOCpools(0.20Mt) and TOC-densities (10.45kg m-3) of Polar Fox

New: TOCpools(**0.23Mt**) and TOC-densities (**12.54kg m**-3) of Polar Fox

These values have also been updated within figure 9b.

We are aware that we also cited the paper of Andreev et al. accepted. This paper has also now been published, as such all occurrences of the reference to Andreev et al. accepted have now been changed to Andreev et al., 2021 within the revised manuscript version.

Figure 1: The inlet map in 1a and labels are hard to read and could be a little larger

Thank you for pointing this out. We have made major changes to figure 1 following suggestions also from reviewer #2 to improve the readability of figure 1 in general. This has included the removal of the older simplified bathymetric map originally displayed in figure 1c which has subsequently been replaced with an enlarged lake polygon with plotted hydroacoustic profile paths. Moreover, we have also added an additional inset to mark the location and form of a glacial cirque within the catchment. We have also followed your additional suggestions and increased the font size of text within the inset of figure 1a to further improve the readability of the revised manuscript version.

Line 184: I think it would be helpful and interesting if the other elements could be presented in the supplement, especially because only a relatively small selection was made at the end for discussion.

Thank you for this suggestion. We wanted to limit our usage of excessive data within the manuscript due to the already large volume of included proxies that would otherwise lead to a reduction in the readability and interpretability of the presented manuscript as well as an extension to the current length of the manuscript. We do agree however that this would be helpful and interesting to include within the supplement. We hence have included stratigraphic plots of elemental data within the revised supplement as supplementary figures 1a (10 kV) and 1b (30 kV). Please be aware, that the labelling of other supplementary figures has now been updated in order to accommodate the inclusion of the element plots within the supplement. A reference to the supplementary figures has been placed within section 3.2.3.

Line 189: I don't quite see the advantage of log transformation of the data, especially in terms of comparability with other studies. Please clarify.

You are correct that many studies do not use log transformation of the data and we take note of your suggestion that for comparability with the vast majority of other studies dealing with geochemical data presented for lake systems, it would be unnecessary and not advantageous to display log transformed data here. We hence now present the non-log transformed element ratios within the revised manuscript version. As such we have removed lines 188 to 190 from 3.2.3 i.e. "Ratios of element intensities were log transformed using the additive log ratio (ALR) transformation within the package "compositions" (version 1.40) in R (Aitchison, 1984; van den Boogaart et al., 2020; Weltje and Tjallingii, 2008)" as well as the

pertaining references. We have now displayed the non-log transformed data within the relevant figures and included them within statistical calculations in the revised manuscript version. Please be aware that we have additionally included the Si/AI ratio within the revised manuscript version as an additional indicator of coarser grain-sizes alongside Zr/K. This has been included within statistics and is referred to within the methods, results and discussion sections.

**Figure 3: Please enlarge labels and legends**

Thank you for this suggestion. We have now enlarged the font size of all labels and legends to improve the readability of figure 3 as per your suggestion. We have also carried out some additional changes to figure 3 as suggested by reviewer #2 which includes the adaptation of the scale so that intervals in sub-figures 3b to 3d are 0.75 m apart. Moreover, interpolation artifacts that were previously marked in the colour white, have been changed to grey. These changes have improved the readability of figure 3.

Line 305 – 310: It is really impressive to see such a good age model, which is only made up of bulk ages and depends on low levels of organic matter in the sediment, but reflects an almost continuous and seamless stratigraphy for the last 30,000 years. The authors are discussing sediment mixing or re-deposition of organic material from the catchment area here already, but only for two inverse ages. How can you rule out mixing and rearrangement of older (and/or younger) organic matter from the catchment area for the rest of the stratigraphy? Can we always assume the actual sedimentation age here? I think a little more explanation on this in the discussion chapter would be useful.

Thank you for your positive comments regarding our age model. We agree with you that dating of Arctic glacial lakes is often very challenging, especially when dealing with sediments of low organic matter content, such as at Lake Rauchuagytgyn.

In practice, it is very difficult to completely rule out mixing and rearrangement of older and/or younger organic matter for other intervals within the stratigraphy.

In addition to the two inverse ages present within LU-I, we could also expect some reworking of catchment organic material within LU-III and the lowermost LU-II units due to the larger scatter observed amongst the ages of dating samples. This may be related to processes associated with the presence of a catchment glacier that could provide a mechanism to rework organic-containing catchment sediments. We have alluded to this in more detail within the revised manuscript version by including extra discussion and references within section 5.1.1 as follows: "It must be however noted that the radiocarbon age scatter in LU-III contributes uncertainty regarding sediment age and hence the SRs and MARs derived for this interval as marked by the wider uncertainty band within the presented age model (Fig. 4b). Such age scatter may be in part associated with catchment glacial activity that could have led to the reworking of older catchment organic containing sediments with subsequent deposition within the glaciolacustrine environment (cf. Lunkka et al., 2001). The usage of bulk sediment radiocarbon dating may additionally exacerbate this problem and add to radiocarbon uncertainty within LU-III as noted by Oswald et al. 2005 and others regarding dating of organic poor, Last Glacial Maximum lacustrine sediments in Alaska and Siberia."

We have also added extra discussion and references regarding dating uncertainty within section 5.2.3 Regional and local controls on carbon accumulation and methodological

limitations in lines 716 to 720 to read as follows: "from empirical equations of DBD and carbon content where discrete, volumetric measurements do not exist (Avnimelech et al., 2001; Kastowski et al., 2011) **as well as** varied approaches used for the measurement of sample carbon contents (Elemental analyser vs LOI) (Munroe and Brencher, 2019). Limitations associated with radiocarbon dating of Arctic glacial lakes due to the absence of appreciable amounts of datable organic material, particularly during glacial phases as well as the reworking of old organic carbon present within catchment sediments that can be brought to the lake basin through permafrost and glacial processes during warm and cold phases respectively, can additionally lead to uncertainty with regards to sediment ages and hence reconstructed accumulation rates (Abbott & Stafford, 1996; Björck & Wohlfarth 2002; Oswald et al., 2005)."

It must however be said, that apart from the two excluded dates within LU-I and scatter within LU-III, much of the rest of the sequence showed a lack of age inversions which may support the reduced influence of re-deposition of organic material from the catchment for most of the sedimentary succession. Moreover, we attempted to account for age uncertainty within this study by including the uncertainty bands for sedimentation rates, mass accumulation rates, and organic carbon accumulation rates to give an estimation of the possible error that might be induced through age-model uncertainty, that is advancement upon many studies within this field.

Later the authors discuss wind-driven shoreline erosion and sediment redistribution during the Holocene as well as heightened availability of catchment sediments by increasing active layer thickness. They also explain the complex morphology of the lake basin, in particular the primary inflow in the south and the associated presence of a large alluvial fan. I don't want to doubt all of that, but I would like to see a little more critical examination of the dating results and the sedimentation history of the lake.

Thank you for this comment and the suggestion to more critically examine the dating results and sedimentation history. We have responded to the dating results and also considered the implications for accumulation rate estimations in response to the previous comment and hence added extra information within sections 5.1.1 and 5.2.3 to account for this within the discussion.

With regards to the sedimentation history, we anticipate a complex Holocene sedimentation dynamic that likely reflects a mixture of processes that we believe has been aptly discussed within the current manuscript version. Delving further would be beyond the current scope of this paper, particularly due to the current length and the hybrid aims of this manuscript. Multiple sedimentological studies from Lake El´gygytgyn (ca. 150 km away), have suggested a similarly complex intertwinement of processes that we have considered here in our discussion in relation to our record. These processes were interpreted to have led to higher sedimentation rates and coarser grain sizes during interglacial phases when compared with glacial phases. Asikainen et al. 2007, Francke et al. 2013 and Vologina et al. 2003 have all suggested that the increased duration of summer ice-free conditions during the Holocene and other interglacials at Lake El´gygytgyn played a crucial role in controlling detrital input to the lake basin by regulating wind-driven sediment redistribution by summer storms and aeolian input. During these warmer phases, increased moisture has been implied to have increased fluvial sediment delivery and warmer temperatures to a thickening of the catchment active layer which enhanced the sediment availability at Lake El´gygytgyn.

Due to the proximity of both lake systems and a similar proxy response, a similarly complex interaction may be interpreted for Lake Rauchuagytgyn during the Holocene during which time, the lakes were likely exposed to similar climate conditions.

We have taken additional account of the complex nature of interacting processes at Lake Rauchuagytgyn regarding the regional and local controls on carbon accumulation in section 5.2.3 by referring to the concept of geomorphic connectivity as suggested by reviewer #2 as follows: **"As geomorphic systems are hierarchical and operate at multiple spatiotemporal scales according to the concept of geomorphic connectivity, the diverse linkages and interrelationships between different catchment components and processes likely plays a large and complex role in regulating sediment and carbon dynamics at Lake Rauchuagytgyn (Singh et al., 2021).**"

Section 5.2.1 and Line 616: I think in this context that the authors should also briefly discuss the completely different environmental and catchment area conditions of boreal, thermokarst, and glacial lakes.

Thank you for this suggestion. We have considered these factors in more detail within the section 5.2.3 "Regional and local controls on carbon accumulation and methodological limitations" but we agree with your suggestion and have considered the major environmental and catchment area differences between these lake systems within section 5.2.1. We have thus added additional discussion to line 616 as follows: " As such, limited comparable studies exist and are restricted to studies of Siberian thermokarst lake systems that are generally younger and smaller (Anthony et al., 2014) and to the much larger Lake Baikal (Martin et al., 1998). Comparisons must therefore be additionally drawn to boreal and glacial lakes from North America and northern Europe, as well as to proglacial and bedrock-catchment lakes from Greenland. Significant differences however exist between these lake systems relating to contrasting environmental conditions prevailing at different latitudes as well as high variability with regards to lake and catchment spatial extent, lake water depth and catchment environmental and vegetation properties."

We have also added some more simple information regarding the size and water depth of lake systems discussed by the comparison studies within section 5.2.1 in Lines 623 to 637: "The range of Holocene rates is on average lower but generally overlaps with Holocene organic carbon accumulation rates of small (0.033- 0.73 km2) Greenlandic lakes (mean 6 g OC m-2 a-1625 ) (Anderson et al., 2009), and to small (0.022- 0.145 km2) Uinta glacial lakes, USA (mean 5.4 g OC m-2 a-1) (Munroe & Brencher, 2019). A strong resemblance is also observed when comparing to rates of accumulation calculated for Finnish Boreal lakes that became Ice-free at the Holocene start (Fig. 8, 9a, 9b) (Pajunen, 2000; Kortelainen et al. 2004). The average Holocene and whole core Rauchuagytgyn rates plot well within the range of Finnish lakes and close to the mean of shallow Quebec boreal lakes when considering sediment volumes derived from sub-bottom profiling approaches or estimated from core length and lake surface area (Uinta lakes) (Fig. 9a). Recent syntheses of average carbon accumulation rates within European lakes also suggest similar mean accumulation rates ca. 5.6 g OC m-2 a-1 (Kastowski et al., 2011). Pronounced differences exist when compared with larger lake systems reported from Alberta, Canada (mean 15 g OC m-2 a-1) (Campbell et al., 2000) and global lakes, reservoirs and peatlands (Dean & Gorham, 1998; Mendonça et al., 2017). Furthermore, average Holocene rates calculated for thermokarst lakes from the 635 Cherskii - Kolyma Tundra, far east Siberia are markedly higher (mean 47 g OC m-2 a-1) than Rauchuagytgyn rates (Anthony et al., 2014)(Fig. 8)."

As mentioned in the previous response we also acknowledge the complexity of interacting factors on a regional and local scale within section 5.2.3 and refer additionally to the concept of geomorphic connectivity suggested by reviewer #2.

**Figure 8: Please enlarge the inlet labels. Also, what is the meaning of the red bar for Alberta?**

Thank you for noticing the small font size of the inset labels. We have subsequently increased the font size of the labels within the inset of figure 8 in the revised manuscript version to improve readability. The red bar refers to the average carbon accumulation rate (15 g C m-2 a-1) that was estimated for Alberta lakes by Campbell et al. 2000. Unfortunately only this value was available from the literature for plotting and comparison as the individual values per lake have not been reported in literature sources. This was plotted in a similar fashion within the paper of Munroe and Brencher, 2019 regarding Uintas lakes. To clarify this, we have added an extra line to the figure caption of figure 8 as follows: "Only the average OCAR across Alberta lakes was provided in Campbell et al. 2000 and hence the average is displayed as a red bar."

Line 642: What is the meaning of this sentence. Isn't that just the other way around? What is meant here by inorganic detritus?

Thank you for your question here. As currently written we agree that this phrasing may be unclear. We meant to convey that the sediment is predominantly organic poor as represented by the very low TOC values and is hence dominated by inorganic sediment. We have now restructured and rephrased line 642 to read as follows: "**The sediment at Lake Rauchuagytgyn is predominantly inorganic as represented clearly by the very low total organic carbon content of sediments**".

Line 685: Please change to "aeolian pathways"

Sorry for this bad capitalization. We have now altered line 685 to read as follows: "dissolved organic carbon (DOC) via fluvial and/or **aeolian pathways**".

Supplement:

**I cannot find Figure S5 and Figure S6 referenced in the text.**

We apologize for this lack of clarity. We did reference both figures S5 and S6 within line 595 of the original manuscript version that reads as follows: "This is further supported by coarse, sand-dominated surface sediments close to the alluvial fan front (site EN18220) (Fig. 1 & **Figs S5, S6**)." Please be aware that we have updated the supplementary figures for the revised version and hence figure labels have been updated within the text and supplement.

**Additional references cited in the authors responses**

Lunkka, J.P., Saarnisto, M., Gey, V., Demidov, I., Kiselova, V.: Extent and age of the Last Glacial Maximum in the southeastern sector of the Scandinavian Ice Sheet, Glob. Planet. Change, 31, 407-425, https://doi.org/10.1016/S0921-8181(01)00132-1, 2001.

**Point-by-point responses to Assoc. Prof. Mgr.Daniel Nývlt**

Dear Assoc. Prof. Mgr. Daniel Nývlt,

Firstly we would like to thank you for taking the time to review our manuscript entitled "Sediment and carbon accumulation in a glacial lake in Chukotka (Arctic Siberia) during the late Pleistocene and Holocene: Combining hydroacoustic profiling and down-core analyses" in detail. We are especially grateful for all your comments and suggestions particularly during this time of the global pandemic. We appreciate highly that you consider the scientific significance and importance of our manuscript for increasing our understanding of carbon storage in Arctic glacial lake systems. We believe this is a significant outcome of this study and we hope that our paper can form the basis of future works in this area. In the following response, we provide detailed replies to each individual comment and provide an overview of changes and adjustments to the manuscript that have been carried out and shown within the revised manuscript version. As such your comments are highlighted in black and *italicised* and our replies are highlighted in blue. Mentioned line numbers refer to those within the original manuscript version. Please see also the mark-up version of the manuscript, where the changes are clearly highlighted.

We hope that are you satisfied with our replies and our alterations.

Thank you again once for taking the time to review manuscript,

On behalf of all the authors,

Stuart Andrew Vyse

**Reviewer comments and author responses**

It should be noted the the MIS1 starts at 14.7 ka in the marine isotope stratigraphy. Therefore your "Mid-to-Late MIS2" should read "Mid MIS2-early MIS1"

Thank you very much for pointing this out. We are sorry for the incorrect usage of the marine isotope stratigraphy. We have now adopted this phrasing throughout the entire revised manuscript version so that it reflects your comment to "**Mid MIS2-early MIS1**". We have also altered the position of the boundary between LU-II and LU-I following your suggestions later within the comments to 346 cm rather than the previous 341 cm that fits better with the Holocene start (11.69 cal. ka BP). The ages have subsequently been updated.

Lakes act also as sinks of atmospheric deposition, which is not necessarily of a material derived from its catchment...

We agree with this and the importance of aeolian deposition that can be derived from further afield than the lake catchment area. We have now accounted for this by rewording and adding in extra an extra reference to this in line 40 so that it reads as follows: "Lakes act as sinks of clastic sediment derived from local catchment weathering processes **as well as from atmospheric deposition** and as such gradually accumulate sediment mass over time (**Dietze et al., 2014**; Hinderer and Einsele, 2001). We have added an appropriate reference

to **Dietze et al., 2014** that considers aeolian processes within lake sediments from the Tibetan plateau.

**important for what?**

Sorry for the poor wording. "Important" has now been replaced with "environmentally sensitive" so that line 55 reads as follows: "The region of Chukotka (Arctic Siberia) represents an **environmentally sensitive area** with limited lacustrine environmental reconstructions (Lozhkin and Anderson, 2013)".

do not use however in two subsequent sentences...

Sorry for this. We have now removed the "however" and subsequent comma from line 71 to avoid double occurrences. Lines 69 to 72 thus now read as follows: "The reconstruction of accumulation rates in these syntheses has however been avoided due to significant reworking of carbon material within permafrost landscapes (Strunk et al., 2020; Windirsch et al., 2020). The role of Arctic Siberian glacial lakes as sediment and carbon sinks has not yet been accounted for."

V-shaped valleys are generally considered fluvial in origin, glacially eroded valleys are described as U-shaped valleys. You should probably better describe this to avoid any confusion. It looks as a U-shaped valley in the Fig 1a.

We agree with this. The valley is in fact U-shaped and not V-shaped as written within the previous manuscript version and thus we accept your suggestion and have changed this to a "U-shaped valley" in line 114. The line now reads as follows: "Lake Rauchuagytgyn (67.7922° N, 168.7312° E) is situated within the glacially eroded **U-shaped**, Rauchua mountain valley...".

...Cretaceous extrusive and intrusive igneous rocks consisting of silicic-intermediate lithologies by andesite...

...to have it in English

Thank you for making this small change here. We accept the suggestion and have changed line 122 to read as follows: "The bedrock surrounding the lake and within the catchment is predominantly composed of cretaceous extrusive and intrusive igneous **rocks** consisting of silicic-intermediate lithologies dominated by Andesite (Zhuravlev and Kazymin, 1999)".

"moraines", rather than "moraine structures"

Thank you for the suggestion. We have now changed it to read "moraines" so that line 124 reads as follows: "Catchment evidence for glaciation includes **moraines** to the north of the lake that denote the maximum extent of glaciation".

...average July and January temperatures of 13 °C and -30 °C, respectively...

We have added in "respectively" so that line 126 reads as follows: "The Arctic continental climate of the area is characterised by mean annual temperatures of -11.8 °C and average July and January temperatures of 13 °C and -30 °C, **respectively** with low annual precipitation of ca. 200 mm (Menne et al., 2012)."

It would be very helpful to add glacial cirques in the Figure 1a

Thank you for this suggestion. We agree that figure 1a would benefit from the addition of some extra information regarding the position of glacial cirques alluded to in section 2. We have subsequently revised figure 1a by adding the position of the most clearly identifiable glacial cirque from satellite data to the map in the revised manuscript version to account for this. We have also added an additional Inset to figure 1a which shows a more detailed image of this glacial cirque that is associated with a small proglacial lake and a typical, steep headwall. In addition, we have added reference again to figure 1a in line 125 where glacial cirques are mentioned. Please also note further changes tofFigure 1 that includes increases in font size to improve readability.

In inset is shown the situation of Lake Rauchuagytgyn (1) compared to other studied regional lakes: (2) Lake Ilirney and (3) Lake El´gygytgyn (ESRI 2020).

We agree with your suggestion and have now altered the caption text to read "In Inset is **shown** the situation of Lake Rauchuagytgyn (1) compared to other studied regional lakes: (2) Lake llirney and (3) Lake El´gygytgyn (ESRI 2020)." We have also added an additional sentence to acknowledge the inclusion of a new inset regarding the position of a glacial cirque. This reads as follows: "A second inset shows the outline of a glacial cirque associated with a small proglacial lake within the eastern catchment."

**Enlarged orthophoto map of the lake ...**

Thank you for suggesting this improved terminology. We have now changed the caption text to integrate this change. The revised text thus now reads as follows: "**Orthophoto map** of the lake and surrounding features."

**Simplified bathymetric map...**

Thank you for the comment. This comment is however no longer relevant as we have decided to remove the older bathymetric map (figure 1c) in response to your subsequent suggestions in the following comments. We have now replaced figure 1c with an overview lake polygon with all hydroacoustic profile paths obtained during the 2018 expedition.

Could you please add the hydroacoustic profiles paths in any of the detail map? I think the map in Fig. 1c could be enlarged in profiles could be included without the detailed relief of the lake surroundings.

Thanks for the comment. We have now modified figure 1c in response to your suggestion. As such we have removed the older bathymetry and the relief of the lake surroundings that was previously presented in figure 1c. In the place of figure 1c, we now show an enlarged lake polygon with the plotted hydroacoustic profile paths obtained during 2018. We have also altered the figure caption to mirror this change to "(c) Lake polygon with all hydroacoustic profile paths retrieved during 2018".

**Why do you present an older bathymetry here when bathymetry is one of your main findings in thsi study. I find this unnecessary.**

This comment has been acknowledged in the responses above. We have subsequently removed this bathymetric map as you are correct that it is unnecessary to have an older bathymetric map presented in this study. We have thus replaced the older bathymetric map with a lake polygon with plotted hydroacoustic profile paths in figure 1c.

**show the profiles paths in the map**

This comment is acknowledged above in previous responses and has been addressed by including a lake polygon with hydroacoustic profile paths in the revised manuscript version as figure 1c.

What do you mean by the basal sediment within the basin? I think they might be the Pleistocene pre-lacustrine sediments of ?glacial, or ?(glacio)fluvial origin - it would be helpful to desribe it better.

Thank you for this comment. We meant the lowermost deposited sediments that comprise the lake bottom substrate within the basin which may have a mixed origin and may have been deposited through fluvial or perhaps glacio-fluvial processes. It must however be said, that the core penetration was very limited into these sediments and it is difficult to say based on the very limited sedimentological data for certain what these sediments are and their age. To acknowledge this comment we have now altered the description to " Further core penetration and retrieval was prevented by **coarse material** comprising of pebble-sized clasts at the core base that most likely represents the **lowermost deposited sediments** within the basin." We have also made changes within the results (section 4.3) and discussion (section 5.1.1) sections to consider this in more detail. This includes the inclusion of more comparison to other basal sediments at other lake locations and the inclusion of additional references.

How large the samples for radiocarbon were from the viewpoint of the core depth? It seems they were sampled as 0.5 cm thick what I guess from the Table 1, but I think this should be also stated here.

We agree with your suggestion. Indeed the samples were taken in 0.5 cm thick slices to retain as high a dating resolution as possible. We have now included a remark to this in the manuscript text of section 3.2.2 to read as follows: "Due to the lack of suitable plant remains and low organic content of the retrieved sediment core, 25 bulk sediment samples **(0.5 cm thickness)** and one surface sample (0–0.5 cm **sediment depth**) were dated for radiocarbon....".

Why these two samples were not used for age-depth modelling? I see the same possible way in deleting the samples 3002 and 3003, or even only the sample 3002. Please explain better what reasons you have for ommiting these two samples. It clearly shows a higher sedimentation rate in this part of the lacustrine succession.

Thank you for your comment and input here. We found during the development of the agedepth model that including the two samples or even just sample AWI - 3001.1.1 would produce a sedimentation rate that was unreasonably high. Considering the nature of the sediment deposited during this interval, our sediment core yields limited sedimentological evidence for a drastically higher sedimentation rate across these depths, such as a turbidite event. Thus, we opted for a model that would - based on sediment characteristics - most likely represent a more realistic sedimentation rate for these depths. Moreover, it has been commonly found amongst Arctic lakes that input of older organic material influences more strongly radiocarbon reliability than younger ages, which supports the exclusion of at least sample AWI - 3002.1.1 from the age-depth modelling (Bronk Ramsey, 2008; Gaglioti et al., 2014; Abbott and Stafford, 1996). In response to your comment, we have included an extra explanation and justification between lines 162 and 165 as follows: "For modelling, we used 23 bulk sediment samples. Two samples (Lab-ID: AWI - 3001.1.1; AWI - 3002.1.1) were slightly older than their successive dates further down. This suggests possible reworking in these depths (81.25 cm and 114.75 cm, respectively) **and would lead to unrealistically high sedimentation rates when included within age-depth modelling that is not mirrored by sedimentological proxies**. We thus treated these two dates as outliers and excluded them from the modelling process."

**Mn and Fe are also rock-forming elements...**

Thank you for pointing this out. We have now restructured lines 183 and 184 to read more efficiently as follows: "The main rock-forming (Aluminium (Al), Silicon (Si), Calcium (Ca), Potassium (K), Titanium (Ti), Manganese (Mn), Iron (Fe), Rubidium (Rb), Strontium (Sr), Zircon (Zr)) and productivity (Bromine (Br)) linked elements were selected for further processing".

**?element proportions**

We have now removed this sentence and hence this comment is no longer applicable. Based on suggestions from reviewer #1 we utilise instead the simple, non-log transformed elemental ratios which show the same patterns as the log-transformed ratios to increase comparability to the vast majority of literature that use this method. Please also note that we have included the Si/Al ratio additionally which is now alluded to within section 3.2.3, as you mentioned it in reference to our discussion section. This has been included within our multivariate statistical analysis and within the text of the results and discussion sections along with additional references to

The boundary of 4 micrometers is used in most sedimentological literature for the clay/silt boundary. The 2 microns boundary is more common in geochemical studies.

Thank you for pointing this out. Indeed Udden (1914) and Wentworth (1922) put the division between clay and silt at 4 micrometers but later studies such as Friedman and Sanders (1978) set the boundary between clay and silt instead at 2 micrometers. As we have used the very commonly (cited 1928 times) used software "Gradistat" from Blott and Pye (2001) to process our grain-size data that utilises the Friedman and Sanders (1978) boundary at 2 micrometers, we have opted to retain our clay/silt boundary here. We have added an additional citation to the grain-size classification of Friedman and Sanders (1978) to line 202 to justify our usage of this grain-size boundary so that it reads as follows: "Intervals of 2 mm– 63  $\mu$ m, 63–2  $\mu$ m and <2  $\mu$ m, were used to define percentages of sand, silt, and clay respectively (**Friedman and Sanders, 1978**)".

**The Folk and Ward method ... upper case in names**

Sorry for this. We have now capitalised these letters here. Line 202 now reads as follows: "The Folk and Ward method was used for mean grain-size calculation." We have also added additionally the citation to the original paper of Folk and Ward, 1957 to the end of the line.

et al.

Thank you for noticing this. It has now been corrected for in the revised manuscript version to "Avnimelech **et al.** (2001)". We have also checked throughout the manuscript for similar

occurrences affecting references, which have subsequently been corrected for where needed.

Do not start a sentence with a number. e.g.: Subsequently, 65 dried and milled...

We agree with your comment here and have changed line 214 to read as follows: "**Subsequently**, 65 dried and milled.......".

delete 3x the commas after al.

Thank you for seeing this. The three commas have been deleted after the al. in line 245.

Is the term shelf correct when referring to the lake? I would prefer the describe it as submerged paleoterrace.

Thank you for this comment. The term "shelf" has been used within published literature on seismic and hydroacoustic methods to describe similar morphological features at other Arctic glacial lake sites. Examples may be observed in Lebas et al. 2019 "Seismic stratigraphical record of Lake Levinson-Lessing, Taymyr Peninsula: evidence for ice-sheet dynamics and lake-level fluctuations since the Early Weichselian" a concrete example from the paper reads as follows: "The bathymetric map of Lake Levinson-Lessing (Fig. 3) allows the main morphological characteristics of the lake basin to be distinguished. Four major morphological elements characterize the lake floor: (i) a large, deep central depression with water depths of >100m, which covers most of the lake, (ii) a small sub-basin, located north of the central depression, (iii) a northwestem and (iv) a southern **shelf".** Based on the similar morphology and description of the "shelves" at Lake Levinson-Lessing and Lake Rauchuagytgyn, we opt to retain the term "shelf" here within our revised manuscript version.

Please unify the use of Birtish vs. American English. The text is written in American English, but in the Figure appear "Palaeoterrace", which is written in British English.

Sorry for the mixed usage of British and American English within this manuscript. We have now thoroughly checked the manuscript for discrepancies existing between British and American English. All corrections have been made and are noted within the mark-up version of the manuscript. As such, the revised manuscript version is now written completely in British English.

**northern-shelf - see my comment above**

This comment is addressed in a previous response. As such we have retained the usage of "shelf" as it has been used to describe similar morphological features within other Arctic glacial lake systems (see Lebas et al. 2019 for further concrete examples).

**shelf - see my comment above**

This comment is addressed in a previous response. As such we have retained the usage of "shelf" as it has been used to describe similar morphological features within other Arctic glacial lake systems (see Lebas et al. 2019 for further concrete examples).

I would avoid using the term hummocks here, as hummocky-cross stratification/bedding is a sedimentary feature, which leads to a sediment morphology, rather than to an erosional landform as is the case here. Use evelations and depressions as not genetically-bounded terms here.

Thank you for this suggestion. We agree that this could be a confusing. We have instead opted to use the proposed terminology of "elevations and depressions" that you suggest. Lines 297-298 will thus read as follows: "AU2 possesses a volume of ca. 18055352 m3 (0.018 km3) with complex internal architecture with **elevation and depression-like** structures seen within some areas."

kilometers - either "km", or "kilometers" (lower case initial) - all parts of the figure

Thank you for pointing this out. We have now corrected all the upper case initials to lower case initials throughout figure 3 within the revised manuscript version. We have also increased the font size throughout the figures to enhance figure readability.

The same thickness scale in Figs 3b, c, d would help to visually see the differences. Now the pink is 6-7 m in 3b, 8-9,5 m in 3c and 13-15 m in 3d - this is not good for visual comparison.

Thank you for this hint to improve the readability of figure 3. We have accepted your suggestions and have adopted this in the new revised version of figure 3. We have decided to use an interval of **0.75 m** for all figures 3b to 3d to improve the visual comparison between the plots.

the artifacts are coloured in white, which is also the colour for the most thick sequence in Fig. 3c and 3b. I think using grey colour for the gaps (artifacts) would be better, as this colour is out of the scale used.

Thank you for noticing this. We agree that this may be little bit confusing within the figures and thus we agree with your suggestion and have changed the colour of the artifacts to grey to account for this in the revised manuscript version. This has been further noted within the respective legends of figures 3b to 3d. Font size of text throughout figure 3 has been increased additionally to improve readability.

these figures show that the delta at the inflow is rather of Pleistocene origin, but how you know that it is of Pleistocene age as both parts (Pleistocene and Holocene) were not delimited in this part of the lake basin as shown in Fig. 2b.

Thank you for this comment. It is difficult from the presented data to know the true age of the delta and hence we do not assign a strict age to the main inflow delta within our manuscript. We only know that processes have been operating at the delta during the Holocene due to the observance of active fluvial channels and coarser-grained surface sediments proximal to the delta and alluvial fan that evidence input of fluvial material. What would certainly help us to know more information about the age of the delta would be reduced acoustic blanking in this area and perhaps the usage of another geophysical supporting method such as seismic stratigraphy (see Lebas et al. 2019 for further details) and the retrieval of cores through the delta sediments. Unfortunately, acoustic blanking, no current possibility for additional seismic data, and no additional cores from the delta region of the lake, hinder further interpretation as to the age of the delta here.

**shelf - see my comment above**

This comment has been addressed in previous responses. As such we have retained the usage of shelf as it has been used to describe similar features within other Arctic glacial lake systems, such as Lake Levinson-Lessing (see Lebas et al., 2019 for concrete examples for this usage).

Is the age of 29 cal. ka BP modelled, or calibrated - I see this is a calibrated age of the lowermost sample - why do you mention modelling here? And how does Elias and Brigham-Grette, 2013 refer to this information? This is simply an inappropriate reference here, as the age is part of your results! Besides, references should not be used in the Results chapter, as it should contain your results and no reference are needed here! Some journals even prohibit the use of references in the Results chapter.

Thank you for pointing out this lack of clarity. It is common in the age-depth modelling domain that both calibrated radiocarbon samples and modelled ages are expressed as "calibrated years BP", in our case "cal. ka BP". Age determination samples are calibrated internally in the age-depth modelling software and hence the depth in between those samples are regarded as calibrated as well.

As stated in the current manuscript version we used the age-depth modelling software "Undatable" (Lougheed & Obrochta, 2019). In this software the lowermost point in the age-depth model does correspond with lowermost radiocarbon sample we took. Hence, we say "modelled" to emphasize that we derived the value from the age-depth modelling software instead of a separate calibration program, i.e. CALIB 8.2 (Stuiver et al., 2020).

Sorry for the inappropriate references here. We agree that results should not contain references wherever possible. We have subsequently moved the references (Abbott & Stafford, 1996; Björck & Wohlfarth 2002) to the discussion section 5.2.3 where we have added some extra lines from lines 716 to 720 to consider possible reworking that can affect radiocarbon dating to read as follows: "from empirical equations of DBD and carbon content where discrete, volumetric measurements do not exist (Avnimelech et al., 2001; Kastowski et al., 2011) as well as varied approaches used for the measurement of sample carbon contents (Elemental analyser vs LOI) (Munroe and Brencher, 2019). Limitations associated with radiocarbon dating of Arctic glacial lakes due to the absence of appreciable amounts of datable organic material, particularly during glacial phases as well as the reworking of old organic carbon present within catchment sediments that can be brought to the lake basin through permafrost and glacial processes during warm and cold phases respectively, can additionally lead to uncertainty with regards to sediment ages and hence reconstructed accumulation rates (Abbott & Stafford, 1996; Björck & Wohlfarth 2002; Oswald et al., 2005)." We have also removed the reference to Elias and Brigham-Grette, 2013 from line 306 and also check for further uses of references in the results chapter that will be subsequently removed.

**See my comment above regarding the two samples omitted from the age-depth model calculation.**

Thank you for the comment. We have addressed this comment in a previous response. We excluded these samples on the basis of an older age of sample 3002 relative to lower samples that suggested possible reworking and input of old carbon that we have now acknowledged within section 5.2.3 of the discussion. Moreover, the inclusion of samples 3002 and 3001 within age-depth modelling would produce unrealistically high sedimentation rates that would not be reconcilable from the sedimentological data that shows no evidence of a drastic increase in sedimentation rate in these depths. We have thus made changes to lines 162 to 165 to account for your suggestions.

Isn't the larger age scatter in Pleistocene samples (and age model), when compared with Holocene ages, connected with a larger age scatter after radiocarbon age calibration, because of larger uncertainties and less data for calibration curve calculation?

Thank you for your question. We are using the newest published calibration curve (IntCal 2020) in this study that was published by Reimer et al. 2020 and hence the calibration curve represents the most recent, state-of-the-art dataset with reduced uncertainties compared with previous calibration curves. Moreover, the uncalibrated years already show scatter and therefore the calibration curve likely has little influence. It is more likely that the scatter is related to processes possibly related to the presence of a catchment glacier during this interval that could have led to reworking of catchment organic material and redeposition within the lake basin. We have now alluded to this in more detail within section 5.1.1 as follows: "It must be however noted that the radiocarbon age scatter in LU-III contributes uncertainty regarding sediment age and hence the SRs and MARs derived for this interval as marked by the wider uncertainty band within the presented age model (Fig. 4b). Such age scatter may be in part associated with catchment glacial activity that could have led to the reworking of older catchment organic containing sediments with subsequent deposition within the glaciolacustrine environment (cf. Lunkka et al., 2001). The usage of bulk sediment radiocarbon dating may additionally exacerbate this problem and add to radiocarbon uncertainty within LU-III as noted by Oswald et al. 2005 and others regarding dating of organic poor, Last Glacial Maximum lacustrine sediments in Alaska and Siberia." Moreover we have also considered this within section 5.2.3 when considering potential limitations as mentioned within a previous response. We have attempted to take account of scatter within our manuscript by including sigma ranges in sedimentation rate calculations.

a comma before but - "...low rates, but with..."

Thanks for pointing this grammatical mistake out. Line 315 has now been corrected to the following "demonstrates **low rates, but with** larger uncertainty".

"...uncertainty ranges (dark and light grey ribbons)." 1sigma is dark grey 2sigma is lighter grey

Thank you for noticing this within the figure caption. We have adopted your change so that the caption of figure 4 reads as follows: "1 and  $2\sigma$  uncertainty ranges (**dark and light** grey ribbons)."

What do you mean by: Br/Al ration values demostrate their lowest values at any depth? It clearly has the lowest values in LU-III, not at any depth...

Sorry for this confusing sentence structure here. We agree with your suggestion and have now changed line 336 and 337 to read as follows: "Br/Al ratio values demonstrate their lowest values within LU-III alongside TC.......".

"cal. ka BP" - lowercase cal.

Thank you for noticing this. We have now corrected to the lowercase **cal.** as you suggested. We have also checked occurrences throughout the manuscript and supplement to ensure "cal." is written in lowercase.

The Grain-size box should be logically made from the left by clay, then silt in the middle and sand to the right - it is very erroneous to put sand between clay and silt. Why the terms initiate with upper cases? - this is inappropriate here.

We agree with your comment and agree that the current presentation is not logical. We have subsequently followed your suggestion and altered figure 5 to show clay, silt, and sand in that order as suggested. The starting letters of the terms displayed in figure 5 have also been altered to lower case as suggested. The boundary between LU-II and LU-I has also been shifted within the figure to 346 cm from 341 cm as per your suggestions in the following responses.

What do you mean by high grain-size? Is a coarse grain-size?

Sorry for the poor wording. Here we meant "coarser mean grain-size". We have subsequently altered the figure caption text of figure 5 to the following "mean grain-size plot refers to three excluded data points (650.5, 341, 321 cm) of **coarser mean grain-size** (up to  $25 \mu$ m)."

**"cal. ka BP" - lowercase cal.**

Thank you for noticing this. We have corrected the lowercase cal. in figure 6 as suggested as was also the case for figure 5. Again, we have also checked this throughout the manuscript and supplement and ensured that cal. is everywhere written in lower case.

Why is the TOC displeyed twice? I do not see the need to show it in wt% and retain only the g OC cm-3

Thank you for the comment. We agree with you that we do not need to present TOC twice within figure 6. We decided in this instance to remove the TOC curve in g OC cm-3 as the wt% curve displays both the TC and TOC curves and is more traditionally presented within palaeolimnological studies. This has improved the readability of figure 6.

Basing on all graphs I have seen, I would put the boundary between LU-II and LU-I slightly lower!!! It is impossible to put the boundary at the peak of TOC (and some toher proxies), I strongly recommend to put this boundary slightly lower to have the peak already in LU-I, not at the boundary and it will probably also fit better with the beginning of the Holocene (11.7 ka b2k, rather than your 11.5 ka BP). I would put the boundary in the mid-point of the PC1 score rapis increase, if it fits with lithological data. Please think about this change!

We thank you for this important suggestion and we agree with it. We have subsequently moved the LU-II and LU-I boundary to the suggested mid-point of the rapid increase in PC1 scores at a depth of **346** cm that actually corresponds to an age of **11.69** cal. ka BP and hence more effectively represents the Holocene start. We have corrected this boundary in all relevant figures and text passages within the manuscript and in the supplement that show and/or refer to this boundary to accommodate for this change throughout the manuscript. Thank you again for making this suggestion.

sand, silt, clay, mean GS - lower cases - both in text and in Figure 7

Sorry for the capitalisation mistake in figure 7 and in the text. We have now changed the first letters to lower case as per your comment in both the text in line 388 and in figure 7.

What is Early MIS 2, Late glacial and Holocene and how it relates to LU-I, LU-II, or LU-III units? Please explain it better in figure caption.

Thank you for this comment. We have subsequently changed the terms within figure 7 to LU-I, LU-II, and LU-III as the grouping is based on the lithological unit definition. We have modified the caption of figure 7 to read as follows: "Principal component analysis (PCA) biplot of sedimentological, biogeochemical and accumulation rate data from core EN18218. **Samples are coloured and clustered according to their lithological unit definitions (LU-III, LU-II, LU-I).**". We have also added an additional sentence regarding the stratigraphic plot of the PC1 scores within the caption that was not mentioned in the first manuscript version to read as follows: "Also shown are the PC1 scores plotted stratigraphically in relation t o depth and modelled age."

palaeo - British vs. American English

Sorry for this. As we have mentioned previously within our responses, we have now checked the use of British vs American English throughout the manuscript. The manuscript is now written it British English for its entirety.

This is incorrect use of a reference. Mangerun and Svendsen worked on Svalbard - how this relates to Chukotka? You should rather use here "(cf. Mangerud & Svansen, 1990)."

Sorry for this reference error. We have subsequently removed the reference here from section 5.1.1 as part of more substantial changes included with the revised version of this manuscript.

rather "basal sediments" than "basement sediments"

Thank you for this suggestion. We now refer to "basal" sediments within the revised manuscript version.

Again, the term "basement" is mostly used for solid rocks, rather than for sediments by geologists.

Ok, we refer instead to "basal structures" in the revised manuscript version.

again - (cf. Lebas et al, 2019; Lebas et al., 2021).

Thank you again for pointing this out. We agree with your comment and now use (**cf.** Lebas et al., 2019; Lebas et al., 2021) for the references.

Could you explain how can a glacigenic sediment be layered? Glacigenic sediments are those deposited directly from glacier without subsequent sorting, which means tills and tills are hardly layered. But glaciofluvial and glaciolacustrine sediments could be layered. Please make this clear.

Sorry, we understand that the portions of section 5.1.1 were possibly unclear as previously written. We have subsequently made significant adjustments to and clarifications within this section in order to address your comments here that can be seen most clearly within the mark-up version of the manuscript.

Firstly, we refer now to the coarse basal sediment with pebble-sized clasts that was found directly at the core base and likely below and prevented further core retrieval as "diamicton".

The very limited sedimentological data available that resulted from the lack of core penetration due to blockage of the drilling platform with a pebble does not allow a comprehensive interpretation and hence we have made this clear now within the text. The limited information we do have suggests some similarities to the coarse basal diamicton inferred for Harding Lake in Alaska that represented lake bottom substrate present at the base of unit 1 inferred by Finkenbinder et al. 2014.

For the overlying, fine-grained, minerogenic, layered sediments prevalent throughout the unit LU-III (see additional figure below for an example of layering from unit LU-III), we no longer refer to them as "alacigenic" within the revised manuscript version. We agree that the use of this term may be misleading as currently written when used in this context. The characteristics of these sediments most likely instead relates to sedimentation under glaciolacustrine conditions with input of generally fine-grained, minerogenic suspended load material derived from catchment glacial activity. Our interpretation of this unit is based on similar published findings at Harding Lake presented by Finkenbinder et al. (2014) as well as well-studied glaciolacustrine sediments deposited in Lake Silvaplana by Leemann and Niessen, 1994b. At Harding Lake, following deposition of the basal diamicton prior to ca. 30.7 cal. ka BP at the base of unit 1, sedimentation from ca. 30.7 cal. ka BP to 15 cal. ka BP throughout the rest of unit 1 consisted of finer-grained deposition of minerogenic sediment of high magnetic susceptibility, high and variable dry bulk density (of remarkably similar values represented for much of LU-III at Lake Rauchuagytgyn), and an abundance of the element Titanium. These sediments were interpreted to represent lacustrine sediments deposited continuously within a perenially ice-covered lake basin with enhanced minerogenic sediment input during the global last glacial maximum. Moreover, sedimentation rates within unit 1 at Harding Lake displayed comparable values to those calculated for LU-III at Lake Rauchuagytgyn of ca. 0.01 cm/yr.

Example of layering within LU-III.

In response to sediments within LU-III being layered, studies of glacially influenced layered glaciolacustrine sediments from Lake Silvaplana by Leemann and Niessen, 1994b have shown that catchment glaciation can lead to the synchronous deposition of fine-grained, minerogenic (clay & silt) suspension load within layers. This may be a similar mechanism to that acting at Lake Rauchuagytgyn during this interval. Karlen and Matthews (1992) have similarly reported the occurrence of silt/clay bands found within glaciolacustrine sediments from southern Norway that possessed low organic content with high magnetic susceptibility that likely represent minerogenic sediment bands associated with catchment glaciation. Moreover, van der Bilt et al. 2015 investigated sediments of high bulk density, magnetic

susceptibility, and titanium concentration represented glacial suspended load that could be used to identify periods of glacial advance. Sediments that were associated with glaciallyderived suspended load for example in unit 1 of van der Bilt et al. (2015), demonstrated layering in the form of centimeter scale lamination.

Based on these literature comparisons, it is our interpretation that the fine-grained sediments deposited throughout the majority of LU-III likely represent glaciolacustrine sediments that were influenced by the input of minerogenic glacially-derived suspended load in the form of silt and clay from a catchment glacier and were additionally impacted also by the presence of a quasi-permanent lake ice layer similar to Alaska lacustrine sediments deposited at Harding Lake and elsewhere (Finkenbinder et al., 2014).

In the revised manuscript version, we have thus made these interpretations much clearer and refer to "glaciolacustrine" deposition as well as "glacially-derived minerogenic suspended load". We have integrated additional references to Finkenbinder et al. 2014 and Leemann and Niessen, 1994b to support our interpretations along with some more logical restructuring to account for your comments that can be viewed within the mark-up version.

How do you know that these are glacigenic sediments? Sediments and their facies should be described based on their texture and structure, not based on colour, elemental composition, or PCA. You should be more cautious when using sedimentological term without knowing the sedimentology of the material!

Thank you very much for your comment here, we have considered this comment in some detail within the previous response. We meant to refer to fine-grained, minerogenic sediments that have been derived through glacial processes acting within the lake catchment that were deposited within a glaciolacustrine setting. In general, we originally meant to use the term "glacigenic" to refer to minerogenic glacial suspended load deposited within a glaciolacustrine setting which is generally dominated by clay and silt and characterised by a high bulk density as has been described by Leeman and Niessen, 1994b, Bakke et al. (2005) and van der Bilt et al. (2015). Van der Bilt et al. (2015) for example, referred to glacigenic sediment when discussing the input of clay and silt recorded by lacustrine sediment cores in Svalbard. Moreover, Van der Bilt et al. (2015) also used multivariate statistical approaches (PCA) to effectively "fingerprint" the sedimentological and geochemical nature of glacigenic sediment in a similar manner to that what we have carried out in this manuscript. As mentioned in the previous response, studies of glaciolacustrine sediments from southern Norway have also suggested that sediment bands of silt/clay composition with high magnetic susceptibility represent glacigenic sediment deposited within a lacustrine environment (Karlen and Matthews, 1992).

As we have mentioned in the previous response, we now refer to fine-grained, minerogenic suspended load sediment that was derived through catchment glacial processes rather than referring to glacigenic sediment as per your comments in the revised manuscript version.

again, Van der Bilt et al., 2015 did not work on Chukotka, so "cf. Van der Bilt et al., 2015" would be better to refer to.

Thank you for pointing this out. We now refer to (cf. Van der Bilt et al., 2015).

The principal question is - was there a lake at that time? Probably not, as you wrote about glacigenic sediments. So, why you write about lake here? It is all about the sound interpretation of individual sedimentary facies described in the section, without that it is very hard to follow your story!

Sorry for this lack of clarity. We have now carried out changes to this section to make our story clearer as mentioned in the previous responses. It is our opinion that apart from the coarse-grained basal sediments, that LU-III predominantly represents glaciolacustrine sedimentation that was influenced by the input of minerogenic, glacially-derived suspended load from a catchment glacier. The lake was likely also covered with a quasi-permanent surface ice layer that acted to reduce sediment input and shares similarities to continuous lacustrine deposition that was interpreted for unit 1 above the basal diamicton at Harding Lake in Alaska (Finkenbinder et al., 2014).

Which of the two Kokorowski et al. 2008 this is? They should be referred to as Kokorowski et al. 2008a and 2008 b both in the text and in the list of references.

Sorry for this referencing mistake. We meant to refer to Kokorowski et al. 2008**a** in this case. We have now corrected the reference to Kokorowski et al. 2008**a** and the correct **a** and **b** notation is added throughout the manuscript. We have also updated the references of both Kokorowski et al. 2008 papers within the references section in lines 1063 and 1066.

**?or in line with non-lacustrine environment?**

Thank you for this comment. We have addressed this comment in previous responses addressing section the upper part of section 5.1.1. We envisage a glaciolacustrine environment for much of LU-III.

again - "cf. Baumer et al., 2020;..."

Ok, thank you for pointing this out. We will now alter the references here to **cf.** i.e. (**cf.** Baumer et al., 2020; Biskaborn et al., 2019; Fritz et al., 2018; Heinecke et al., 2017; Naeher et al., 2013).

Is really Zr/K a proxy to coarse-grained lithology? What do you mean by "coarse" here? For most geoscientists coarse would mean gravel. Generally, all facies described in this study are rather fine-grained. Therefore, you might explain it better here. Generally, the Zr/K index increases with an increase of coarse silt to medium sand fraction, what we found in numerous our studies from Central Europe, Svalbard, Greenland, or Antarctic Peninsula. For detection of even coarses grains (fine sand to fine gravel), the Si/Al index is used, as quartz is a predominant in sand to fine gravel fraction and aluminosilicates are common in clay to medium silt fractions.

Thank you for this comment. We agree that the sediment deposited within the lake is generally fine-grained in nature and should not be referred to as "coarse". In the revised manuscript we have been more cautious with the application of "coarse" and have subsequently altered lines 425 and 426 to read as follows: "Finer grain-sizes directly measured by laser diffraction are supported by indirect, XRF-derived grain-size proxies for finer, clay-dominated sediment (K/Ti) and of coarser sediment (Zr/K, Si/AI)" In our context, we mean to use both the Zr/K and Si/AI ratios to represent a general coarse silts and sand fractions to the grain-size signal during the Holocene. The Zr/K ratio as

you said is likely related to the increase in the proportions of coarse silt and very coarse silt and some sand fractions that is observed within Holocene sediments. Equally, the Si/Al ratio increases in a similar manner with coarsening of the fine-grained sediment and hence may also be related to grain-size increases and increased contribution of coarser sediment. Si/Al may however relate additionally to biogenic sediment due to the occurrence of diatoms within the lake sediments – we allude to this later within section 5.1.3 with the addition of an extra reference to Procházka et al. 2019.

**Harding Lake**

Sorry for this spelling mistake in line 436. We have followed your suggestion and change to "Harding Lake". We have also checked for further misspellings of "Lake" throughout the manuscript and corrected for them in the revised manuscript version.

Is really 0.15 mm a-1 a low sedimentation rates? It equals to 15 cm/ka, which is higher that what is described above. It would be helpful to show comparison to Holocene sedimentary rates in Tajikistan.

Thank you for pointing this out. Sorry, we originally meant to say lower Pleistocene sedimentation rates in comparison to the Holocene at Lake Karakul. We have subsequently altered the phrasing of these lines to make this distinction clearer and also added in the Holocene sedimentation rate for comparison as suggested. Line 440 will now read as follows: "A similar finding at lake Karakul, Tajikistan of **lower** sedimentation rates during MIS2 **since ca. 29 cal. ka BP** (0.15 mm a-1) **when compared with Holocene sedimentation rates (0.84 mm a-1)** was also explained by reduced sediment input during MIS2 compared to the Holocene alongside reduced organic matter accumulation (Heinecke et al., 2017)".

We have also added in additional reference to the similar sedimentation rates recorded at Harding Lake during the deposition of minerogenic sediment within unit 1, which is comparable with sedimentation at Lake Rauchuagytgyn within LU-III. "Sedimentation rates reported for Lake El'gygytgyn crater lake (Fig. 1) were low during MIS2 (4.8 cm/ka), compared to higher rates (7.6 cm/ka) during the Holocene (Nowaczyk et al., 2007). A similar finding was made for Harding Lake in Alaska whereby sedimentation rates d uring the Last Glacial Maximum (30.7 to 15.7 ka) were generally < 0.1 mm/yr but temporally continuous throughout deposition (Finkenbinder et al., 2014)."

**Rauchuagytgyn**

Sorry for the spelling mistake. This has now been corrected in line 442 and has been checked for correctness throughout the rest of the manuscript.

**Late Glacial**

Sorry for the spelling mistake here too. It has now been changed in line 443 and also corrected at other locations throughout the manuscript.

**Last Glacial Maximum**

We apologize again for the spelling mistake here. It has subsequently been changed in the revised manuscript version in line 446 and checked throughout the manuscript.

"glacial erosion", rather than "glacial denudation"

Thank you for this suggestion. We agree and have now changed "denudation" in line 449 to "erosion" in the revised manuscript.

When was the glacier 25 km long? This is important to describe it here. If it was during the LGM than the early MIS2 facies are very probably glacigenic and not lacustrine. This is what is not well described and proven in the entire manuscript.

Thank you for this comment. According to Glushkova, 2011, the glacier was likely a length of ca. 25 km and was suggested to have been so during the Sartan glaciation which is synonymous with glaciation during MIS2. This was somewhat of a relative approach within the Rauchua valley, as no absolute dating methodologies were utilized and up to now, have not been available. Some dating methods, for example using cosmogenic nuclides on glacial features in the catchment, may provide some additional information regarding the timing, but are unfortunately currently not available for the Rauchua river valley. This may however be a goal of future work within the region. We have now added some mention to this uncertainty within section 5.1.1 when considering limitations as follows: "The lack of absolute dating of glacial structures within the Rauchua valley by Glushkova, 2011, adds additional uncertainty regarding the temporal dynamics of the catchment glacier." We have also added further information in the revised manuscript version so that lines 454 to 457 read as follows: "Remote sensing based studies of Chukotkan glacial geomorphology and structures within the Rauchua valley have suggested the presence of a passive catchment glacier, ca. 25 km in length that extended along the length of the Rauchua river valley and discharged into the Rauchuagytgyn basin during marine isotope stage 2 (Glushkova, 2011)." We have addressed the rest of the comments regarding glacigenic (now altered) input to the lake basin in previous responses. We consider sediments deposited within LU-III to be generally of glaciolacustrine origin with input of minerogenic suspended load from the catchment glacier.

...not contributed significant sediment volume to the lake...

Thank you for this suggestion. We have now altered lines 457 to 459 to read as follows: "Thus, the catchment glacier may have been predominantly non-erosive during the early MIS2 and hence **not contributed significant sediment volume** to the lake basin supporting the low rates **of sediment and mass accumulation** discussed here (Gurnell et al., 1996)."

**See my comment in the abstract for the timing of MIS2**

Thank you. We have noted your comment within the abstract and have agreed with your suggestion and have subsequently altered this throughout the manuscript to "**Mid MIS2-early MIS1**". The discussion title in line 472 has consequently now been altered to "5.1.2 **Mid MIS2-early MIS1** accumulation during progressive climate amelioration (ca. 23.4–**11.69** cal. ka BP)". We have also checked for other occurrences within the manuscript and adjusted accordingly.

(cf. Lebas et al., 2021;...)

Sorry for this. We have now corrected the reference to (cf. Lebas et al., 2021) in line 477.

I do not agree completely with this! Alexis Dreimanis made already 80 years ago pioneering studies (summarised e.g. in Dreimanis and Vagners 1971 In: Goldthwait RP, ed.: Till, a

Symposium, Ohio State University Press) of what terminal grade (grain-size) are produced by glacial grinding and milling and found for most minerals that the final granolometry is in fine to middle silt fraction. Clay-sized grains are assumed to be sourced from clay minerals within the bedrock only (Haldorsen 1983 Norsk Geologisk Tidsskrift), but in principle most of the glacially derived material lies in silt fraction (e.g. Haldorsen 1981 Boreas). This means that rock flour is principally silt-sized. Besides, when applied 2 microns as the boundary between clay and silt then most of the glacially grinded and milled terminal grades would terminate in silt fraction, i.e. >2 micrometers.

Thank you for your comment. We appreciate your input on the interpretation here and we recognise that these studies have suggested that rock flour is principally silt-sized. Despite this, it has been recognised more recently from the exceptional, ca. 155 ka Owens Lake record, that glacially produced rock-flour can be found within the clay-size fraction (Bischoff et al. 1997). To account for your comment, we have now removed reference to rock-flour and altered lines 477 to 480 to read as follows: "A clay maximum and grain size minimum at ca. 550 cm (ca. 22.7 cal. ka BP) may suggest initial increases in lake water-depth through glacial melt additions that could have led to the observed reduced SRs and MARs this time. This may be supported by the high values of K/Ti (clay contribution) and low values of Zr/K and Si/Al (proxies for coarser grain-sizes) (**Kilian et al., 2013**; Kříbek et al., 2017; Cuven et al., 2010)." We have also included an additional reference of K/Ti as a proxy for clay contribution (**Kilian et al., 2013**).

This may be supported ...

Thank you, we have now changed this in line 480 in the revised manuscript version.

...controlled by inflowing rivers...

Thank you for noticing. We have now altered this in line 490 in the revised manuscript.

Late Glacial

Sorry for the misspelling. This has been corrected in line 506 in the revised manuscript version.

**Late Glacial**

Sorry for the misspelling. This has been corrected in line 510 in the revised version.

What do you mean by "small grain-size fining"?

Sorry for this poor wording. We meant a small reduction in sediment mean grain-size. We have subsequently adopted this wording to make it clearer so that line 519 reads as follows: "A reduction in accumulation rates ca. 12.6 to ca. 11.5 cal. ka BP, broadly associated with **a small reduction in sediment mean grain-size......**".

**Younger Dryas**

Thank you for noticing the lack of capitalisation of "Younger". It has been corrected for in the revised manuscript version.

This sentence does not provide any real information - what do you mean by a "more limited Younger Dryas event"? Is it meant as a glacial event, a climatic event - clarify this!

Thank you for this comment. We have phrased this differently in the revised manuscript version and have removed the "more limited Younger Dryas event" as it does not add any real information as you stated. We have also corrected for the Kokorowski paper as we were referring to the Kokorowski et al. 2008b paper here. As such lines 523 and 524 now read as follows: "These findings are consistent with recent regional and transregional records that suggest a **spatially variable Y**ounger Dryas **climatic event** in **F**ar and **E**ast Russia and parts of **E**astern Beringia (Anderson & Lozhkin, 2015; Kokorowski et al., 2008b; Lozhkin & Anderson, 2013; Lozhkin et al., 2018)". Moreover, we have added extra discussion from literature sources from Burial Lake, Alaska, that discussed the possibility of limited support for a Younger Dryas climatic event may also partially reflect seasonal differences in the sensitivity of physical and geochemical proxies to climatic or environmental change, as has been discussed in studies of sediment cores from Burial Lake, Alaska (Finkenbinder et al., 2015)."

**Younger Dryas, Far and East Russia, Eastern Beringia**

Sorry for these misspellings. They have been corrected in the revised manuscript version and have been addressed in the response to the previous comment. We have also checked for correct capitalisation of these words throughout the manuscript.

So, why you have put the boundary between LU-II and LU-I to the higher TOC value? Because the organic proxy values are decreasing now, as the highest value is at the boundary.

Thank you for the comment. We have addressed the boundary between LU-II and LU-I in previous responses. The boundary between LU-II and LU-I has now been shifted to **346** cm (**11.69** cal. ka BP) within the revised manuscript version and hence the highest TOC values will no longer be directly at the boundary between LU-II and LU-I as was previously the case. Please be aware that we will also correct all the ages referring to the previous LU-II and LU-I boundary within section 5.1.3 to account for the movement of the boundary position to 346 cm.

**(Figs 5, 6, 7) - no dot after Figs and spaces after comma**

Many thanks for pointing this out. We have followed your suggestions and removed the dot and added spaces after each comma in line 529.

**...for a local Holocene thermal maximum...**

We agree that we should refer to a "Local" Holocene thermal maximum. We thus accepted your suggestion and have changed this within the revised manuscript version so that lines 541 and 542 read as follows: "which show evidence for a **loca**l Holocene thermal maximum ca. 10.6–7 cal. ka BP (Andreev et al., 2021).".

**...greater sand proportion may relate... why so complicated?**

Sorry for making this sound unnecessarily complicated. We have subsequently restructured lines 543 and 544 to make them easier to read by removing "**to the grain-size distribution**". The lines thus read as follows: "Increasing early Holocene sediment and mass accumulation rates alongside greater sand contribution may relate to the input of coarser grained fluvial

detrital material from a paraglacial.....". We have also added additional reference to support the input of coarser grained material within this section, as well as a potential limitation of the Si/Al ratio due to diatoms that reads as follows: "This is supported by increasing Zr/K and Si/Al ratio values during the early Holocene that support coarser grain sizes that may relate to enhanced input of sand and coarser silts (cf. Píšková et al., 2019). It must however be noted, that higher Si/Al ratio values may also additionally reflect abundant diatoms within Holocene sediments (Procházka et al., 2019)."

**Late Glacial**

Sorry for the misspelling. This has been corrected in line 547 in the revised version.

**Late Glacial**

Sorry for the misspelling. This has been corrected in line 551 in the revised version.

**Westerlies**

Thank you for noticing. We have corrected this in line 554.

**Last Glacial Maximum**

It has now been corrected for by capitalisation in the revised manuscript in lines 554 and 555. All occurrences were checked and capitalised where necessary.

**?finer grain-size**

Thank you for the suggestion. We have now altered "lower grain-size values" to "finer grainsize" so that lines 563 and 564 read as follows: "alongside **finer grain-size** may reflect some local environmental change".

**Why is the reference given twice in one sentence?**

This was a referencing mistake. We have removed the second occurrence of the reference on line 584 (Francke et al., 2013) to correct for this.

?a shortening of the summer open water season.

Thanks for the suggestion. We agree with your wording and have thus changed line 588 to the following "This could tentatively be interpreted to represent a **shortening of the** summer open water season."

**... coarser fluvial and alluvial detrital material...**

Many thanks for the suggestion. We agree with your suggestion and have adopted the phrasing so that line 594 will read as follows: "paraglacial processes that resulted in an input source of **coarser fluvial and alluvial detrital material** into the southern sub-basin...".

again, Peter Doran have not worked here - cite it as follows: cf. Doran, 1993...

Sorry for this. We have changed the citation to your suggestion so that line 595 will read "mass accumulation rates (**cf.** Doran, 1993; Smith and Jol, 1997)."

Figs

Thanks for pointing this out. We have removed the decimal point from line 597 so that it reads as follows: "front (site EN18220) (Fig. 1 & **Figs** S5, S6)". As aforementioned, we have also checked the entire manuscript for similar incidences and corrected for these too.

....sediment transport into the deeper...

We accept your suggestion and have changed line 608 to read as follows: "This likely represents low deposition due to **sediment transport into the deeper basin** and feasibly......".

Are really the lakes in Greenland boreal?

Thank you for the question. The use of Boreal should have referred to lakes studied in North America and northern Europe and not to Greenland, where the studied lakes were predominantly of proglacial or bedrock-catchment type (Perren et al., 2009). We have subsequently altered line 616 to read as follows: "Comparisons must therefore be additionally drawn to boreal and glacial lakes from North America and northern Europe, as well as to **proglacial and bedrock-catchment lakes from Greenland**." We have also checked the usage of "boreal" for Greenlandic lakes throughout the manuscript. In addition, we have included reference to the major differences between the comparison regions directly after to read as follows: "Significant differences however exist between these lake systems relating to contrasting environmental conditions prevailing at different latitudes as well as high variability with regards to lake and catchment spatial extent, lake water depth and catchment environmental and vegetation properties."

**ice free**

Thanks for noticing. We have now remove the capitalisation in line 627 so that is reads as follows: "accumulation calculated for Finnish Boreal lakes that became **ice free** at the Holocene start".

Great figure to compare boreal to polar northern hemisphere lakes' OCAR!

Thank you very much for this nice comment to figure 8! We are super happy that this study provides a northern Sentinel for looking at OCARs in Siberia.

yedoma - lower case sediments - lower case

Sorry for the incorrect capitalsation within the figure caption of figure 9. We have now removed the capitalisation in the revised manuscript version so that it reads as follows: "......Bykovsky thermokarst lagoons and Central Yakutian **yedoma** deposits. Rauchuagytgyn **sediments** possess......".

**I do not think that yedoma and alas are local names, therefore they should be written with lower case initials.**

We agree with your suggestion as yedoma and alas refer to permafrost deposits and not to areas. We have hence removed the capitalisation of "yedoma" and "alas" in lines 662 and 663 so that they now read "(**yedoma**: 0.057 Mt, 5.27 kg m-3, **alas**: 0.032 Mt, 6.07 Kg m-3)". These terms have been checked for and corrected throughout the manuscript. Please also note that some values regarding the carbon pools and densities presented by Jenrich et al. accepted have been altered slightly. This comes following the correction of a calculation

mistake within their manuscript. We thus use the updated values here within the text and figures.

**kg - lower case**

Sorry for this. It has been corrected in the revised manuscript version in line 663.

**Aeolian**

This has been corrected in line 685 to remove the capitalisation.

**What about to apply the geomorphological concept of connectivity?**

Thank you for this great suggestion. We feel however that applying this concept fully would be beyond the scope of the current manuscript. That being said, we have added a mention of this concept within section 5.2.3 as well as an additional reference to the paper of Singh et al. 2021 "Geomorphic connectivity and its application for understanding landscape complexities: a focus on the hydro-geomorphic systems of India". We will consider applying this concept also within future studies!

The geomorphological concept of connectivity has been mentioned from lines 704 to account for this as follows: "As geomorphic systems are hierarchical and operate at multiple spatio-temporal scales according to the concept of geomorphic connectivity, the diverse linkages and interrelationships between different catchment components and processes likely plays a large and complex role in regulating sediment and carbon dynamics at Lake Rauchuagytgyn (Singh et al., 2021)".

...future changes in sediment and carbon...

Thank you for noticing this mistake. We have now changed line 709 to read as follows: "....could lead to future changes **in** sediment and carbon dynamics that are yet.....".

two times within in the sentence - maybe to changes as follows: ...palaeoenvironmental context of a Chukotkan...

We have followed your suggestion and subsequently adjusted lines 724 and 725 as follows: "This study aimed to improve the understanding of accumulation rates and pools within a palaeoenvironmental context **of a** Chukotkan Arctic glacial lake......".

**See my comment in the abstract for the timing of the MIS2**

Thank you for this. We have addressed this in previous responses and have followed your suggestions for this. As such, line 733 reads "Mid MIS2-early MIS1 accumulation (ca. 23.4–11.69 cal. ka BP) reflects the increasing influence of paraglacial processes, longer surface ice-free summers, a thickening catchment active layer, and increasing moisture availability. Carbon accumulation increased throughout and accompanied progressive climate amelioration."

**References used in this author response**

Bakke, J., Lie, øyvind, Nesje, A., Dahl, S.O., Paasche, ø.: Utilizing physical sediment variability in glacier-fed lakes for continuous glacier reconstructions during the Holocene,

northern Folgefonna, western Norway, Holocene, 15, 161–176. https://doi.org/10.1191/0959683605hl797rp, 2005.

Bronk Ramsey, C.: Deposition models for chronological records, Quat. Sci. Rev., 27, 42-60, https://doi.org/10.1016/j.quascirev.2007.01.019, 2008.

Dietze, E., Maussion, F., Ahlborn, M., Diekmann, B., Hartmann, K., Henkel, K., Kasper, T., Lockot, G., Opitz, S., and Haberzettl, T.: Sediment transport processes across the Tibetan Plateau inferred from robust grain-size end members in lake sediments, Clim. Past, 10, 91–106, https://doi.org/10.5194/cp-10-91-2014, 2014.

Finkenbinder, M.S., Abbott, M.B., Edwards, M.E., Langdon, C.T., Steinman, B.A., Finney, B.P.: A 31,000 year record of paleoenvironmental and lake-level change from Harding Lake, Alaska, USA, Quat. Sci. Rev., 87, 98–113, https://doi.org/10.1016/j.quascirev.2014.01.005, 2014.

Friedman, G.M. and Sanders, J.E.: Principles of Sedimentology, Wiley, New York, 1978.

Gaglioti, B. V., Mann, D. H., Jones, B. M., Pohlman, J. W., Kunz, M. L., and Wooller, M. J.: Radiocarbon age-offsets in an arctic lake reveal the long-term response of permafrost carbon to climate change, J. Geophys. Res. Biogeosciences, 119, 1630–1651, https://doi.org/10.1002/2014JG002688, 2014.

Karlen, W and Matthews, J.A.: Reconstructing Holocene Glacier Variations from Glacial Lake Sediments: Studies from Nordvestlandet and Jostedalsbreen-Jotunheimen, Southern Norway, Geografiska Annaler. Series A, Physical Geography, 74, 327–348, 1992.

Kilian, R., Baeza, O., Breuer, S., Ríos, F., Arz, H., Lamy, F., Wirtz, J., Baque, D., Korf, P., Kremer, K., Ríos, C., Mutschke, E., Simon, M., De Pol-Holz, R., Arevalo, M., Wörner, G., Schneider, C & Casassa, G.: Late Glacial and Holocene Paleogeographical and Paleoecological Evolution of the Seno Skyring 1060 and Otway Fjord Systems in the Magellan Region, Anales Instituto Patagonia (Chile), 41, 5–26, http://dx.doi.org/10.4067/S0718-686X2013000200001, 2013.

Leemann A, Niessen F.: Holocene glacial activity and climatic variations in the Swiss Alps: reconstructing a continuous record from proglacial lake sediments, The Holocene, 4(3): 259-268, doi:10.1177/095968369400400305, 1994b.

Lougheed, B. C. and Obrochta, S. P.: A Rapid, Deterministic Age-Depth Modeling Routine for Geological Sequences With Inherent Depth Uncertainty, Paleoceanogr. Paleoclimatology, 34, 122–133, https://doi.org/10.1029/2018PA003457, 2019.

Singh, M., Sinha, R., Tandon, S.K.: Geomorphic connectivity and its application for understanding landscape complexities: a focus on the hydro-geomorphic systems of India, Earth Surf. Process. Landf., 46, 110-130, 2020.

Stuiver, M., Reimer, P. J., and Reimer, R. W.: CALIB 8.2, http://calib.org, 2020.

Van der Bilt, W., Bakke, J., Vasskog, K., D´Andrea, W.J., Bradley, R.S., Ólafsdóttir, S.: Reconstruction of glacier variability from lake sediments reveals dynamic Holocene climate in Svalbard. Quat. Sci. Rev., 126, 15, 201–218. https://doi.org/10.1016/j.guascirev.2015.09.003, 2015.